# PAL: Sample-Efficient Personalized Reward Modeling for Pluralistic Alignment

**Daiwei Chen, Yi Chen, Aniket Rege, Zhi Wang, and Ramya Korlakai Vinayak**
University of Wisconsin–Madison, Madison, WI, USA

## Abstract

Foundation models trained on internet-scale data benefit from extensive alignment to human preferences before deployment. However, existing methods typically assume a homogeneous preference shared by all individuals, overlooking the diversity inherent in human values. In this work, we propose a general reward modeling framework for pluralistic alignment (**PAL**), which incorporates diverse preferences from the ground up. PAL has a modular design that *leverages commonalities* across users while catering to *individual personalization, enabling efficient few-shot localization* of preferences for new users. Extensive empirical evaluation demonstrates that ***PAL matches or outperforms state-of-the-art methods on both text-to-text and text-to-image tasks***: on Reddit TL;DR Summary, PAL is 1.7% more accurate for seen users and 36% more accurate for unseen users compared to the previous best method, **with** $100\times$ **less parameters**. On Pick-a-Pic v2, PAL is 2.5% more accurate than the best method with $156\times$ fewer learned parameters. Finally, we provide theoretical analysis for generalization of rewards learned via PAL showcasing the reduction in number of samples needed per user. Our code is publicly available at https://github.com/RamyaLab/pluralistic-alignment.

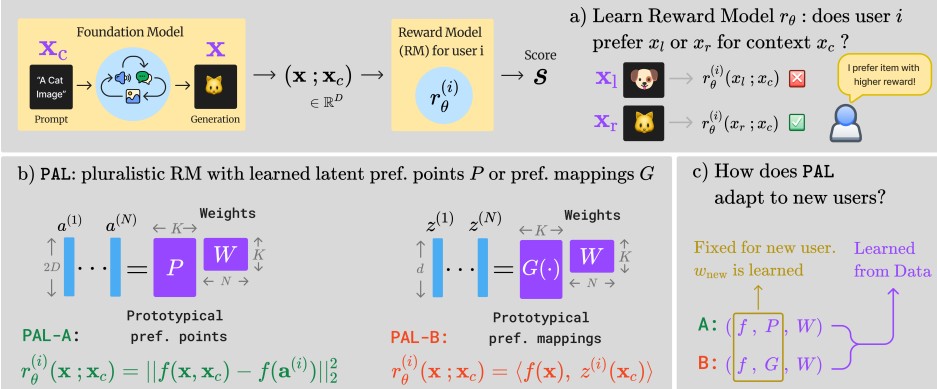

Figure 1: (a) Using preference data, the PAL framework learns a personalized reward model for each user $i$, $r_\theta^{(i)}(\cdot)$, which captures the user's preference for any output $\mathbf{x}$ given context $\mathbf{x}_c$. (b) PAL models the common perceptions of similarity across users through a shared representation $f(\cdot)$, and represents the individual aspects of preferences via either a preference point $\mathbf{a}^{(i)}$ in PAL-A or a preference mapping $z^{(i)}(\mathbf{x}_c)$ in PAL-B. In particular, we assume a *low-rank* structure with $K$ prototypical preference points or preference mappings; see Section 2 for details. (c) PAL enables efficient few-shot preference learning for a new user—only a $K$-dimensional weight vector is learned. This reduces computational cost as well as data needed for generalization.

## 1 Introduction

Foundation models trained on internet data are often not readily deployable and undergo *alignment* to human preferences using large amounts of *pairwise comparison* feedback (Ouyang et al., 2022). While aligning AI/ML models to human preferences, it is important to consider *whose preferences are we aligning them to*? The status quo for popular alignment frameworks is to assume a homogeneous preference shared by all humans. However, humans have diverse preferences, values and opinions (Bakker et al., 2022; Durmus et al., 2024; Nadal & Chatterjee, 2019; Wildavsky, 1987). The need to capture this *plurality* in the context of AI alignment was also highlighted recently by Sorensen et al. (2024). However, the methods suggested therein and other recent works look at

learning multiple rewards with a top-down approach, where the system designer decides the number and axes that one should care about (Cheng et al., 2023; Kovač et al., 2023; Ouyang et al., 2022; Santurkar et al., 2023), e.g., helpfulness vs. harmlessness (Bai et al., 2022b;a; Ganguli et al., 2022; Rame et al., 2024). In reality, human preferences are more complex than the designer-specified axes (Bakker et al., 2022), especially on subjective aspects, which leads us to propose the following goal.

> **Goal:** Develop a sample-efficient, personalized reward modeling framework for pluralistic alignment from the ground up which learns and generalizes to heterogeneous preferences.

**Our Contributions.** Towards this goal, we make the following contributions:

1. We propose `PAL`, a **novel, sample-efficient, personalizable reward modeling framework for pluralistic alignment** from the ground up (Section 2). Our modular and versatile framework achieves **superior performance to the state-of-the-art** (SoTA) in both language (Section 3.1) and vision (Section 3.2, 3.3) tasks, *while utilizing only a fraction of their learnable parameters*.

2. `PAL` reward models **achieve competitive performance with simple 2-layer MLPs** on top of frozen foundation models of varying sizes **in practical settings across modalities** (Section 3). `PAL` enables democratic alignment via strong accuracy-compute optimality (Rege et al., 2023).

3. `PAL` is complementary to existing alignment frameworks, and works seamlessly across compute budgets (see Figure 2) demonstrating the strength of its **flexible and modular design**.

4. We provide **sample complexity guarantees for generalization** towards new preference predictions for users in the dataset as well as for unseen users via few-shot learning, in the fully connected linear layer setting for one of our models (Section 4.1), and we verify these results with extensive numerical simulations (Section 4.2).

The learned `PAL` reward models can be used flexibly for personalizing downstream task of alignment through (i) train-time methods such as PPO-based RLHF (Christiano et al., 2017; Wu et al., 2023a) and (ii) inference time methods via best-of-n sampling such as controlled decoding (Liu et al., 2024; Mudgal et al., 2024). In addition, the modular nature of `PAL` has high potential for future adaptability as it enables seamlessly updating learned reward models via switching data encoders and distance metrics, or adding new prototypes to account for dynamically changing heterogeneous preferences. We note that, in this paper, we focus on developing sample-efficient personalizable reward modeling, and understanding its efficacy via extensive experiments and theoretical analysis.

**Background.** Our reward modeling builds on the popular preference learning models, the ***Bradley-Terry-Luce (BTL) model*** (Bradley & Terry, 1952) and the ***ideal point model*** (Coombs, 1950), both of which are special cases of the *linear stochastic transitivity* (LST) models. We provide a brief discussion of these models, including their assumptions and limitations here. Let $D$ denote the dimension of the representation space of the foundation models. Given a context/prompt $\mathbf{x}_c \in \mathbb{R}^D$, generative model(s) can produce different outputs, denoted by $\mathbf{x} \in \mathbb{R}^D$.[1] The LST models make the following assumption: Suppose each output for the prompt is associated with a *true* score $s^\star(\mathbf{x}; \mathbf{x}_c) \in \mathbb{R}$. Given any list of alternates, the true scores lead to a *true* ranking of them: for any two alternates, $\mathbf{x}_l$ is preferred over $\mathbf{x}_r$ if $s^\star(\mathbf{x}_l; \mathbf{x}_c) > s^\star(\mathbf{x}_r; \mathbf{x}_c)$. However, when we elicit comparison feedback from humans, the answers may be *noisy* and are modeled as, $\Pr(\mathbf{x}_l \succ \mathbf{x}_r | \mathbf{x}_c) = h(s^\star(\mathbf{x}_l; \mathbf{x}_c) - s^\star(\mathbf{x}_r; \mathbf{x}_c))$, where $h : \mathbb{R} \to [0, 1]$ is a strictly monotonic *link function* that satisfies $h(z) = 1 - h(-z)$. In other words, $\Pr(\mathbf{x}_l \succ \mathbf{x}_r | \mathbf{x}_c) = 1/2$ when $s^\star(\mathbf{x}_l, \mathbf{x}_c) = s^\star(\mathbf{x}_r, \mathbf{x}_c)$ and $\Pr(\mathbf{x}_l \succ \mathbf{x}_r | \mathbf{x}_c) > 1/2$ when $s^\star(\mathbf{x}_l, \mathbf{x}_c) > s^\star(\mathbf{x}_r, \mathbf{x}_c)$.

The BTL model uses the logistic sigmoid function as the link function:

$$\Pr(\mathbf{x}_l \succ \mathbf{x}_r | \mathbf{x}_c) = \frac{1}{1 + \exp\left(s^\star(\mathbf{x}_r; \mathbf{x}_c) - s^\star(\mathbf{x}_l; \mathbf{x}_c)\right)}. \tag{1}$$

On the other hand, the ideal point model uses a latent vector $\mathbf{a} \in \mathbb{R}^D$ to denote the *ideal* preference point of a user, along with (negative) distance-based scoring function; the model is given by,

$$\Pr(\mathbf{x}_l \succ \mathbf{x}_r | \mathbf{x}_c) = h(\mathrm{dist}^2(\mathbf{x}_r, \mathbf{a}) - \mathrm{dist}^2(\mathbf{x}_l, \mathbf{a})), \tag{2}$$

where $h$ can be any valid link function. The key idea here is that the larger the difference in distances between the alternates to the ideal point, the easier it is to choose between them, and hence the

---

[1]These representations can be taken from penultimate layer(s) of a foundation model. While we use the same $D$ for the prompt and output for simplicity, this can easily be extended to different dimensional spaces.

answer is less noisy. We note that with a sigmoid link function, the ideal point model can be viewed as the BTL model with a distance-based scoring function.

A key limitation of these approaches is that they *assume a single true ranking of the alternates* and model the differences in elicited preference as noisy observations. In reality, the differences in elicited preferences are not merely noise, instead a reflection of plurality of human preferences.

The ideal point model provides a natural starting point to incorporate individual preferences since each user can be modeled using their own latent preference. However, the assumption that we know the distance function that reflects a human notion of similarity of alternates can be too strong in practice. Further, completely individualized models devoid of any shared structure will lead to unnecessary burden on per user sample complexity for learning and would be difficult to generalize. Our aim is to capture the heterogeneity of preferences while also leveraging commonalities (Section 2), allowing for sample efficient learning and generalization.

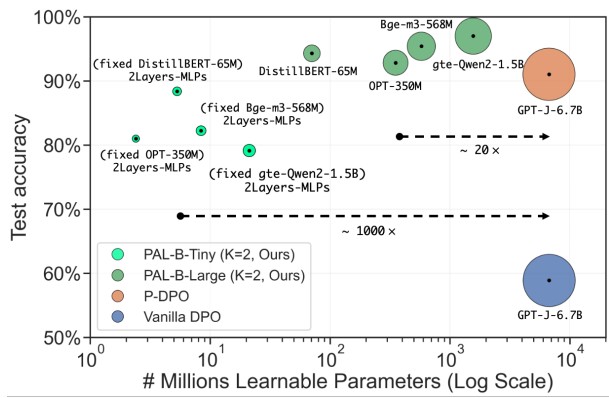

Figure 2: On Reddit TL;DR, `PAL` is accuracy-compute optimal and shows state-of-the-art (SoTA) performance.

## 2  `PAL`: Reward Modeling for Pluralistic ALignment

In this section, we describe our proposed personalizable reward modeling framework that captures commonalities shared across the population which can be learned using the pooled data and individual aspects that is learned per user in a sample efficient way. For user $i$, given a context $\mathbf{x}_c$, the probability of alternate $\mathbf{x}_l$ being preferred over $\mathbf{x}_r$ is as follows,

$$\Pr(\mathbf{x}_l \succ \mathbf{x}_r|\mathbf{x}_c, i) = h^{(i)}(r_\theta^{(i)}(\mathbf{x}_r; \mathbf{x}_c) - r_\theta^{(i)}(\mathbf{x}_l; \mathbf{x}_c)), \tag{3}$$

where $h^{(i)}$ is any valid link function that can depend on the user, and $r^{(i)}(\cdot)$ is the personalized reward function for user $i$.[2] We do not assume the knowledge of the link function for our learning algorithms (Section 2.1). We propose two models for the personalized reward function[3]:

**`PAL`-A: Diverse preferences modeled via latent ideal preference points.** The shared sense of similarity of different alternates being compared is modeled as Euclidean distance in an *unknown* mapped space, captured by $f : \mathbb{R}^{2D} \rightarrow \mathbb{R}^d$ that jointly maps the output and context, $(\mathbf{x}; \mathbf{x}_c)$, to this unknown space. The *latent* preference of each user $i$ is modeled using an *unknown* ideal preference point $\mathbf{a}^{(i)} \in \mathbb{R}^{2D}$. How much the user $i$ values output $\mathbf{x}$ for given context $\mathbf{x}_c$ is modeled as inversely proportional how far away the mapping of $(\mathbf{x}; \mathbf{x}_c)$ is from the user's ideal point. To further capture the commonality in the preferences among users, each user's preference points is modeled as a convex combination of $K$ prototypical points, that is, $\mathbf{a}^{(i)} := \sum_{k=1}^K w_k^{(i)} \mathbf{p}_k$ where the weights $w_k^{(i)} \geq 0$ and $\sum_{k=1}^K w_k^{(i)} = 1$, and $\{\mathbf{p}_1, ..., \mathbf{p}_K\}$ with $\mathbf{p}_i \in \mathbb{R}^{2D}$ are $K$ prototypical ideal preference points. More formally, the personalized reward function and the corresponding personalized probabilistic preference model is given by,

$$\mathbf{PAL\!-\!A:} \quad r_\theta^{(i)}(\mathbf{x}; \mathbf{x}_c) := ||f(\mathbf{x}; \mathbf{x}_c) - f(\mathbf{a}^{(i)})||_2^2, \tag{4}$$

$$\Pr(\mathbf{x}_l \succ \mathbf{x}_r|\mathbf{x}_c, i) = h(||f(\mathbf{x}_r; \mathbf{x}_c) - f(\mathbf{a}^{(i)})||_2^2 - ||f(\mathbf{x}_l; \mathbf{x}_c) - f(\mathbf{a}^{(i)})||_2^2), \tag{5}$$

Denoting $\mathbf{P} := [\mathbf{p}_1, \cdots, \mathbf{p}_K]$, each user's preference point $\mathbf{a}^{(i)} := \mathbf{P}\mathbf{w}^{(i)}$, where $\mathbf{w}^{(i)} := [w_1^{(i)}, \cdots, w_K^{(i)}]^\top$, lies in the $(K-1)$-dimensional simplex, $\Delta^{K-1}$.

---

[2]We drop the superscript $i$ on $h$ denoting user specificity for simplicity in the rest of discussions, however, we note that the link function need not be the same for all users and our learning methods are agnostic to them.

[3]In Appendix B.2, we discuss when to choose one model over the other in practice.

**`PAL-B`: Diverse preferences modeled via latent preference mappings.** Here each user's preference also incorporates the given context and is modeled using an *unknown preference mapping* $z : \mathbb{R}^D \to \mathbb{R}^d$. The shared sense of similarity of different alternates being compared is modeled as cosine similarity in an *unknown* mapped space, $f : \mathbb{R}^D \to \mathbb{S}^{d-1}$. The commonality in the preferences among users is captured by modeling each user's preference mapping as a convex combination of $K$ prototypical mappings, i.e., $z^{(i)}(\mathbf{x}_c) = \sum_{k=1}^{K} w_k^{(i)} g_k(\mathbf{x}_c)$, where $\{g_1, ..., g_K\}$ with $g_k : \mathbb{R}^D \to \mathbb{S}^{d-1}$ are the prototypical mappings, $w_k^{(i)} \geq 0$ and $\sum_{k=1}^{K} w_k^{(i)} = 1$. Formally, the personalized reward function and the corresponding probabilistic preference model is given by,

$$\texttt{PAL-B:} \quad r_\theta^{(i)}(\mathbf{x}; \mathbf{x}_c) := \langle f(\mathbf{x}), z^{(i)}(\mathbf{x}_c) \rangle, \tag{6}$$

$$\mathrm{Pr}_i(\mathbf{x}_l \succ \mathbf{x}_r | \mathbf{x}_c) = h\left( \langle f(\mathbf{x}_r), z^{(i)}(\mathbf{x}_c) \rangle - \langle f(\mathbf{x}_l), z^{(i)}(\mathbf{x}_c) \rangle \right), \tag{7}$$

For any context $\mathbf{x}_c$, denoting $\mathbf{G}(\mathbf{x}_c) := [g_1(\mathbf{x}_c), \cdots, g_K(\mathbf{x}_c)]$, each user's preference mapping of the given context is, $z^{(i)}(\mathbf{x}_c) := \mathbf{G}(\mathbf{x}_c)\mathbf{w}^{(i)}$ with $\mathbf{w}^{(i)} \in \Delta^{K-1}$.

`PAL` modeling framework is **modular**, i.e., it provides a *systematic way* to incorporate *shared and personalized portions of preferences*, as well as transparent way to control multiple notions of complexity via cross validation: (i) the complexity of mapping $f$ (and $g$'s in `PAL`-B) captures the shared human notion of similarity between alternates. If the underlying foundation model used to obtain the representations of outputs and contexts ($\mathbf{x}$ and $\mathbf{x}_c$) are rich and semantically meaningful, then much smaller models suffice in capturing the rewards (Figure 2); (ii) the number of prototypes $K$ capture the level of heterogeneity of human preferences in the dataset: more diverse preferences mandate a larger $K$ for good generalization (Figure 5(b); and (iii) in conjunction with the prototypes, the individualized weights allow for personalization with much fewer samples per user for both seen and unseen users. This reduces the annotation burden on individuals during data collection (Figure 5(c)) and the amount of samples needed for few-shot learning for new users arriving on deployment (Figure 3). We illustrate the `PAL` framework in Figure 1 and Figure B.2 (Appendix B).

## 2.1 LEARNING `PAL` MODELS FROM DIVERSE PREFERENCES

Let $\mathcal{D} := \left\{ \{(\mathbf{x}_l, \mathbf{x}_r; \mathbf{x}_c, y)_j^{(i)}\}_{j=1}^{m_i} \right\}_{i=1}^{N}$ be a dataset of preference comparisons, where $m_i$ denotes the number of pairs answered by user $i$, and $y$ is the answer given to the pair ($y = -1$ if $\mathbf{x}_l$ is preferred, $y = 1$ otherwise). This can be looked at as a supervised learning problem with binary labels. The **goal** of the learning algorithm in the `PAL` framework is to learn the mappings and prototypes shared across the population, and for each user $i$, the weights $\mathbf{w}^{(i)} \in \Delta^{K-1}$. For `PAL`-A, the mapping $f$ and the prototypes $\{\mathbf{p}_k\}_{k=1}^{K}$ are shared, while for `PAL`-B, the mapping $f$ and the prototype mappings $\{g_k\}_{k=1}^{K}$ are shared. These shared portions are learned using data pooled from all users while the user specific weights are learned using each individual user's preferences.

Given the comparison dataset $\mathcal{D}$, loss function $\ell : \mathbb{R} \to [0, 1]$, model class for $f_\theta$ and prototypical mappings $\{g_1, ..., g_K\}$, the learning algorithm for `PAL`-B starts by randomly initializing these functions, and user weights $\mathbf{w}^{(i)} \in \Delta^{K-1}$, $i = 1, ..., N$. Then, in each iteration until convergence criteria, the following steps are repeated,

- **Sample** a random mini-batch $\left\{ (\mathbf{x}_l, \mathbf{x}_r; \mathbf{x}_c, y)_j^{(i)} \right\}$ of comparison data from $\mathcal{D}$.
- For each comparison $j$ from user $i$:
  - **Compute user ideal mappings:** $z^{(i)}(\mathbf{x}_c) := [g_1(\mathbf{x}_c) \ \ldots \ g_K(\mathbf{x}_c)] \cdot \mathbf{w}^{(i)}$.
  - **Compute distances**: $d_{l,j}^{(i)} = \langle f_\theta(\mathbf{x}_l), z^{(i)}(\mathbf{x}_c) \rangle$, $d_{r,j}^{(i)} = \langle f_\theta(\mathbf{x}_r), z^{(i)}(\mathbf{x}_c) \rangle$.
  - **Compute loss**: $\psi_j^{(i)}(\mathbf{x}_l, \mathbf{x}_r; \mathbf{x}_c, y) = \ell\left( y \cdot (d_{l,j}^{(i)} - d_{r,j}^{(i)}) \right)$.
- **Update Step:** $\mathrm{argmin}_{\theta, \{g_1, ..., g_K\}, \{\mathbf{w}^{(i)}\}_{i=1}^{N}} \sum_{i,j} \psi_j^{(i)}(\mathbf{x}_l, \mathbf{x}_r; \mathbf{x}_c, y)$.

Learning steps are similar for `PAL`-A. See Appendix B for pseudocode details.

## 2.2 GENERALIZATION OF PREFERENCE PREDICTIONS FOR *seen* VERSUS *unseen* USERS

When learning a reward function from diverse preferences, there are two types of generalization to consider: Predicting preferences for (1) *unseen pairs* for *seen users*, i.e., the people for whom the

weights have already been learned from the training data; (2) *unseen users*, i.e., people whose data was not part of the training data at all. For such new users, a portion of their data will be used to learn weights to localize them within the learned model while keeping the shared mappings and prototypes fixed. We also note that the weighted combination of the prototypes, i.e., an average of all the seen users, can be used as the *zero-shot* ideal point for new users (e.g. Netflix recommendations for new accounts). However, we emphasize that it is important for reward functions to generalize to *unseen* users and PAL provides a natural way to localize new users, as we demonstrate in Section 3.1. Section 4.1 provides theoretical guarantees on the per-user sample complexity of PAL for few-shot generalization to unseen users.

## 3 EXPERIMENTS ON REAL DATASETS

In this section, we verify the following claims through extensive empirical evaluation:

**C1.** PAL can effectively capture the diversity of user preferences and outperform status-quo homogeneous reward models.

**C2.** Compared to existing pluralistic reward modeling methods, PAL can achieve state-of-the-art (SoTA) performance with significantly fewer parameters.

**C3.** PAL works for both text-to-text (T2T) and text-to-image (T2I) tasks.

**C4.** PAL demonstrates strong few-shot generalization to new, unseen users.

We employ two strategies for defining and training the learnable mapping $f$ between the embeddings from a foundation model and latent space. In the Tiny strategy, $f$ is a simple two-layer MLP operating on a frozen foundation model. In the Large strategy, $f$ is a combination of the foundation model and a two-layer MLP, with both components being learnable. Models employing these strategies are referred to as PAL-A-Tiny/ PAL-B-Tiny and PAL-A-Large/ PAL-B-Large respectively. Appendix D.1 describes the general training procedure of our algorithm.

We perform experiments on datasets in different domains: the Reddit TL;DR Summary dataset (Section 3.1) and the Pick-a-Pic dataset (Section 3.2). Due to limitations in currently available datasets for pluralistic alignment, which we highlight in Section 3.2, we also created semi-synthetic datasets, Pick-a-Filter (Section 3.3) and Persona (Appendix D.4), to further validate the above claims.

### 3.1 REDDIT TL;DR SUMMARY (TEXT-TO-TEXT)

**Dataset.** Reddit TL;DR Summary dataset curated by Stiennon et al. (2020) contains a series of preferences over summaries generated by language models. For each pair of summaries, $\mathbf{x}_l$ and $\mathbf{x}_r$, a labeler $i$ determines if $\mathbf{x}_l$ is preferred or not. Each pair is also accompanied by the unique identifier of the labeler. We used the variant of the TL;DR dataset proposed by Li et al. (2024), which uses the summary length as the preference. The majority group prefers longer summaries while the minority prefers shorter summaries.

**Setup.** We train PAL with sentence embeddings from foundation models including OPT-350M (Zhang et al., 2022), DistilBERT (Sanh et al., 2019), BGE-M3 (Chen et al., 2024), and gte-Qwen2-1.5B (Li et al., 2023), which make up Large and Tiny variants depending on their size (#parameters). The loss design follows the typical loss of the Reward Model, we use the cumulative loss

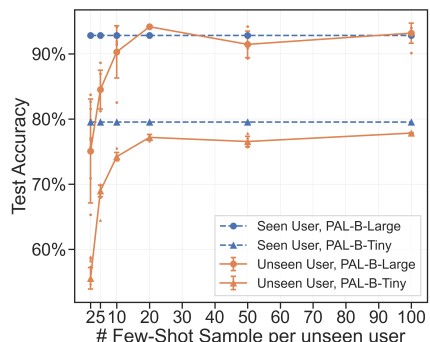

Figure 3: On Reddit TL;DR, PAL generalizes well to unseen data using just 20 samples per unseen user (few-shot).

which weights the per-token reward loss. Details of the loss function, hyperparameter setting, unseen dataset, and training setup are deferred to Appendix D.3.

**Baselines.** We compare to personalized (Li et al., 2024) and vanilla DPO (Rafailov et al., 2024).

**Results.** We train our model with 5 different seeds and report the mean and standard deviation. Figure 2 shows the performance of PAL with foundation models of different sizes. PAL shows strong accuracy-compute optimality with no hyperparameter tuning: compared to SoTA, PAL-B-Large ( gte-Qwen2-1.5B) is 5.9% more accurate on seen users with $4\times$ less parameters while PAL-B-Tiny (DistilBERT) is on-par with $1000\times$ fewer parameters. Furthermore,

Table 1: On Reddit TL;DR Summary, `PAL-B-Large` outperforms SoTA P-DPO on seen users (**+1.7%**) and unseen users (**+36%**) with 6.3 billion fewer parameters. Note that we use only 10 samples per unseen user to localize their weights.

| Model | Seen Acc (%) | Unseen Acc (%) |
|---|---|---|
| Vanilla DPO | 58.91 | 55.37 |
| P-DPO Individual | 91.04 | 55.34 |
| P-DPO Cluster ($K = 5$) | 91.12 | 54.55 |
| `PAL-B-Tiny` ($K = 2$) | $79.54 \pm 0.54$ | $74.72 \pm 0.54$ |
| **`PAL-B-Large`** ($K = 2$) | $92.82 \pm 0.95$ | $91.63 \pm 0.54$ |

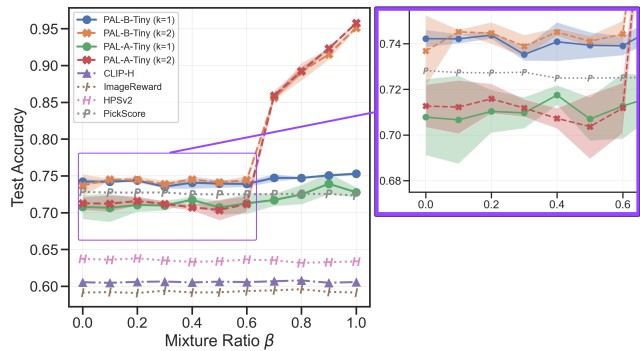

Figure 4: On Pick-a-Filter, `PAL-B-Tiny` outperforms homogeneous models as user groups become more heterogeneous ($\uparrow \beta$).

Figure 3 illustrates `PAL`'s ability to generalize effectively to unseen users in few-shot settings. As the number of samples per unseen user increases, `PAL` progressively adapts to their preferences. With just 20 samples per unseen user, `PAL` achieves performance comparable to that of seen users. Additionally, as depicted in Figure D.3 (Appendix D), `PAL` exhibits superior performance over baseline models on unseen users, even when provided with as few as 2 samples.

Table 1 reports the performance of `PAL-B-Large` (`OPT-350M`) compared to SoTA P-DPO ( (Li et al., 2024)). We observe that `PAL`, with around 6.3 billion fewer parameters, is 1.7% more accurate on seen users, and 36% more accurate on unseen users (**C2, C4**). Figure 3 shows that for new users, `PAL` can match seen user performance with only 20 samples, showing its promising potential to flexibly adapt to new users. We further highlight `PAL`'s strong few-shot generalization to unseen users compared to baselines in Figure D.3 in the Appendix. Lastly, we also show exhaustive results for all our model configurations based on `OPT-350M` in Table D.1 and D.3 (Appendix D.3).

## 3.2 PICK-A-PIC (TEXT-TO-IMAGE)

In this section, we examine how agnostic `PAL` is to data modality (**C3**) via the Pick-a-Pic dataset (Kirstain et al., 2024). Popular T2I reward models usually require fine-tuning foundation models with billions of parameters (Wu et al., 2023b; Kirstain et al., 2024; Xu et al., 2024). We show that `PAL` achieves competitive performance or SoTA performance while having significantly fewer parameters than baselines (**C2**).

**Dataset.** The Pick-a-Pic dataset (Kirstain et al., 2024) is a large, open dataset for human feedback in T2I generation. There are two versions of Pick-a-Pic, v1 and v2, where v2 extends v1. To ensure a fair model evaluation, we divide the v2 test set into "no-leakage" and "leakage" subsets due to overlap ("leakage") with the v1 train set. We only consider the 18391 samples with no preference ties, i.e. one generated image is always preferred to the other. Out of these, 10587 samples ($\sim 58\%$) overlap with the training and validation sets of v1, which was used to train PickScore Kirstain et al. (2024) – we call this the v2 "leakage" subset. The remaining 7804 test samples ($\sim 42\%$) in v2 do not overlap with v1 train and val, ensuring they are distinct for evaluation purposes – we call this the v2 "no-leakage" subset.

**Setup.** We train `PAL-B` with logistic loss on both v1 and v2 for 10 epochs on top of CLIP-H/14 or PickScore (Kirstain et al., 2024) embeddings. We discuss hyperparameter tuning in Appendix D.5.

**Baselines.** We compare to SoTA PickScore (Kirstain et al., 2024) and a vanilla CLIP-H/14.

**Results.** Table 2 shows (a) `PAL` matches SoTA PickScore when trained on V1 with $165\times$ fewer parameters (b) `PAL` exceeds SoTA performance on v2-no-leakage (i.e. fair comparison) by **2%** when training on v1, and by 2.5% if training on v2 (**C1, C3**). Training PAL from "scratch" i.e. with CLIP-H embeddings, outperforms training on PickScore embeddings (which were trained on v1). We note that `PAL-B-Tiny` ($\sim$6M params) exceeds SoTA performance while training on a single RTX 4090 GPU (see Appendix E), whereas PickScore ($\sim$1B params) is trained with $8\times$A100 GPUs – highlighting the suitability of `PAL` for efficient and democratic reward modeling (**C2**).

**Remark.** Since the data collection process for existing datasets involves the usage of strict rubrics (Stiennon et al., 2020; Kirstain et al., 2024; Wu et al., 2023b), labeler performance monitoring (Xu et al., 2024), and a disproportionate amount of data provided by a small fraction of

Table 2: Seen user test accuracy of PAL compared to baselines on Pick-a-Pic v2. Entries with asterisk* have inflated accuracy due to the V2 test set overlap with V1 train (See dataset details in Section 3.2).

| Model | Params | Trainset | V1 Test Accuracy (%) | V2 Test Accuracy (%) | |
|---|---|---|---|---|---|
| | | | | No-Leakage | Leakage |
| CLIP-H/14 | 986M | - | 59.23 | 62.57 | 58.59 |
| ImageReward | 447M | - | 61.10 | - | - |
| HPS v2.1 | 986M | - | 66.70 | - | - |
| PickScore | **986M** | V1 | **71.85** | 68.04 | 74.16* |
| PAL-A-Tiny (CLIP-H) | 8.4M | V1 | $69.29 \pm 0.6$ | - | - |
| PAL-B-Tiny (CLIP-H) | **6.3M** | V1 | **$71.13 \pm 0.3$** | **$70.02 \pm 0.39$** | $79.32 \pm 1.68$* |
| PAL-B-Tiny (CLIP-H) | 6.3M | V2 | - | **$70.51 \pm 0.22$** | $68.67 \pm 0.51$ |
| PAL-B-Tiny (Pickscore) | 6.3M | V2 | - | $70.16 \pm 0.19$ | $74.79 \pm 0.13$* |

users, these datasets may not be heterogeneous. We note that a strict rubric leads to uniformity as it essentially crowdsources the criteria of the rubric instead of eliciting the preferences of the users. Therefore, even using PAL with $K = 1$, we can surpass existing SoTA performance. These results highlight the need for more nuanced approaches to collect datasets that elicit diverse opinions.

### 3.3 PICK-A-FILTER (TEXT-TO-IMAGE)

As PickScore has been shown to lack significant heterogeneity (Kirstain et al., 2024), we artificially inject plurality to create the *Pick-a-Filter* dataset, and show that PAL significantly surpasses homogeneous reward models when pluralistic preferences are present (**C1, C2, C3**).

**Dataset.** Motivated by a natural human color preference distribution Palmer & Schloss (2010), we choose adding different color filters to the generated images as the mechanism by which we explicitly "inject" preference diversity into the v1 dataset (group 1 prefers 'red' tones and group 2 prefers 'blue' tones). To avoid the model latching on purely to color features when learning preferences, we use a hyperparameter called **mixture ratio** $\beta = N_f/N_o$, where $N_f$ is the number of v1 pairs we choose to apply filters on, and $N_o$ is the total number of original v1 pairs. The larger the $\beta$, the more color-filtered v1 pairs in the training set. We show and discuss our careful construction of Pick-a-Filter in Figure D.7 and Appendix D respectively.

**Setup.** We train PAL-B-Tiny with logistic loss on the *Pick-a-Filter* dataset with different mixture ratios. Detailed training setups are deferred to Appendix D.6.

**Baselines.** We compare PAL to CLIP-H, following Kirstain et al. (2024), as well as two strong T2I reward models: ImageReward (Xu et al., 2024) and HPS v2 (Wu et al., 2023b). Note that we cannot compare to PickScore as its train set overlaps with Pick-a-Filter's val set (leakage).

**Results.** Figure 4 shows that PAL-B-Tiny can learn diverse user preference groups across mixture ratios (**C1, C3**). We can view $\beta$ as indicating how much the two user groups prefer their respective color filters (higher $\beta \rightarrow$ affinity for filters). PAL significantly outperforms the homogeneous reward model in predicting user preferences – at $\beta = 1$, PAL achieves 95.2% test accuracy compared to 75.4% from the homogeneous reward model (**C2**).

## 4 SAMPLE COMPLEXITY FOR LEARNING REWARDS UNDER PAL MODELING

In this section, we shed light on the per-user sample complexity of PAL for (1) generalization to unseen pairs involving known users, and (2) few-shot generalization to unseen users. We present theoretical guarantees in Section 4.1, and empirical results from numerical simulations in Section 4.2.

### 4.1 THEORETICAL GUARANTEES

To shed light on the benefits of using mixture modeling approach, we provide theoretical analysis of PAL-A with fully-connected linear layer in neural network mapping $f$. For simplicity of exposition, we subsume the context/prompt vectors into the item embeddings, treating $(\mathbf{x}; \mathbf{x}_c)$ as $\mathbf{x} \in \mathbb{R}^D$. Consider a class linear transformations, $\mathcal{F} \subset \left\{ f : \mathbb{R}^D \rightarrow \mathbb{R}^D \mid f(\mathbf{x}) = \mathbf{A}\mathbf{x}, \mathbf{A} \in \mathbb{R}^{D \times D} \right\}$. Then, the difference of scores between two items for a user with ideal point $\mathbf{a} = \mathbf{P}\mathbf{w}$ is given by

$$\|f(\mathbf{x}_l) - f(\mathbf{a})\|_2^2 - \|f(\mathbf{x}_r) - f(\mathbf{a})\|_2^2 = (\mathbf{x}_l - \mathbf{a})^\top \mathbf{A}^\top \mathbf{A}(\mathbf{x}_l - \mathbf{a}) - (\mathbf{x}_r - \mathbf{a})^\top \mathbf{A}^\top \mathbf{A}(\mathbf{x}_r - \mathbf{a})$$
$$= \mathbf{x}_l^\top (\mathbf{A}^\top \mathbf{A})\mathbf{x}_l - \mathbf{x}_r^\top (\mathbf{A}^\top \mathbf{A})\mathbf{x}_r - 2(\mathbf{x}_l - \mathbf{x}_r)^\top (\mathbf{A}^\top \mathbf{A})\mathbf{P}\mathbf{w}.$$

Observe that the right-hand side of the first equality represents the difference between the squared Mahalanobis distances[4] from $\mathbf{x}_l$ to $\mathbf{a}$ and from $\mathbf{x}_r$ to $\mathbf{a}$, where the Mahalanobis distance is defined by $\mathbf{A}^\top \mathbf{A}$. In other words, the problem is equivalent to simultaneously learning a Mahalanobis distance and the user ideal points (Xu & Davenport, 2020; Canal et al., 2022).

Now, let $\mathbf{M} := \mathbf{A}^\top \mathbf{A}$ and $\mathbf{Q} := -2\mathbf{M}\mathbf{P}$. The difference in scores can then be written as:

$$\|f(\mathbf{x}_l) - f(\mathbf{a})\|_2^2 - \|f(\mathbf{x}_r) - f(\mathbf{a})\|_2^2 = \mathbf{x}_l^\top \mathbf{M} \mathbf{x}_l - \mathbf{x}_r^\top \mathbf{M} \mathbf{x}_r + (\mathbf{x}_l - \mathbf{x}_r)^\top \mathbf{Q}\mathbf{w}.$$

Given these reparameterizations from $f$ to $\mathbf{M}$ and from $\mathbf{P}$ to $\mathbf{Q}$, for the remainder of this section, we assume that the reward modeling problem is defined over $\mathbf{M} \in \left\{ \mathbf{M} \in \mathbb{R}^{D \times D} : \|\mathbf{M}\|_\mathrm{F} \leq \zeta_M, \mathbf{M} \succeq 0 \right\}$, $\mathbf{Q} \in \left\{ \mathbf{Q} \in \mathbb{R}^{D \times K} : \|\mathbf{Q}_k\|_2 \leq \zeta_v \ \forall k \in [K] \right\}$, and $\mathbf{w}^{(i)} \in \Delta^{K-1}$ for each user $i$. In addition, we let $\ell : \mathbb{R} \to [0, 1]$ be an $L$-Lipschitz loss function.

Work of Canal et al. (2022, Theorem 3.1) shows that a set of $N \geq \Omega(D^2)$ known users, the per-user sample complexity for generalization to *unseen pairs* of items is $\tilde{\mathcal{O}}(D)$. Key questions remain: Does our mixture modeling lead to improved per-user sample complexity, e.g., $\tilde{\mathcal{O}}(K)$? Can we characterize PAL's generalization ability to *unseen users*, and determine the per-user sample complexity for few-shot localization of preference points (going beyond analysis in Canal et al. (2022))?

**Generalization for seen users and unseen pairs.** Without loss of generality, let us assume that each user answers $m$ preference comparisons. Let $S_i = \left\{ \left( \mathbf{x}_{j,l}^{(i)}, \mathbf{x}_{j,r}^{(i)}, y_j^{(i)} \right) \right\}_{j=1}^m$ denote an i.i.d. sample of size $m$ from user $i$, and let $\mathcal{S} = \bigcup_{i=1}^N S_i$ represent the dataset from $N$ users. Given $\mathcal{S}$, for any $(\mathbf{M}, \mathbf{Q}, \{\mathbf{w}^{(i)}\}_{i=1}^N)$, we define its empirical risk $\widehat{R}_\mathcal{S}(\mathbf{M}, \mathbf{Q}, \{\mathbf{w}^{(i)}\}_{i=1}^N)$ as

$$\frac{1}{Nm} \sum_{(i,j)} \ell \left( y_j^{(i)} \left( \mathbf{x}_{j,l}^{(i)\top} \mathbf{M} \mathbf{x}_{j,l}^{(i)} - \mathbf{x}_{j,r}^{(i)\top} \mathbf{M} \mathbf{x}_{j,r}^{(i)} + (\mathbf{x}_{j,l}^{(i)} - \mathbf{x}_{j,r}^{(i)})^\top \mathbf{Q} \mathbf{w}^{(i)} \right) \right). \tag{8}$$

Let $(\widehat{\mathbf{M}}, \widehat{\mathbf{Q}}, \{\widehat{\mathbf{w}}^{(i)}\}_{i=1}^N)$ minimize the empirical risk given $\mathcal{D}$, and $(\mathbf{M}^*, \mathbf{Q}^*, \{\mathbf{w}^{*(i)}\}_{i=1}^N)$ minimize the true risk, $\mathbb{E}_\mathcal{S}[\widehat{R}_\mathcal{S}(\mathbf{M}, \mathbf{Q}, \{\mathbf{w}^{(i)}\}_{i=1}^N)]$. Theorem 1 below provides an upper bound on the excess risk. All proofs are deferred to Appendix C; see Section C.1 for the proof of Theorem 1.

**Theorem 1.** *(Seen user generalization)* *Suppose $K < \min(N, D)$, and for each comparison, the user is asked to compare two items drawn i.i.d. from the uniform distribution on the unit sphere,* $\mathrm{Unif}(\mathbb{S}^{D-1})$. *Then, with probability at least $1 - \delta$,*

$$\mathbb{E}_\mathcal{S}[\widehat{R}_\mathcal{S}(\widehat{\mathbf{M}}, \widehat{\mathbf{Q}}, \{\widehat{\mathbf{w}}^{(i)}\}_{i=1}^N)] - \mathbb{E}_\mathcal{S}[\widehat{R}_\mathcal{S}(\mathbf{M}^*, \mathbf{Q}^*, \{\mathbf{w}^{*(i)}\}_{i=1}^N)]$$

$$\leq 12L\sqrt{\frac{\zeta_M^2 + \left( \frac{KN}{D} + K \right) \zeta_v^2}{Nm} \log(N + D)} + \sqrt{\frac{2 \log \frac{2}{\delta}}{Nm}}.$$

To interpret this result in a practical setting, let us further assume that the entries of $\mathbf{M}^*$ and $\mathbf{Q}^*$ are bounded by some constant, and set $\zeta_M = D$ and $\zeta_v = \sqrt{D}$, as done in (Canal et al., 2022). Then, the above bound becomes $\tilde{\mathcal{O}}\left( \sqrt{\frac{D^2 + KD + KN}{Nm}} \right)$, where $\tilde{\mathcal{O}}$ hides the parameters $\delta$ and $L$ and ignores logarithmic terms. Observe first that the bound decays as either the number of users $N$ or the number of samples per user $m$ increases. In addition, when $N \geq \Omega(D^2)$, the bound simplifies to $\tilde{\mathcal{O}}(\sqrt{K/m})$, which implies a per-user sample complexity of $\tilde{\mathcal{O}}(K)$. This contrasts with the existing result of $\tilde{\mathcal{O}}(D)$ without mixture modeling (Canal et al., 2022, Theorem 3.1, see also Section C.1.3 in the Appendix). The result captures the intuition that if the users amortize the cost of learning the common $\mathbf{M}$ and $\mathbf{Q}$, then each user only needs to individually learn their weights $w^{(i)} \in \Delta^{K-1}$.

**Generalization for unseen users.** So far, we have discussed the generalization ability of PAL to unseen pairs among known users, but how well does our model generalize to new users, whose data was not included in the training set at all? In this work, we provide the first answer to this question in the context of preference alignment under the ideal point model, leveraging the framework and tools for multi-task learning and learning-to-learn (Maurer et al., 2016) (Remark C.6 of the Appendix).

---

[4]The Mahalanobis distance defined by a symmetric, positive semidefinite matrix $\mathbf{M}$ is given by $d_\mathbf{M}(\mathbf{x}, \mathbf{y}) := \sqrt{(\mathbf{x} - \mathbf{y})^\top \mathbf{M}(\mathbf{x} - \mathbf{y})}$.

Suppose there is a distribution $\eta$ over a set of users $\mathcal{U}$. Given any representation $\mathbf{M}$ and prototypes $\mathbf{Q}$, along with an i.i.d. sample $S = \left\{ (\mathbf{x}_{j,l}, \mathbf{x}_{j,r}, y_j) \right\}_{j=1}^{m}$ of $m$ comparisons from some new user $u \in \mathcal{U}$, the natural approach is to few-shot learn the weights that minimize the empirical loss:

$$\tilde{\mathbf{w}}_{S;\mathbf{M},\mathbf{Q}} := \operatorname*{argmin}_{\mathbf{w} \in \Delta^{K-1}} \frac{1}{m} \sum_{j=1}^{m} \ell \left( y_j \left( \mathbf{x}_{j,l}^{\top} \mathbf{M} \mathbf{x}_{j,l} - \mathbf{x}_{j,r}^{\top} \mathbf{M} \mathbf{x}_{j,r} + (\mathbf{x}_{j,l} - \mathbf{x}_{j,r})^{\top} \mathbf{Q} \mathbf{w} \right) \right);$$

and one can evaluate the expected performance of $(\mathbf{M}, \mathbf{Q}, \tilde{\mathbf{w}}_{S;\mathbf{M},\mathbf{Q}})$ for user $u$:

$$\tilde{L}(\mathbf{M}, \mathbf{Q}; u, S) := \mathbb{E}_{(\mathbf{x}_l, \mathbf{x}_r, y)} \left[ \ell \left( y \left( \mathbf{x}_l^{\top} \mathbf{M} \mathbf{x}_l - \mathbf{x}_r^{\top} \mathbf{M} \mathbf{x}_r + (\mathbf{x}_l - \mathbf{x}_r)^{\top} \mathbf{Q} \tilde{\mathbf{w}}_{S;\mathbf{M},\mathbf{Q}} \right) \right) \right].$$

The unseen-user risk of $(\mathbf{M}, \mathbf{Q})$ is defined as: $L^{\text{unseen}}_{\text{user}}(\mathbf{M}, \mathbf{Q}) := \mathbb{E}_u \left[ L(\mathbf{M}, \mathbf{Q}; u) \right]$, where $L(\mathbf{M}, \mathbf{Q}; u) := \mathbb{E}_S \left[ \tilde{L}(\mathbf{M}, \mathbf{Q}; u, S) \right]$. Suppose that for training, $N$ users are drawn according to $\eta$, and each answers $m$ comparison queries, resulting in the dataset, $S_1 \cup \ldots \cup S_N$. Let $(\widehat{\mathbf{M}}, \widehat{\mathbf{Q}})$ be components of the minimizer of the empirical risk given by Eq. (8). Theorem 2 provides an upper bound on its excess risk when compared with

$$(\mathbf{M}^*, \mathbf{Q}^*) := \operatorname*{argmin}_{\mathbf{M}, \mathbf{Q}} \mathbb{E}_u \left[ \min_{\mathbf{w} \in \Delta^{K-1}} \mathbb{E}_{(\mathbf{x}_l, \mathbf{x}_r, y)} \left[ \ell \left( y \left( \mathbf{x}_l^{\top} \mathbf{M} \mathbf{x}_l - \mathbf{x}_r^{\top} \mathbf{M} \mathbf{x}_r + (\mathbf{x}_l - \mathbf{x}_r)^{\top} \mathbf{Q} \mathbf{w} \right) \right) \right] \right],$$

which assumes oracle knowledge of each user's preferences. See Section C.2 for its proof.

**Theorem 2.** *(**Unseen user generalization**)* *Suppose $K < \min(N, d)$, and for each comparison, the user is asked to compare two items that are drawn i.i.d. from the uniform distribution on the unit sphere, $\mathrm{Unif}(\mathbb{S}^{D-1})$. Then, with probability at least $1 - \delta$ over $S_1, \ldots, S_N$,*

$$L^{\text{unseen}}_{\text{user}}(\widehat{\mathbf{M}}, \widehat{\mathbf{Q}}) - L^{\text{unseen}}_{\text{user}}(\mathbf{M}^*, \mathbf{Q}^*) \leq 18L \sqrt{\frac{\zeta_M^2 + K^2 \zeta_v^2}{N}} + 3L \sqrt{\frac{K \zeta_v^2}{Dm}} + \sqrt{\frac{8 \log \frac{4}{\delta}}{N}}.$$

Again, let us set $\zeta_M = D$ and $\zeta_v = \sqrt{D}$. The above bound becomes $\tilde{\mathcal{O}}\left( \sqrt{\frac{D^2 + DK^2}{N}} + \sqrt{\frac{K}{m}} \right)$.

Intuitively, the first term captures how well the common mapping and the prototypes learned on seen users' dataset translate to new unseen users. This term decays as the number of seen users $N$ increases. The second term characterizes how well our few-shot preference localization for a new unseen user generalizes to unseen pairs of this user. This term indicates a sample complexity of $\tilde{\mathcal{O}}(K)$ and suggests efficient generalization, especially since $K$ can be quite small in practice.

## 4.2 NUMERICAL SIMULATION

In this section, we carefully and systematically examine how `PAL` adapts to a plurality of preferences via numerical simulations. To this end, we construct a synthetic, heterogeneous dataset similar to (Canal et al., 2022), where each item $\mathbf{x} \sim \mathcal{N}(\mathbf{0}, \frac{1}{d}I)$ and the user weight $\mathbf{W} \sim \mathcal{N}(\mathbf{0}, I)$. The true $f^\star$ is a linear mapping from $\mathbb{R}^d \to \mathbb{R}^d$. Let $K^\star$ denote the number of user prototypes and let $\mathcal{P} = \{\mathbf{p}_i\}_{i=1}^{K^\star}$ denote the set of user prototypes, where each $\mathbf{p}_i \sim \mathcal{N}(\mathbf{0}, \frac{1}{d}I)$. We assume that the distance between any pair of user prototypes is lower bounded by some value $\delta$.

**Experiment Setting.** We study a mixture setting, where each user is located in the convex hull of $\mathcal{P}$. Let $\mathbf{a}_i$ denote the $i^{\text{th}}$ user. To learn this user's ideal point, we draw $n$ pairs of items $\{\mathbf{x}_l, \mathbf{x}_r\}$ uniformly at random and assign the user's preference as $\text{sign}(\|f^*(\mathbf{x}_l) - f^*(\mathbf{a}_i)\|_2 - \|f^*(\mathbf{x}_r) - f^*(\mathbf{a}_i)\|_2)$. We generate datasets with different $K^\star$, $K$, latent dimension $d$, number of prototypes, and number of samples per user $n$. We evaluate `PAL-A` on these synthetic datasets.

**Results.** Figure 5 (a) shows that `PAL` can learn the user ideal points in the representation space. Figure 5 (b) shows that the homogeneous reward model ($K = 1$) can only achieve sub-optimal performance when diverse preferences exist. Incorporating plurality via multiple learnable prototypes with `PAL`, we gain a significant 7% accuracy boost. Figure 5 (c) shows that as we increase the number of training samples for seen users, `PAL` achieves higher test accuracy, and is also more accurate in capturing the true number of prototypes in the dataset. Figure 5 (d) presents `PAL`'s potential to generalize to new unseen users via few-shot learning to only learn their weights. We also studied a

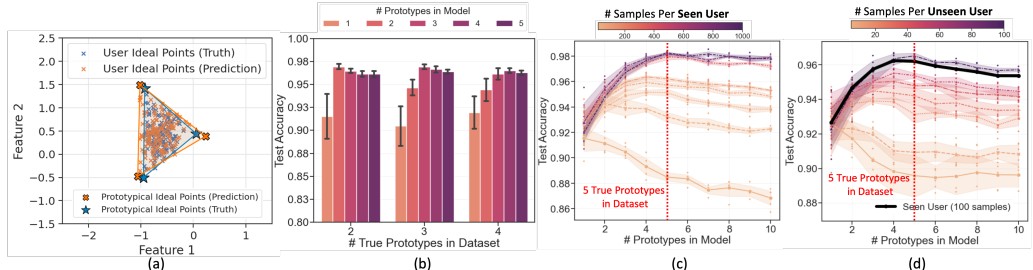

Figure 5: **(a)** shows that `PAL` can accurately capture the user prototypes and each user's ideal point. We set $d = 2$ for visualization of the user's ideal point. **(b)** illustrates the performance of `PAL` on diverse preference datasets. **(c)** empirically highlights the importance of the number of samples per seen user and the number of prototypes in `PAL`. **(d)** demonstrates the sample efficiency of the `PAL` framework on unseen users. With as few as 40 samples, we can still achieve performance comparable to that of seen users.

partition setting, where each user is drawn from $\mathcal{P}$ uniformly at random. Figure D.1 in the Appendix D.2 shows the learned ideal points under partition setting have similar trends to the mixture setting.

To further validate our findings, we also examine `PAL`'s generalization properties via Anthropic personas (Perez et al., 2022), which we restrict to Appendix D.4 for brevity.

## 5 RELATED WORK

We provide a brief summary of related works here, and a more detailed version in Appendix A. Popular foundation models (Achiam et al., 2023; Ouyang et al., 2022; Touvron et al., 2023) typically use RLHF (Azar et al., 2024; Ethayarajh et al., 2024; Rafailov et al., 2024; Stiennon et al., 2020) to align models after pretraining. These methods assume homogeneity either explicitly or implicitly by using BTL-model (Bradley & Terry, 1952). Consensus-based methods (Bakker et al., 2022) aims to find agreement among labelers for specific designed-defined goals (Bai et al., 2022b;a; Irvine et al., 2023; Ganguli et al., 2022), which prioritize the universal preference (and biases) induced by the labelers (Cheng et al., 2023; Kovač et al., 2023; Santurkar et al., 2023). Many works have highlighted that in reality, humans have diverse preferences (Nadal & Chatterjee, 2019; Sorensen et al., 2024; Wildavsky, 1987). However, approaches to suit this heterogeneity are still top-down in nature (Rame et al., 2024; Wang et al., 2024b; Wu et al., 2024), where the system designer makes learning decisions apriori, such as collecting datasets to train diverse rewards or multi-objective training. Li et al. (2024) propose personalized reward modeling with a cluster structure for users in the dataset, but model unseen users homogeneously. There is a rich literature on preference learning (Fürnkranz & Hüllermeier, 2010) and metric learning (Bellet et al., 2022). For the ideal point model, several works (Ding, 2016; Huber, 1976; Jamieson & Nowak, 2011; Massimino & Davenport, 2021; Singla et al., 2016) study sample complexity of ranking and localization when the distance is known, and some recent works (Canal et al., 2022; Wang et al., 2024c; Xu & Davenport, 2020) have studied simultaneous learning of the Mahalanobis distance which is equivalent to learning a common linear map along with unknown user preference point(s).

## 6 CONCLUSIONS

We propose `PAL`, a novel framework for modeling personalizable rewards for pluralistic alignment (Section 2) which leverages shared structures across the population while learning to personalize in a sample-efficient way. We demonstrate that `PAL` is agnostic to modality, showing strong results on both text (Section 3.1) and image (Section 3.2, 3.3) tasks. We also provide sample complexity bounds for generalization of the learned rewards to both seen and unseen users (Section 4.1). Our work aids in building much-needed foundations toward plurality for the alignment of ML/AI models. Our experiments also highlight the limitations of many real human preference datasets that are collected with rubrics that make the dataset homogeneous, and call for a more nuanced approach to data collection in the future (Section 3.2). While the mixture modeling approach of `PAL` is flexible, a limitation of using it in a static setting is that it will not generalize to new users who fall outside the convex hull of learned prototypes (Section 4.2). A more pragmatic and exciting approach would be a continually learning prototypes to adapt to new users on the fly, which we leave for future work.

## 7    REPRODUCIBILITY STATEMENT

`PAL` makes a strong case for open, democratic alignment by enabling training alignment modules with cheap encoders on simple GPUs in relatively short wall clock time (e.g. see Section 3.2, Results and Appendix E). To enable reproducibility and the practical utility of `PAL` for downstream tasks, we describe our algorithm in Appendix B, and give detailed experiment setups in Appendix D. We provide exhaustive hyper-parameter details for our experiments (Tables D.2 in Appendix D). We also attempt to provide insights over our design choices (which configurations worked and which didn't throughout the course of our experimentation) to aid others in using `PAL` for their own data and tasks. Lastly, our code and datasets are openly available at https://github.com/RamyaLab/pluralistic-alignment.

## 8    ACKNOWLEDGMENTS

We thank Zaid Harchaoui for providing feedback on the manuscript. We would also like to thank the anonymous reviewers for their valuable feedback, which helped polish this paper. This work was also partially supported by NSF grants NCS-FO 2219903 and NSF CAREER Award CCF 2238876.

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

OUTLINE OF THE APPENDICES

## A  EXTENDED RELATED WORK

**Alignment Status Quo.** Popular existing foundation models (Achiam et al., 2023; Anthropic, 2024; Ouyang et al., 2022; Touvron et al., 2023) typically use RLHF (Christiano et al., 2017; Ethayarajh et al., 2024; Stiennon et al., 2020; Wu et al., 2023a) to align models after pretraining. Recent foundation models such as Zephyr (Tunstall et al., 2023) and the Archangel suite[5] have shifted to directly optimizing on human preferences (Azar et al., 2024; Ethayarajh et al., 2024; Rafailov et al., 2024) to avoid the nuances of RL optimization (Dulac-Arnold et al., 2021). There has also been significant recent work in collecting large human preference datasets for reward model training in the text-to-image space (Kirstain et al., 2024; Wu et al., 2023b; Xu et al., 2024) (typically diffusion models (Rombach et al., 2022)).

**Reward Modeling.** These existing alignment frameworks generally assume that all humans share a single unified preference (e.g. LLM "helpfulness" or "harmlessness" (Bai et al., 2022a)) and ascribe to the Bradley-Terry (Bradley & Terry, 1952) model of pairwise preferences. Consensus-based methods (Bakker et al., 2022) aim to find agreement among labelers for specific goals like harmlessness (Bai et al., 2022b; Ganguli et al., 2022), helpfulness (Bai et al., 2022a), or engagement (Irvine et al., 2023). By design, these methods inherently prioritize the universal preference (and biases) induced by the labelers (Cheng et al., 2023; Kovač et al., 2023; Santurkar et al., 2023). In reality, humans have diverse, heterogeneous preferences (Nadal & Chatterjee, 2019; Sorensen et al., 2024; Wildavsky, 1987) that depend on individual contexts, and may even share a group structure (Bakker et al., 2022). Rewarded soups (Rame et al., 2024) make a case to capture diversity through post-hoc weight-space interpolation over a mixture of experts that learn diverse rewards. However, these rewards are learned by pre-defining what aspects are important which is done by the system designer. Separate datasets are collected to elicit human preferences on these axes as to how much people care of them. DPA (Wang et al., 2024b) models rewards as directions instead of scalars, and trains a multi-objective reward model for RLHF. Wu et al. (2024) propose fine-grained multi-objective rewards to provide more focused signal for RLHF. Recently, Li et al. (2024) propose personalized reward modeling by learning a general user embedding and treating each individual as a perturbation to the embedding. As this preference formulation is still homogeneous for users not in the dataset, they use the fixed general user embedding for generalizing to unseen users, i.e., do not personalize to new users.

Recent survey works provide excellent summaries of literature for alignment (Ji et al., 2023) and reward modeling (Wang et al., 2024a).

**Human Preference Datasets.** The preference universality assumption also extends into the data annotation/labeling processing, where labelers are given a rubric to select preferences (e.g. to rank an image pair considering image aesthetics and image-prompt alignment (Kirstain et al., 2024)). Due to this rubric, the current largest scale text-to-image generation preference datasets (Kirstain et al., 2024; Wu et al., 2023b; Xu et al., 2024) show limited diversity among labelers. In the Pick-a-Pic train set (Kirstain et al., 2024), there are only 701 disagreements among the 12487 image pairs labeled by different users (94.38% agreement), and there are zero disagreements in validation (1261 pairs) and test (1453 pairs) sets. HPS (Wu et al., 2023b) found that labeler agreement over diffusion model generations was higher for models of similar quality or size, though this diversity comes with the caveat of the labelers being provided a rubric to provide their preferences. Imagereward (Xu et al., 2024) use researcher agreement as a *criteria* to hire labelers. In the LLM domain, the popular Summarize from Feedback dataset (Stiennon et al., 2020) is also collected with rigid rubric, with labeler performance measured via agreement to the preferred answer of the authors. During the data collection period, only labelers with satisfactory agreement were retained, which led to a small number of users, all in agreement with the authors' rubric, being responsible for a majority of labeled comparisons. Status quo preference datasets used to align foundation models thus suffer from a lack of diversity due to the nature of their data collection.

**Preference learning.** There is rich literature on preference learning and ranking in various domains ranging from psychology, marketing, recommendation systems, quantifying social science surveys to crowdsourced democracy, voting theory and social choice theory. We provide a few relevant works here and direct reader to surveys such as (Fürnkranz & Hüllermeier, 2010). Ranking based models, e.g., BTL-model (Bradley & Terry, 1952; Luce, 1959), stochastic transitivity

---

[5]https://github.com/ContextualAI/HALOs

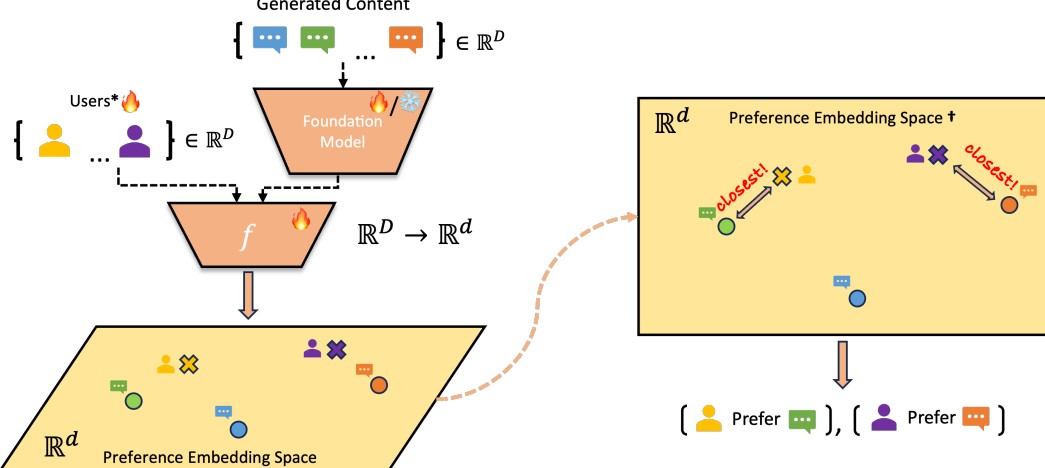

* User Ideal points are represented as learnable vectors (to learn the user's preference);
† The distance calculation method can be adapted, with options such as Euclidean distance or cosine similarity.
🔥: Learnable Module; ❄️: Frozen Module

Figure B.1: In the ideal point model considered in `PAL-A`, items and users (represented by their ideal points) are mapped to $\mathbb{R}^d$ via an unknown function $f$. A user's preference for an item (e.g., an image or a text summary) is inversely related to the distance between the item and the user's ideal point. For example, the yellow user prefers the green text summary, and the purple user prefers the orange text summary, as these items are "closer" to their respective ideal points.

models (Shah et al., 2016) focus on finding ranking of $m$ items or finding top-k items by pairwise comparisons (Hunter, 2004; Kenyon-Mathieu & Schudy, 2007; Braverman & Mossel, 2007; Negahban et al., 2012; Eriksson, 2013; Rajkumar & Agarwal, 2014; Shah & Wainwright, 2017). Ranking $m$ items in these settings requires $\mathcal{O}(m \log m)$ queries. There is also rich literature that stems from ideal point model (Coombs, 1950; Huber, 1976; Jamieson & Nowak, 2011; Ding, 2016; Singla et al., 2016; Xu & Davenport, 2020; Canal et al., 2022). Under the ideal point based models, the query complexity for ranking $m$ items reduces to $\mathcal{O}(d \log m)$, where $d$ is the dimension of the domain of representations which is usually much smaller than the number of items being ranked (Jamieson & Nowak, 2011). This is due to the fact that once the preference point is learned, it can then be used to predict rankings of new items without needing more comparisons.

**Metric learning** has been studied quite extensively and we direct the reader to surveys (Kulis, 2013) and books (Bellet et al., 2022). In particular, metric learning based on triplet querying has also been quite extesively studied (Shepard, 1962a;b; 1966; Schultz & Joachims, 2003; Kulis, 2013; Tamuz et al., 2011; Kleindessner & Luxburg, 2014; Bellet et al., 2015; Bellet & Habrard, 2015; Mason et al., 2017) which aims to learn the underlying unknown metric under the assumption that the people base their judgement for a triple query with concepts $\mathbf{x}_a, \mathbf{x}_b, \mathbf{x}_c \in \mathcal{D}$ on the relative similarities based on the distances between these concepts under the unknown metric.

**Simultaneous metric and preference learning.** More recently a few works have considered the problem of unknown metric in preference learning and proposed methods (Xu & Davenport, 2020; Canal et al., 2022; Wang et al., 2024c) and provided sample complexity analysis (Canal et al., 2022; Wang et al., 2024c) for simultaneously learning an unknown Mahalanobis metric and unknown user preference(s). Learning the unknown Mahalanobis metric can be viewed as learning linear layer on top of the embeddings from a foundation model. From our reframing of alignment, these works can be looked as `PAL-A` with linear function for $f$ and individual user preferences instead of having any structure over them.

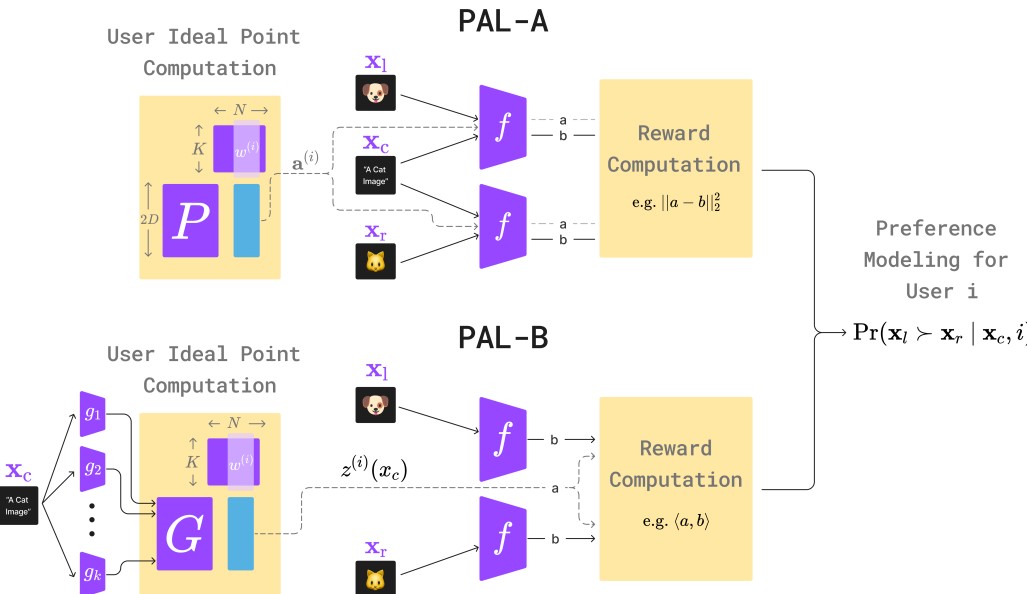

Figure B.2: Illustration of PAL framework for learning from diverse preferences (Section 2). For any user $i$, the probability of preferring $\mathbf{x}_l$ to $\mathbf{x}_r$ for the context $\mathbf{x}_c$ is computed by a reward model $r_\theta^{(i)}$ which uses a mixture modeling approach to assign a scalar reward to a sample (e.g. $\mathbf{x}_l$ or $\mathbf{x}_r$) given context ($\mathbf{x}_c$). In PAL-A, each user $i$'s preference $a^{(i)}$ is modeled as a convex combination of $K$ prototypical preferences, i.e. $a^{(i)} = Pw^{(i)}$. In PAL-B, each user $i$'s preference $z^{(i)}(\mathbf{x}_c)$ is modeled as a convex combination of $K$ prototypical functions $g_1 \cdots g_K$, i.e. $z^{(i)}(\mathbf{x}_c) =$. Reward functions formulated using the PAL framework can be used flexibly, e.g., with fixed preference points (PAL-A), with preference points that are functions of the context/prompt $\mathbf{x}_c$ (PAL-B).

## B  DETAILED MODEL OVERVIEW

### B.1  ILLUSTRATIONS AND PSEUDOCODE FOR THE PAL FRAMEWORK

Figure B.1 illustrates the ideal point model considered in PAL-A. We show the modeling mechanism of PAL (Section 2) in slightly more detail in Figure B.2. Algorithm 1 and 2 provide the pseudocode for the learning algorithms of PAL-A and PAL-B.

---

**Algorithm 1** Learning algorithm for PAL-A

**Require:** Dataset $\mathcal{D} = \bigcup_{i=1}^{N} \left\{ \left( \mathbf{x}_{j,l}^{(i)}, \mathbf{x}_{j,r}^{(i)}; \mathbf{x}_{j,c}^{(i)}, y_j^{(i)} \right) \right\}_{j=1}^{m_i}$, loss function $\ell$, model class for $f_\theta$, prototypes $\mathbf{P} = [\mathbf{p}_1, ..., \mathbf{p}_K]$ where $\mathbf{p}_k \in \mathbb{R}^D$, user weights $\mathbf{W} = [\mathbf{w}^{(1)}, ..., \mathbf{w}^{(N)}]$ where $\mathbf{w}^{(i)} \in \Delta^{K-1}$.
1: **for** each iteration **do**
2:   **Sample** a mini-batch $\left\{ \left( \mathbf{x}_{j,l}^{(i)}, \mathbf{x}_{j,r}^{(i)}; \mathbf{x}_{j,c}^{(i)}, y_j^{(i)} \right) \right\}$      $\triangleright$ random pairs, not ordered by users
3:   **User Ideal Points:** $\mathbf{a}^{(i)} = \mathbf{P} \cdot \mathbf{w}^{(i)}$
4:   **Distances**:
5:     $d_{l,j}^{(i)} = || f_\theta \left( \mathbf{x}_{l,j}^{(i)}; \mathbf{x}_{c,j}^{(i)} \right) - f_\theta(\mathbf{a}^{(i)}) ||_2^2, \quad d_{r,j}^{(i)} = || f_\theta \left( \mathbf{x}_{r,j}^{(i)}; \mathbf{x}_{c,j}^{(i)} \right) - f_\theta(\mathbf{a}^{(i)}) ||_2^2$
6:   **Loss:** $\psi_j^{(i)}(\mathbf{x}_{l,j}^{(i)}, \mathbf{x}_{r,j}^{(i)}; \mathbf{x}_{c,j}^{(i)}, y_j^{(i)}) = \ell(y_j^{(i)} \cdot (d_{l,j}^{(i)} - d_{r,j}^{(i)}))$
7:   **Update Step:** $\operatorname{argmin}_{\theta, \mathbf{P}, \{\mathbf{w}^{(i)}\}_{i=1}^{N}} \sum_{i,j} \psi_j^{(i)}(\mathbf{x}_{l,j}^{(i)}, \mathbf{x}_{r,j}^{(i)}; \mathbf{x}_{c,j}^{(i)}, y_j^{(i)})$
8: **end for**

---

---

**Algorithm 2** Learning algorithm for `PAL-B`

---

**Require:** Dataset $\mathcal{D} = \bigcup_{i=1}^{N} \left\{ \left( \mathbf{x}_{j,l}^{(i)}, \mathbf{x}_{j,r}^{(i)}; \mathbf{x}_{j,c}^{(i)}, y_j^{(i)} \right) \right\}_{j=1}^{m_i}$, loss function $\ell$, mapping function $f_\theta$,
   prototype mapping functions $\{g_{\theta_k}\}_{k=1}^{K}$, user weights $\{\mathbf{w}^{(i)} := [w_1^{(i)}, ..., w_K^{(i)}]\}_{i=1}^{N}$.

1: **for** each iteration **do**
2:     **Sample** a mini-batch $\left\{ \left( \mathbf{x}_{j,l}^{(i)}, \mathbf{x}_{j,r}^{(i)}; \mathbf{x}_{j,c}^{(i)}, y_j^{(i)} \right) \right\}$          ▷ random pairs, not ordered by users
3:     **User Ideal Point (conditioned on prompts)**:
4:         $\mathbf{a}^{(i)} = \left[ g_{\theta_i}(\mathbf{x}_{c,j}^{(i)}), ..., g_{\theta_K}(\mathbf{x}_{c,j}^{(i)}) \right]^\top \cdot \mathbf{w}^{(i)}$
5:     **Distance**:
6:         $d_{l,j}^{(i)} = \langle f_\theta \left( \mathbf{x}_{l,j}^{(i)} \right), \mathbf{a}^{(i)} \rangle, \quad d_{r,j}^{(i)} = \langle f_\theta \left( \mathbf{x}_{r,j}^{(i)} \right), \mathbf{a}^{(i)} \rangle$
7:     **Loss**: $\psi_j^{(i)}(\mathbf{x}_{l,j}^{(i)}, \mathbf{x}_{r,j}^{(i)}; \mathbf{x}_{c,j}^{(i)}, y_j^{(i)}) = \ell(y_j^{(i)} \cdot (d_{l,j}^{(i)} - d_{r,j}^{(i)}))$
8:     **Update Step:** $\mathrm{argmin}_{\theta, \mathbf{P}, \{\mathbf{w}^{(i)}\}_{i=1}^{N}} \sum \psi_j^{(i)}(\mathbf{x}_{l,j}^{(i)}, \mathbf{x}_{r,j}^{(i)}; \mathbf{x}_{c,j}^{(i)}, y_j^{(i)})$
9: **end for**

---

### B.2    MODELING CHOICES: WHEN TO USE `PAL-A` OR `PAL-B` IN PRACTICE?

We note that `PAL-B` is more natural in generative modeling settings, as it learns a personalized mapping $z^{(i)}(\mathbf{x}_c)$ for each user $i$ and any given prompt $\mathbf{x}_c$, and it learns a separate mapping for outputs $\mathbf{x}$. In contrast, `PAL-A` learns an ideal point $\mathbf{a}^{(i)}$ for each user $i$ fixed across all prompts and it learns to jointly map the prompt $\mathbf{x}_c$ and output $\mathbf{x}$ in the same space.

From experiments, we found that `PAL-B` serves as a reliable default choice (see Figure 2, Figure 3 and Table 1 on Reddit TL;DR Summary; Table 2 on Pick-a-Pic; and Figure 4 on Pick-a-Filter). We are able to get `PAL-A` to work competitively to `PAL-B` in most settings (see Table D.1, Table D.3 and Section D.4 in the Appendix); however, in practice, this may require additional engineering effort to optimize effectively.

## C   PROOFS AND ADDITIONAL THEORETICAL RESULTS

**Additional notation.**   Given two matrices $\mathbf{A}$ and $\mathbf{B}$, we use $\langle \mathbf{A}, \mathbf{B} \rangle$ to denote the Frobenius inner product between them, i.e., $\langle \mathbf{A}, \mathbf{B} \rangle = \mathrm{tr}(\mathbf{A}^\top \mathbf{B})$. As an example, recall that, given the reparameterization in Section 4.1, the difference of scores for a user can be written as $\mathbf{x}_l^\top \mathbf{M} \mathbf{x}_l - \mathbf{x}_r^\top \mathbf{M} \mathbf{x}_r + (\mathbf{x}_l - \mathbf{x}_r)^\top \mathbf{Q} \mathbf{w}$; in what follows, for better clarity, we express it in an equivalent form using the Frobenius inner product, $\langle \mathbf{x}_l \mathbf{x}_l^\top - \mathbf{x}_r \mathbf{x}_r^\top, \mathbf{M} \rangle + \langle \mathbf{x}_l - \mathbf{x}_r, \mathbf{Q} \mathbf{w} \rangle$.

### C.1   GENERALIZATION FOR SEEN USERS AND UNSEEN PAIRS

#### C.1.1   PROOF OF THEOREM 1

**Theorem 1.** *(Seen user generalization) Suppose $K < \min(N, D)$, and for each comparison, the user is asked to compare two items drawn i.i.d. from the uniform distribution on the unit sphere,* $\mathrm{Unif}(\mathbb{S}^{D-1})$. *Then, with probability at least $1 - \delta$,*

$$\mathbb{E}_{\mathcal{S}}[\widehat{R}_{\mathcal{S}}(\widehat{\mathbf{M}}, \widehat{\mathbf{Q}}, \{\widehat{\mathbf{w}}^{(i)}\}_{i=1}^N)] - \mathbb{E}_{\mathcal{S}}[\widehat{R}_{\mathcal{S}}(\mathbf{M}^*, \mathbf{Q}^*, \{\mathbf{w}^{*(i)}\}_{i=1}^N)]$$
$$\leq 12L \sqrt{\frac{\zeta_M^2 + \left(\frac{KN}{D} + K\right)\zeta_v^2}{Nm} \log(N + D)} + \sqrt{\frac{2\log\frac{2}{\delta}}{Nm}}.$$

*Proof of Theorem 1.* We use standard Rademacher complexity theory; see, e.g., (Shalev-Shwartz & Ben-David, 2014; Mohri et al., 2018) for general background and (Canal et al., 2022) for its application in preference and metric learning without mixture modeling.

Recall that $\ell : \mathbb{R} \to [0, 1]$ is an $L$-Lipschitz loss function. Let $\mathbf{W} = \left[\mathbf{w}^{(1)}, \ldots, \mathbf{w}^{(N)}\right] \in \mathbb{R}^{K \times N}$ denote the matrix containing the individual weights of all users.

Consider the following class of functions parameterized by $\mathbf{M}, \mathbf{Q}$ and $\mathbf{W}$:

$$\mathcal{H} = \left\{ h_{\mathbf{M}, \mathbf{Q}, \mathbf{W}} : (i, \mathbf{x}_l, \mathbf{x}_r, y) \mapsto \ell\left(y\left(\langle \mathbf{x}_l \mathbf{x}_l^\top - \mathbf{x}_r \mathbf{x}_r^\top, \mathbf{M} \rangle + \langle \mathbf{x}_l - \mathbf{x}_r, \mathbf{Q} \mathbf{W}_i \rangle\right)\right) \right| $$
$$\|\mathbf{M}\|_{\mathrm{F}} \leq \zeta_M, \quad \|\mathbf{Q}_k\|_2 \leq \zeta_v \ \forall k \in [K], \quad \mathbf{W}_i \in \Delta^{K-1} \ \forall i \in [N] \right\},$$

where $\mathbf{Q}_k$ denotes the $k$-th column of $\mathbf{Q}$, i.e., the $k$-th prototypical ideal point, and $\mathbf{W}_i$ denotes the $i$-th column of $\mathbf{W}$, i.e., the weights for user $i$.

To prove Theorem 1, we use Lemma C.1, which relates the excess risk to the Rademacher complexity of $\mathcal{H}$. In particular, the lemma addresses settings with multiple distributions, where data consists of i.i.d. samples from each distribution. Similar notions of task-averaged Rademacher complexity have been considered in (e.g., Ando et al., 2005; Maurer, 2006; Maurer et al., 2016).

Here, to apply Lemma C.1, we slightly abuse notation by considering $\mathcal{S}$ as a dataset where, for each $i \in [N]$ and $j \in [m]$, the comparison $(\mathbf{x}_{j,l}^{(i)}, \mathbf{x}_{j,r}^{(i)}, y_j^{(i)})$ is augmented with the user index $i$: $(i, \mathbf{x}_{j,l}^{(i)}, \mathbf{x}_{j,r}^{(i)}, y_j^{(i)})$.

Let $\boldsymbol{\sigma} = (\sigma_1, \ldots, \sigma_N) \in \{-1, +1\}^N$ be an array of independent Rademacher random variables. Then, by Lemma C.1, with probability at least $1 - \delta$,

$$\mathbb{E}_{\mathcal{S}}[\widehat{R}_{\mathcal{S}}(\widehat{\mathbf{M}}, \widehat{\mathbf{Q}}, \{\widehat{\mathbf{w}}^{(i)}\}_{i=1}^N)] - \mathbb{E}_{\mathcal{S}}[\widehat{R}_{\mathcal{S}}(\mathbf{M}^*, \mathbf{Q}^*, \{\mathbf{w}^{*(i)}\}_{i=1}^N)] \leq (*) + \sqrt{\frac{2\log\frac{2}{\delta}}{Nm}},$$

where

$$(*) = 2\mathbb{E}_{\mathcal{S}, \boldsymbol{\sigma}}\left[\sup_{\mathbf{M}, \mathbf{Q}, \mathbf{W}} \frac{1}{Nm} \sum_{i=1}^N \sum_{j=1}^m \sigma_{i,j} \ell\left(y_j^{(i)}\left(\langle \mathbf{x}_{j,l}^{(i)} \mathbf{x}_{j,l}^{(i)\top} - \mathbf{x}_{j,r}^{(i)} \mathbf{x}_{j,r}^{(i)\top}, \mathbf{M} \rangle + \langle \mathbf{x}_{j,l}^{(i)} - \mathbf{x}_{j,r}^{(i)}, \mathbf{Q} \mathbf{W}_i \rangle\right)\right)\right].$$

It suffices to show that

$$(*) \leq 12L \sqrt{\frac{\zeta_M^2 + \left(\frac{KN}{D} + K\right)\zeta_v^2}{Nm} \log(N + D)}. \tag{9}$$

To this end, observe that

$$
(*) \overset{(a)}{\leq} \frac{2L}{Nm} \mathbb{E}_{\mathcal{S},\sigma} \left[ \sup_{\mathbf{M},\mathbf{Q},\mathbf{W}} \frac{1}{Nm} \sum_{i=1}^{N} \sum_{j=1}^{m} \sigma_{i,j} \left( \left\langle \mathbf{x}_{j,l}^{(i)} \mathbf{x}_{j,l}^{(i)\top} - \mathbf{x}_{j,r}^{(i)} \mathbf{x}_{j,r}^{(i)\top}, \mathbf{M} \right\rangle + \left\langle \mathbf{x}_{j,l}^{(i)} - \mathbf{x}_{j,r}^{(i)}, \mathbf{Q}\mathbf{W}_i \right\rangle \right) \right]
$$

$$
\overset{(b)}{=} \underbrace{\frac{2L}{Nm} \mathbb{E}_{\mathcal{S},\sigma} \left[ \sup_{\mathbf{M}} \sum_{i=1}^{N} \sum_{j=1}^{m} \sigma_{i,j} \left\langle \mathbf{x}_{j,l}^{(i)} \mathbf{x}_{j,l}^{(i)\top} - \mathbf{x}_{j,r}^{(i)} \mathbf{x}_{j,r}^{(i)\top}, \mathbf{M} \right\rangle \right]}_{\text{(i)}}
$$

$$
+ \underbrace{\frac{2L}{Nm} \mathbb{E}_{\mathcal{S},\sigma} \left[ \sup_{\mathbf{Q},\mathbf{W}} \sum_{i=1}^{N} \sum_{j=1}^{m} \sigma_{i,j} \left\langle \mathbf{x}_{j,l}^{(i)} - \mathbf{x}_{j,r}^{(i)}, \mathbf{Q}\mathbf{W}_i \right\rangle \right]}_{\text{(ii)}},
$$

$$(10)$$

where (a) applies Talagrand's contraction lemma (Mohri et al., 2018, Lemma 5.7) and uses the observation that, for any $i$ and $j$, $\Pr\left(\sigma_{i,j} y_j^{(i)} = 1\right) = \Pr\left(\sigma_{i,j} = 1\right) = \frac{1}{2}$; and (b) follows because for $C = \{a + b : a \in A, b \in B\}$, $\sup C = \sup A + \sup B$.

We tackle the two terms separately. For (i), by the Cauchy-Schwarz inequality, we have

$$
\text{(i)} \leq \frac{2L}{Nm} \mathbb{E}_{\mathcal{S},\sigma} \left[ \sup_{M} \|M\|_{\mathrm{F}} \cdot \left\| \sum_{i=1}^{N} \sum_{j=1}^{m} \sigma_{i,j} \left( \mathbf{x}_{j,l}^{(i)} \mathbf{x}_{j,l}^{(i)\top} - \mathbf{x}_{j,r}^{(i)} \mathbf{x}_{j,r}^{(i)\top} \right) \right\|_{\mathrm{F}} \right]
$$

$$
\overset{(a)}{\leq} \frac{2L}{Nm} \zeta_M \cdot \sqrt{\mathbb{E}_{\mathcal{S}} \left[ \mathbb{E}_{\sigma} \left[ \left\| \sum_{l=1}^{n} \sum_{i=1}^{m} \sigma_{i,j} \left( \mathbf{x}_{j,l}^{(i)} \mathbf{x}_{j,l}^{(i)\top} - \mathbf{x}_{j,r}^{(i)} \mathbf{x}_{j,r}^{(i)\top} \right) \right\|_{\mathrm{F}}^{2} \right] \right]}
$$

$$
\overset{(b)}{\leq} \frac{4L}{Nm} \zeta_M \cdot \sqrt{Nm}
$$

$$
= 4L \sqrt{\frac{\zeta_M^2}{Nm}},
$$

where (a) uses Jensen's inequality; (b) follows from Lemma C.11 and the observation that for any realization of $\mathcal{S}$, $\left\| \mathbf{x}_{j,l}^{(i)} \mathbf{x}_{j,l}^{(i)\top} - \mathbf{x}_{j,r}^{(i)} \mathbf{x}_{j,r}^{(i)\top} \right\|_{\mathrm{F}} \leq 2$ for $i \in [N]$ and $j \in [m]$, as the items are drawn from $\mathbb{S}^{d-1}$.

We now bound term (ii). For each $i \in [N]$ and $j \in [m]$, let $\mathbf{X}_j^{(i)} := \begin{bmatrix} \mathbf{0} & \cdots & \left( \mathbf{x}_{j,l}^{(i)} - \mathbf{x}_{j,r}^{(i)} \right) & \cdots & \mathbf{0} \end{bmatrix} \in \mathbb{R}^{d \times N}$ have zeros everywhere except for the $i$-th column. Then,

$$
\text{(ii)} = \frac{2L}{Nm} \mathbb{E}_{\mathcal{S},\sigma} \left[ \sup_{\mathbf{Q},\mathbf{W}} \sum_{i=1}^{N} \sum_{j=1}^{m} \sigma_{i,j} \left\langle \mathbf{X}_j^{(i)}, \mathbf{Q}\mathbf{W} \right\rangle \right]
$$

$$
\overset{(a)}{\leq} \frac{2L}{Nm} \sup_{\mathbf{Q},\mathbf{W}} \|\mathbf{Q}\mathbf{W}\|_* \cdot \mathbb{E}_{\mathcal{S},\sigma} \left[ \left\| \sum_{i=1}^{N} \sum_{j=1}^{m} \sigma_{i,j} \mathbf{X}_j^{(i)} \right\|_2 \right]
$$

$$
\overset{(b)}{\leq} \frac{2L}{Nm} \sqrt{KN\zeta_v^2} \left( 2\sqrt{\left( \frac{Nm}{D} + \frac{Nm}{N} \right) \log(N + D)} + \log(N + D) \right)
$$

$$
\leq 4L \sqrt{\frac{\left( \frac{KN}{D} + K \right) \zeta_v^2}{Nm} \log(N + D)} + 2L \sqrt{\frac{\frac{K}{m}\zeta_v^2}{Nm}} \log(N + D),
$$

$$
\overset{(c)}{\leq} 6L \sqrt{\frac{2 \left( \frac{KN}{D} + K \right) \zeta_v^2}{Nm} \log(N + D)},
$$

where (a) uses Hölder's inequality; (b) follows from the observation that $\|\mathbf{QW}\|_* \le \sqrt{\mathrm{rank}(\mathbf{QW})} \cdot \|\mathbf{QW}\|_{\mathrm{F}} \le \sqrt{K \sum_{i=1}^{N} \|\mathbf{QW}_i\|^2}$ and Lemma C.2; (c) uses the facts that $\sqrt{a} + \sqrt{b} \le \sqrt{2(a+b)}$ and $2\log(N+D) \ge 1$ for $N, D \ge 1$, along with some algebraic simplifications.

Combining the bounds on (i) and (ii) and by Eq. (10), we have

$$(*) \le 2L \left( \sqrt{\frac{4\zeta_M^2}{Nm}} + \sqrt{\frac{18\left(\frac{KN}{D}+K\right)\zeta_v^2}{Nm} \log(N+D)} \right)$$

$$\overset{(a)}{\le} 12L \sqrt{\frac{\zeta_M^2 + \left(\frac{KN}{D}+K\right)\zeta_v^2}{Nm} \log(N+D)},$$

where (a) uses the facts that $\sqrt{a} + \sqrt{b} \le \sqrt{2(a+b)}$ and $3\log^2(N+D) \ge 1$ for $N, D \ge 1$. This completes the proof. $\square$

### C.1.2 KEY LEMMAS

We now present the key lemmas used in the proof of Theorem 1.

**Lemma C.1** (See also Ando et al. (2005); Maurer (2006); Maurer et al. (2016)). *Let $\mathcal{H} \subset \{h : [N] \times \mathcal{Z} \to [0,1]\}$, and $p_1, \ldots, p_N$ be probability measures on $\mathcal{Z}$. For each $i \in [N]$, let $S_i = \{(i, z_{i,j})\}_{j=1}^m$ where $z_{i,1}, \ldots, z_{i,m}$ are i.i.d. data points drawn according to $p_i$, and $\mathcal{S} = \bigcup_{i=1}^N S_i$. Let $\widehat{h} \in \mathrm{argmin}_{h \in \mathcal{H}} \frac{1}{Nm} \sum_{i=1}^N \sum_{j=1}^m h(i, z_{i,j})$ minimize the empirical risk, and let $h^* \in \mathrm{argmin}_{h \in \mathcal{H}} \frac{1}{N} \sum_{i=1}^N \mathbb{E}_{z_{i,1} \sim p_i} [h(i, z_{i,1})]$ minimize the true risk.*

*Then, with probability at least $1 - \delta$ over $\mathcal{S}$,*

$$\frac{1}{N} \sum_{i=1}^N \mathbb{E}_{z_{i,1} \sim p_i} \left[ \widehat{h}(i, z_{i,1}) \right] - \frac{1}{N} \sum_{i=1}^N \mathbb{E}_{z_{i,1} \sim p_1} [h^*(i, z_{i,1})] \le 2\mathcal{R}_{N,m}(\mathcal{H}) + \sqrt{\frac{2\log\frac{2}{\delta}}{Nm}}.$$

*Here,*

$$\mathcal{R}_{N,m}(\mathcal{H}) := \mathbb{E}_{\mathcal{S}, \boldsymbol{\sigma}} \left[ \sup_{h \in \mathcal{H}} \frac{1}{Nm} \sum_{i=1}^N \sum_{j=1}^m \sigma_{i,j} h(i, z_{i,j}) \right]$$

*denotes the user-averaged Rademacher complexity of $\mathcal{H}$, where $\boldsymbol{\sigma} = (\sigma_{i,j})_{i \in [N], j \in [m]}$ is an array of independent Rademacher random variables.*

*Proof.* We use standard techniques for deriving generalization bounds using Rademacher complexity (e.g., Shalev-Shwartz & Ben-David, 2014; Mohri et al., 2018). For any $h \in \mathcal{H}$, let $\widehat{\mathcal{L}}_{\mathcal{S}}(h) := \frac{1}{Nm} \sum_{i=1}^N \sum_{j=1}^m h(i, z_{i,j})$ and $\mathcal{L}(h) := \frac{1}{N} \sum_{i=1}^N \mathbb{E}_{z_{i,1} \sim p_i} [h(i, z_{i,1})]$. Notice that $\mathcal{L}(h) = \mathbb{E}_{\mathcal{S}} \left[ \widehat{\mathcal{L}}_{\mathcal{S}}(h) \right]$.

Since $\hat{g}$ minimizes $\widehat{\mathcal{L}}_{\mathcal{S}}(h)$, we have

$$\mathcal{L}(\widehat{h}) - \mathcal{L}(h^*) = \mathcal{L}(\widehat{h}) - \widehat{\mathcal{L}}_{\mathcal{S}}(\widehat{h}) + \underbrace{\widehat{\mathcal{L}}_{\mathcal{S}}(\widehat{h}) - \widehat{\mathcal{L}}_{\mathcal{S}}(h^*)}_{\le 0} + \widehat{\mathcal{L}}_{\mathcal{S}}(h^*) - \mathcal{L}(h^*)$$

$$\le \left( \mathcal{L}(\widehat{h}) - \widehat{\mathcal{L}}_{\mathcal{S}}(\widehat{h}) \right) + \left( \widehat{\mathcal{L}}_{\mathcal{S}}(h^*) - \mathcal{L}(h^*) \right).$$

We first look at the second summand. Notice that $h^*$ is independent of $\mathcal{S}$. Since $h(i, z) \in [0, 1]$ for any $h \in \mathcal{H}$, $i \in [N]$, and $z \in \mathcal{Z}$, by Hoeffding's inequality (Hoeffding, 1994), with probability $1 - \frac{\delta}{2}$,

$$\widehat{\mathcal{L}}_{\mathcal{S}}(h^*) - \mathcal{L}(h^*) \le \sqrt{\frac{\log\frac{2}{\delta}}{2Nm}}.$$

Then, by the union bound and a little algebra, it suffices to show that, with probability at least $1 - \frac{\delta}{2}$,

$$\sup_{h \in \mathcal{H}} \left( \mathcal{L}(h) - \widehat{\mathcal{L}}_{\mathcal{S}}(h) \right) \leq 2\mathcal{R}_{N,m}\left( \mathcal{H} \right) + \sqrt{\frac{\log \frac{2}{\delta}}{2Nm}}.$$

To this end, let $\kappa(\mathcal{S}) = \sup_{h \in \mathcal{H}} \left( \mathcal{L}(h) - \widehat{\mathcal{L}}_{\mathcal{S}}(h) \right)$. It is easy to verify that, since $h(i, z) \in [0, 1]$ for any $h \in \mathcal{H}$, $i \in [N]$, and $z \in \mathcal{Z}$, $\kappa(\mathcal{S})$ satisfies the bounded difference property, i.e., for any $\mathcal{S}'$ that differs from $\mathcal{S}$ by exactly one data point, $|\kappa(\mathcal{S}) - \kappa(\mathcal{S}')| \leq \frac{1}{Nm}$. We can therefore apply the bounded difference inequality (e.g., Mohri et al., 2018, Theorem D.8) on $\kappa(\mathcal{S})$. With probability at least $1 - \frac{\delta}{2}$,

$$\kappa(\mathcal{S}) \leq \mathbb{E}_{\mathcal{S}}\left[ \kappa(\mathcal{S}) \right] + \sqrt{\frac{\ln \frac{2}{\delta}}{2Nm}}.$$

It only remains to show that $\mathbb{E}_{\mathcal{S}}\left[ \kappa(\mathcal{S}) \right] \leq 2\mathcal{R}_{N,m}\left( \mathcal{H} \right)$, which follows from a standard symmetrization argument (see, e.g., Mohri et al., 2018, Chapter 3). For each $i \in [N]$, let $\mathcal{S}'_i = \left\{ \left( i, z'_{i,j} \right) \right\}_{i=1}^{m}$ where $\left\{ z'_{i,j} \right\}_{j=1}^{m}$ is another i.i.d. sample of size $m$ drawn according to $p_i$, and let $\mathcal{S}' = \bigcup_{i=1}^{N} \mathcal{S}'_i$. Then,

$$\mathbb{E}_{\mathcal{S}}\left[ \kappa(\mathcal{S}) \right] = \mathbb{E}_{\mathcal{S}} \left[ \sup_{h \in \mathcal{H}} \left( \mathbb{E}_{\mathcal{S}'} \left[ \widehat{\mathcal{L}}_{\mathcal{S}'}(h) \right] - \widehat{\mathcal{L}}_{\mathcal{S}}(h) \right) \right]$$

$$\leq \mathbb{E}_{\mathcal{S}, \mathcal{S}'} \left[ \sup_{h \in \mathcal{H}} \left( \frac{1}{Nm} \sum_{i=1}^{N} \sum_{j=1}^{m} \left( h(i, z_{i,j}) - h(i, z'_{i,j}) \right) \right) \right]$$

$$= \mathbb{E}_{\mathcal{S}, \mathcal{S}', \sigma} \left[ \sup_{h \in \mathcal{H}} \left( \frac{1}{Nm} \sum_{i=1}^{N} \sum_{j=1}^{m} \sigma_{i,j} \left( h(i, z_{i,j}) - h(i, z'_{i,j}) \right) \right) \right]$$

$$\leq \mathbb{E}_{\mathcal{S}, \sigma} \left[ \sup_{h \in \mathcal{H}} \frac{1}{Nm} \sum_{i=1}^{N} \sum_{j=1}^{m} \sigma_{i,j} h(i, z_{i,j}) \right] + \mathbb{E}_{\mathcal{S}', \sigma} \left[ \sup_{h \in \mathcal{H}} \frac{1}{Nm} \sum_{i=1}^{N} \sum_{j=1}^{m} -\sigma_{i,j} h(i, z'_{i,j}) \right]$$

$$= 2\mathcal{R}_{N,m}(\mathcal{H}). \qquad \square$$

**Lemma C.2.** *For $i \in [N]$ and $j \in [m]$, let $\mathbf{Z}_{i,j} = \sigma_{i,j} \mathbf{D}_{i,j}$, where $\sigma_{i,j}$ is an independent Rademacher ramdom variable, and $\mathbf{D}_{i,j} = [\mathbf{0} \ \ldots \ \xi_{i,j} \ \ldots \ \mathbf{0}] \in \mathbb{R}^{D \times N}$ has $\boldsymbol{\xi}_{i,j} = \mathbf{z}_{i,j} - \mathbf{z}'_{i,j} \in \mathbb{R}^{D}$ in its $i$-th column and zeros elsewhere, with $\mathbf{z}_{i,j}$ and $\mathbf{z}'_{i,j}$ drawn independently from $\mathrm{Unif}(\mathbb{S}^{D-1})$. Then,*

$$\mathbb{E} \left[ \left\| \sum_{i=1}^{N} \sum_{j=1}^{m} \mathbf{Z}_{i,j} \right\|_2 \right] \leq 2\sqrt{\left( \frac{Nm}{D} + \frac{Nm}{N} \right) \log(N + D)} + \log(N + D),$$

*where $\|\cdot\|_2$ denotes the spectral norm.*

*Proof.* For any $i \in [N]$ and $j \in [m]$, observe that $\mathbb{E}\left[ \mathbf{Z}_{i,j} \right] = 0$ and $\|\mathbf{Z}_{i,j}\|_2 = \|\boldsymbol{\xi}_{i,j}\|_2 \leq 2$. The Lemma then follows straightforwardly from the Matrix Bernstein inequality (e.g., Tropp et al., 2015, Theorem 1.6.2 , reproduced as Lemma C.10) and Lemma C.3. $\qquad \square$

**Lemma C.3.** *Under the same definitions as in Lemma C.2,*

$$\max \left\{ \left\| \sum_{i=1}^{N} \sum_{j=1}^{m} \mathbb{E}\left[ \mathbf{Z}_{i,j} \mathbf{Z}_{i,j}^{\top} \right] \right\|_2, \left\| \sum_{i=1}^{N} \sum_{j=1}^{m} \mathbb{E}\left[ \mathbf{Z}_{i,j}^{\top} \mathbf{Z}_{i,j} \right] \right\|_2 \right\} \leq 2 \left( \frac{Nm}{D} + \frac{Nm}{N} \right).$$

*Proof.* We first show that

$$\left\| \sum_{i=1}^{N} \sum_{j=1}^{m} \mathbb{E}\left[ \mathbf{Z}_{i,j} \mathbf{Z}_{i,j}^{\top} \right] \right\|_2 \leq \frac{2Nm}{D} \leq 2 \left( \frac{Nm}{D} + \frac{Nm}{N} \right).$$

This follows from the simple observation that, for any $i \in [N]$ and $j \in [m]$,

$$\mathbb{E}\left[\mathbf{Z}_{i,j}\mathbf{Z}_{i,j}^\top\right] = \mathbb{E}\left[\sigma_{i,j}^2 \cdot \boldsymbol{\xi}_{i,j}\boldsymbol{\xi}_{i,j}^\top\right] = \mathbb{E}\left[\boldsymbol{\xi}_{i,j}\boldsymbol{\xi}_{i,j}^\top\right] \overset{(a)}{=} \frac{2\mathbf{I}}{D},$$

where (a) holds because for any $\mathbf{x}, \mathbf{x}'$ drawn independently from $\mathrm{Unif}(\mathbb{S}^{D-1})$,

$$\mathbb{E}\left[(\mathbf{x} - \mathbf{x}')(\mathbf{x} - \mathbf{x}')^\top\right] = \mathbb{E}\left[\mathbf{x}\mathbf{x}^\top\right] + \mathbb{E}[\mathbf{x}'\mathbf{x}'^\top] = \frac{2\mathbf{I}}{D}.$$

It now suffices to show that

$$\left\|\sum_{i=1}^N \sum_{j=1}^m \mathbb{E}\left[\mathbf{Z}_{i,j}^\top \mathbf{Z}_{i,j}\right]\right\|_2 \leq \frac{2Nm}{N} \leq 2\left(\frac{Nm}{D} + \frac{Nm}{N}\right).$$

Observe that, for any $i \in [N]$ and $j \in [m]$, $\mathbb{E}\left[\mathbf{Z}_{i,j}^\top \mathbf{Z}_{i,j}\right]$ is a matrix in $\mathbb{R}^{N \times N}$ with zeros everywhere but in the $(i, i)$-th entry, which has value

$$\mathbb{E}\left[\boldsymbol{\sigma}_{i,j}^2 \cdot \boldsymbol{\xi}_{i,j}^\top \boldsymbol{\xi}_{i,j}\right] = \mathrm{tr}\left(\mathbb{E}\left[\boldsymbol{\xi}_{i,j}\boldsymbol{\xi}_{i,j}^\top\right]\right) = 2.$$

Therefore, we have $\sum_{i=1}^N \sum_{j=1}^m \mathbb{E}\left[\mathbf{Z}_{i,j}^\top \mathbf{Z}_{i,j}\right] = 2m\mathbf{I}$, and it follows that its largest singular value is $2m = \frac{2Nm}{N}$. $\qquad\square$

### C.1.3 Existing Result without Mixture Modeling (Canal et al., 2022)

For completeness, we adapt (Canal et al., 2022, Theorem 3.1) to our specific setting, and provide a proof. Recall that, for the class of linear transformations, $\mathcal{F}$, considered in Section 4.1, the difference of scores for a pair of items $(\mathbf{x}_l, \mathbf{x}_r)$ and a user with ideal point $\mathbf{a}$ can be written as $\langle \mathbf{x}_l\mathbf{x}_l^\top - \mathbf{x}_r\mathbf{x}_r^\top, \mathbf{A}^\top\mathbf{A}\rangle + \langle \mathbf{x}_l - \mathbf{x}_r, -2\mathbf{A}^\top\mathbf{A}\mathbf{a}\rangle$.

Let $\mathbf{M} := \mathbf{A}^\top\mathbf{A}$, $\mathbf{v}^{(i)} := -2\mathbf{M}\mathbf{a}^{(i)}$, and $\mathbf{V} := \left[\mathbf{v}^{(1)} \ \ldots \ \mathbf{v}^{(N)}\right]$. We consider $\mathbf{M} \in \left\{\mathbf{M} \in \mathbb{R}^{D \times D} : \|\mathbf{M}\|_{\mathrm{F}} \leq \zeta_M, \mathbf{M} \succeq 0\right\}$ and $\mathbf{V} \in \left\{\mathbf{V} \in \mathbb{R}^{D \times N} : \|\mathbf{V}_i\|_2 \leq \zeta_v, \forall i \in [N]\right\}$, as done in (Canal et al., 2022). Again, let $S_i := \left\{\left(\mathbf{x}_{j,l}^{(i)}, \mathbf{x}_{j,r}^{(i)}, y_j^{(i)}\right)\right\}_{j=1}^m$ denote a set of $m$ i.i.d. comparisons from user $i \in [N]$, and let $\mathcal{S} := \bigcup_{i=1}^N S_i$. Then, the empirical risk of $(\mathbf{M}, \mathbf{V})$ given $\mathcal{S}$ is:

$$\widehat{J}_{\mathcal{S}}(\mathbf{M}, \mathbf{V}) := \frac{1}{Nm}\sum_{i=1}^N \sum_{j=1}^m \ell\left(y_j^{(i)}\left\langle \mathbf{x}_{j,l}^{(i)}\mathbf{x}_{j,l}^{(i)\top} - \mathbf{x}_{j,r}^{(i)}\mathbf{x}_{j,r}^{(i)\top}, \mathbf{M}\right\rangle + \left\langle \mathbf{x}_{j,l}^{(i)} - \mathbf{x}_{j,r}^{(i)}, \mathbf{V}_i\right\rangle\right),$$

where we again assume $\ell : \mathbb{R} \to [0, 1]$ is $L$-Lipschitz. Let $(\tilde{\mathbf{M}}, \tilde{\mathbf{V}}) \in \mathrm{argmin}_{\mathbf{M}, \mathbf{V}} \widehat{J}_{\mathcal{S}}(\mathbf{M}, \mathbf{V})$, and let $(\mathbf{M}^\star, \mathbf{V}^\star) \in \mathrm{argmin}_{\mathbf{M}, \mathbf{V}} \mathbb{E}_{\mathcal{S}}\left[\widehat{J}_{\mathcal{S}}(\mathbf{M}, \mathbf{V})\right]$.

**Proposition C.4** (Canal et al. 2022, Theorem 3.1). *Suppose that for each comparison, the user is asked to compare two items drawn i.i.d. from $\mathrm{Unif}(\mathbb{S}^{D-1})$. Then, with probability at least $1 - \delta$,*

$$\mathbb{E}_{\mathcal{S}}\left[\widehat{J}_{\mathcal{S}}(\tilde{\mathbf{M}}, \tilde{\mathbf{V}})\right] - \mathbb{E}_{\mathcal{S}}\left[\widehat{J}_{\mathcal{S}}(\mathbf{M}^\star, \mathbf{V}^\star)\right] \leq 6L\sqrt{\frac{\zeta_M^2 + \zeta_v^2 N}{Nm}} + \sqrt{\frac{2\log\frac{2}{\delta}}{Nm}}$$

**Remark C.5.** *Setting $\zeta_M = d$ and $\zeta_v = \sqrt{d}$, the above excess risk bound becomes $\tilde{\mathcal{O}}\left(\sqrt{\frac{D^2 + DN}{Nm}}\right)$, where $\tilde{\mathcal{O}}$ hides the parameters $L$ and $\delta$. When $N \geq \Omega(D^2)$, the bound becomes $\tilde{\mathcal{O}}\left(\sqrt{\frac{D}{N}}\right)$, which suggests a per-user sample complexity of $\tilde{\mathcal{O}}(D)$.*

*Proof.* See (Canal et al., 2022, Section D.1) for the original proof in a slightly different setting. We include a proof here for the reader's convenience.

Consider a class of functions parameterized by $\mathbf{M}$ and $\mathbf{V}$:

$$\mathcal{G} = \left\{ g_{\mathbf{M},\mathbf{V}} : (i, \mathbf{x}_l, \mathbf{x}_r, y) \mapsto \ell\left(y\left(\left\langle \mathbf{x}_l \mathbf{x}_l^\top - \mathbf{x}_r \mathbf{x}_r^\top, \mathbf{M} \right\rangle + \langle \mathbf{x}_l - \mathbf{x}_r, \mathbf{V}_i \rangle \right)\right) \Big| \right.$$

$$\left. \|\mathbf{M}\|_{\mathrm{F}} \leq \zeta_M, \quad \|\mathbf{V}_i\|_2 \leq \zeta_v \;\; \forall i \in [N] \right\},$$

where $\mathbf{V}_i$ denotes the $i$-th column of $\mathbf{V}$, i.e., the reparameterized ideal point for user $i$.

We again apply Lemma C.1[6]. Let $\boldsymbol{\sigma} = (\sigma_1, \ldots, \sigma_N)$ be an array of independent Rademacher random variables. By Lemma C.1, we have, with probability at least $1 - \delta$,

$$\mathbb{E}_{\mathcal{S}}\left[\widehat{J}_{\mathcal{S}}(\tilde{\mathbf{M}}, \tilde{\mathbf{V}})\right] - \mathbb{E}_{\mathcal{S}}\left[\widehat{J}_{\mathcal{S}}(\mathbf{M}^\star, \mathbf{V}^\star)\right] \leq (*) + \sqrt{\frac{2\log\frac{2}{\delta}}{Nm}},$$

where

$$(*) := 2\mathbb{E}_{\mathcal{S},\boldsymbol{\sigma}}\left[\sup_{\mathbf{M},\mathbf{V}} \frac{1}{Nm} \sum_{i=1}^{N}\sum_{j=1}^{m} \sigma_{i,j}\ell\left(y_j^{(i)}\left(\left\langle \mathbf{x}_{j,l}^{(i)}\mathbf{x}_{j,l}^{(i)\top} - \mathbf{x}_{j,r}^{(i)}\mathbf{x}_{j,r}^{(i)\top}, \mathbf{M}\right\rangle + \left\langle \mathbf{x}_{j,l}^{(i)} - \mathbf{x}_{j,r}^{(i)}, \mathbf{V}_i \right\rangle\right)\right)\right].$$

It suffices to show that $(*) \leq 6L\sqrt{\frac{\zeta_M^2 + \zeta_v^2 N}{Nm}}$.

To this end, we first introduce some definitions and notations. For $i \in [N]$ and $j \in [m]$, let $\mathbf{Z}_j^{(i)} := \mathbf{x}_{j,l}^{(i)}\mathbf{x}_{j,l}^{(i)\top} - \mathbf{x}_{j,r}^{(i)}\mathbf{x}_{j,r}^{(i)\top}$, and $\mathbf{X}_j^{(i)} := \begin{bmatrix} \mathbf{0} & \ldots & \mathbf{0} & \underbrace{\mathbf{x}_{j,l}^{(i)} - \mathbf{x}_{j,r}^{(i)}}_{i\text{-th column}} & \mathbf{0} & \ldots & \mathbf{0} \end{bmatrix}$. Note that $\mathbf{Z}_j^{(i)} \in \mathbb{R}^{D \times D}$ and $\mathbf{X}_j^{(i)} \in \mathbb{R}^{D \times N}$. Then, let $\boldsymbol{\Xi}_j^{(i)} := \begin{bmatrix} \mathbf{Z}_j^{(i)} & \mathbf{X}_j^{(i)} \end{bmatrix} \in \mathbb{R}^{D \times (D+N)}$ be the concatenation of $\mathbf{Z}_j^{(i)}$ and $\mathbf{X}_j^{(i)}$. Since items are all drawn from $\mathrm{Unif}(\mathbb{S}^{D-1})$, we have

$$\left\|\boldsymbol{\Xi}_j^{(i)}\right\|_{\mathrm{F}} = \sqrt{\left\|\mathbf{x}_{j,l}^{(i)}\mathbf{x}_{j,l}^{(i)\top} - \mathbf{x}_{j,r}^{(i)}\mathbf{x}_{j,r}^{(i)\top}\right\|_{\mathrm{F}}^2 + \left\|\mathbf{x}_{j,l}^{(i)} - \mathbf{x}_{j,r}^{(i)}\right\|_2^2} \leq 2\sqrt{2}. \tag{11}$$

In addition, let $\mathbf{T} := \begin{bmatrix} \mathbf{M} & \mathbf{V} \end{bmatrix} \in \mathbb{R}^{D \times (D+N)}$ be the concatenation of $\mathbf{M}$ and $\mathbf{V}$. Observe that

$$\|\mathbf{T}\|_{\mathrm{F}} = \sqrt{\|\mathbf{M}\|_{\mathrm{F}}^2 + \sum_{i=1}^{N}\|\mathbf{V}_i\|_2^2} \leq \sqrt{\zeta_M^2 + \zeta_v^2 N}. \tag{12}$$

We are now ready to bound (*). We have

$$(*) = \frac{2}{Nm}\,\mathbb{E}_{\mathcal{S},\boldsymbol{\sigma}}\left[\sup_{\mathbf{M},\mathbf{V}} \sum_{i=1}^{N}\sum_{j=1}^{m} \sigma_{i,j}\ell\left(y_j^{(i)}\left\langle \boldsymbol{\Xi}_j^{(i)}, \mathbf{T}\right\rangle\right)\right]$$

$$\overset{(a)}{\leq} \frac{2L}{Nm}\,\mathbb{E}_{\mathcal{S},\boldsymbol{\sigma}}\left[\sup_{\mathbf{M},\mathbf{V}} \sum_{i=1}^{N}\sum_{j=1}^{m} \sigma_{i,j}\left\langle \boldsymbol{\Xi}_j^{(i)}, \mathbf{T}\right\rangle\right]$$

$$\overset{(b)}{\leq} \frac{2L}{Nm}\sup_{\mathbf{M},\mathbf{V}}\|\mathbf{T}\|_{\mathrm{F}} \cdot \sqrt{\mathbb{E}_{\mathcal{S}}\left[\mathbb{E}_{\boldsymbol{\sigma}}\left[\left\|\sum_{i=1}^{N}\sum_{j=1}^{m}\sigma_{i,j}\boldsymbol{\Xi}_j^{(i)}\right\|_{\mathrm{F}}^2\right]\right]}$$

$$\overset{(c)}{\leq} 6L\sqrt{\frac{\zeta_M^2 + \zeta_v^2 N}{Nm}},$$

where (a) uses Talagrand's contraction lemma (Mohri et al., 2018, Lemma 5.7) and the observation that $\Pr\left(\sigma_{i,j}y_j^{(i)} = 1\right) = \Pr\left(\sigma_{i,j} = 1\right) = \frac{1}{2}$ for any $i$ and $j$; (b) uses the Cauchy-Schwarz inequality and Jensen's inequality; and (c) uses Eq. (12) and Lemma C.11 along with Eq. (11). This completes the proof. □

---

[6]Again, for each $i \in [N]$ and $j \in [m]$, we augment the comparison $(\mathbf{x}_{j,l}^{(i)}, \mathbf{x}_{j,r}^{(i)}, y_j^{(i)})$ in $\mathcal{S}$ with the user index $i$: $(i, \mathbf{x}_{j,l}^{(i)}, \mathbf{x}_{j,r}^{(i)}, y_j^{(i)})$

## C.2 GENERALIZATION FOR UNSEEN USERS

**Theorem 2.** (**Unseen user generalization**) *Suppose $K < \min(N, d)$, and for each comparison, the user is asked to compare two items that are drawn i.i.d. from the uniform distribution on the unit sphere, $\mathrm{Unif}(\mathbb{S}^{D-1})$. Then, with probability at least $1 - \delta$ over $S_1, \ldots, S_N$,*

$$L^{\mathrm{unseen}}_{\mathrm{user}}(\widehat{\mathbf{M}}, \widehat{\mathbf{Q}}) - L^{\mathrm{unseen}}_{\mathrm{user}}(\mathbf{M}^*, \mathbf{Q}^*) \leq 18L\sqrt{\frac{\zeta_M^2 + K^2\zeta_v^2}{N}} + 3L\sqrt{\frac{K\zeta_v^2}{Dm}} + \sqrt{\frac{8\log\frac{4}{\delta}}{N}}.$$

**Remark C.6.** *Maurer et al. (2016) consider multi-task learning and learning-to-learn, modeling each task as a composite function $h_i \circ f$, where $f$ is a shared representation and $h_i$ is task-specific. They consider two settings: (1) multi-task learning with a given set of tasks, and (2) learning-to-learn where tasks are drawn from a distribution. Setting (1) is analogous to seen-user generalization, as studied in (Canal et al., 2022) for simulatenous metric and preference learning. To derive our generalization bound for unseen users in preference alignment (Theorem 2), we build upon the framework and employ the proof techniques from setting (2) in (Maurer et al., 2016).*

*We note that, in our mixture modeling for learning rewards, different users/tasks share not only a common representation, but also a set of prototypical ideal points. In addition, to the best of our knowledge, one cannot directly apply the results in (Maurer et al., 2016) to achieve the bound we present in Theorem 2, particularly the second term which implies $\tilde{\mathcal{O}}(K)$ sample complexity for few-shot localization of preferences.*

*Proof of Theorem 2.* We closely follow the proof strategy of (Maurer et al., 2016, Section 4.3), and tailor it to our specific learning framework and function classes to achieve tighter results (e.g., Lemma C.7 and Lemma C.9).

**Additional notations.** To facilitate our proof, we first introduce some additional notations. For any user $u \in \mathcal{U}$, let $\mathcal{P}_u$ denote the data-generating distribution for user $u$. (To be more precise, we assume that items for all users and comparison are drawn independently from a common distribution, specifically $\mathrm{Unif}(\mathbb{S}^{D-1})$, but the conditional distribution of the user preference answer $y$ given a pair of items is user-dependent.) Given any $(\mathbf{M}, \mathbf{Q}, \mathbf{w})$ and $\mathbf{x} = (\mathbf{x}_l, \mathbf{x}_r)$, we denote the difference of scores by

$$\Phi(\mathbf{x}; \mathbf{M}, \mathbf{Q}, \mathbf{w}) := \left\langle \mathbf{x}_l\mathbf{x}_l^\top - \mathbf{x}_r\mathbf{x}_r^\top, \mathbf{M} \right\rangle + \left\langle \mathbf{x}_l - \mathbf{x}_r, \mathbf{Q}\mathbf{w} \right\rangle.$$

Note that, when the context is clear, we sometimes slightly abuse notation and refer to $(\mathbf{x}_l, \mathbf{x}_r)$ as $\mathbf{x}$; for example, we may denote a sample of $m$ comparisons from some user by $\{(\mathbf{x}_j, y_j)\}_{j=1}^m$, where $\mathbf{x}_j = (\mathbf{x}_{j,l}, \mathbf{x}_{j,r})$. Given any representation $\mathbf{M}$, prototypes $\mathbf{Q}$, and a sample of $m$ comparisons from some user, $S = \{(\mathbf{x}_j, y_j)\}_{j=1}^m$, recall that $\tilde{\mathbf{w}}_{S;\mathbf{M},\mathbf{Q}} \in \mathrm{argmin}_{\mathbf{w}} \frac{1}{m}\sum_{j=1}^m \ell(y_j \cdot \Phi(\mathbf{x}_j; \mathbf{M}, \mathbf{Q}, \mathbf{w}))$; we also let

$$\rho(S; \mathbf{M}, \mathbf{Q}) := \min_{\mathbf{w} \in \Delta^{K-1}} \frac{1}{m}\sum_{j=1}^m \ell(y_j \cdot \Phi(\mathbf{x}_j; \mathbf{M}, \mathbf{Q}, \mathbf{w})).$$

We now prove Theorem 2. Given $S_1, \ldots, S_N$, consider the following decomposition of the excess risk:

$$L^{\mathrm{unseen}}_{\mathrm{user}}(\widehat{\mathbf{M}}, \widehat{\mathbf{Q}}) - L^{\mathrm{unseen}}_{\mathrm{user}}(\mathbf{M}^*, \mathbf{Q}^*)$$

$$\leq \underbrace{\left( L^{\mathrm{unseen}}_{\mathrm{user}}(\widehat{\mathbf{M}}, \widehat{\mathbf{Q}}) - \frac{1}{N}\sum_{i=1}^N \rho(S_i; \widehat{\mathbf{M}}, \widehat{\mathbf{Q}}) \right)}_{(i)} + \underbrace{\left( \frac{1}{N}\sum_{i=1}^N \rho(S_i; \widehat{\mathbf{M}}, \widehat{\mathbf{Q}}) - \frac{1}{N}\sum_{i=1}^N \rho(S_i; \mathbf{M}^*, \mathbf{Q}^*) \right)}_{(ii), \leq 0}$$

$$+ \underbrace{\left( \frac{1}{N}\sum_{i=1}^N \rho(S_i; \mathbf{M}^*, \mathbf{Q}^*) - \mathbb{E}_u\left[ \mathbb{E}_{S\sim\mathcal{P}_u^m}[\rho(S; \mathbf{M}^*, \mathbf{Q}^*)] \right] \right)}_{(iii)}$$

$$+ \underbrace{\mathbb{E}_u\left[ \mathbb{E}_{S\sim\mathcal{P}_u^m}[\rho(S; \mathbf{M}^*, \mathbf{Q}^*)] \right] - L^{\mathrm{unseen}}_{\mathrm{user}}(\mathbf{M}^*, \mathbf{Q}^*)}_{(iv), \leq 0}.$$

$$(13)$$

Let us look at the four terms separately:

(i) Observe that, for any $\mathbf{M}, \mathbf{Q}$,

$$L^{\text{unseen}}_{\text{user}}(\mathbf{M}, \mathbf{Q}) - \frac{1}{N} \sum_{i=1}^{N} \rho(S_i; \mathbf{M}, \mathbf{Q})$$

$$= \underbrace{\mathbb{E}_u \left[ \mathbb{E}_{S \sim \mathcal{P}_u^m} \left[ \mathbb{E}_{(\mathbf{x}_l, \mathbf{x}_r, y) \sim \mathcal{P}_u} \left[ \ell \left( y \cdot \Phi(\mathbf{x}; \mathbf{M}, \mathbf{Q}, \tilde{\mathbf{w}}_{S; \mathbf{M}, \mathbf{Q}}) \right) \right] - \rho(S; \mathbf{M}, \mathbf{Q}) \right] \right]}_{(1)}$$

$$+ \underbrace{\left( \mathbb{E}_u \left[ \mathbb{E}_{S \sim \mathcal{P}_u^m} \left[ \rho(S; \mathbf{M}, \mathbf{Q}) \right] \right] - \frac{1}{N} \sum_{i=1}^{N} \rho(S_i; \mathbf{M}, \mathbf{Q}) \right)}_{(2)}.$$

Lemma C.7 and Lemma C.9 establish uniform bounds on (1) and (2), respectively. It follows that, with probability at least $1 - \frac{\delta}{2}$,

$$L^{\text{unseen}}_{\text{user}}(\widehat{\mathbf{M}}, \widehat{\mathbf{Q}}) - \frac{1}{N} \sum_{i=1}^{N} \rho(S_i; \widehat{\mathbf{M}}, \widehat{\mathbf{Q}}) \leq 18L \sqrt{\frac{\zeta_M^2 + K^2 \zeta_v^2}{N}} + 3L \sqrt{\frac{K \zeta_v^2}{Dm}} + 3 \sqrt{\frac{\log \frac{4}{\delta}}{2N}}.$$

$$(14)$$

(ii) This term is non-positive by the definition of $(\widehat{\mathbf{M}}, \widehat{\mathbf{Q}})$, which are components of the minimizer of the empirical risk for the seen users.

(iii) Since $(\mathbf{M}^*, \mathbf{Q}^*)$ does not depend on $S_1, \ldots, S_N$ and $\rho(S_i; \mathbf{M}, \mathbf{Q}) \in [0, 1]$ for any $i \in [N]$ and $\mathbf{M}, \mathbf{Q}$, we can apply Hoeffding's inequality (Hoeffding, 1994). We have, with probability at least $1 - \frac{\delta}{2}$,

$$\frac{1}{N} \sum_{i=1}^{N} \rho(S_i; \mathbf{M}^*, \mathbf{Q}^*) - \mathbb{E}_u \left[ \mathbb{E}_{S \sim \mathcal{P}_u^m} \left[ \rho(S; \mathbf{M}^*, \mathbf{Q}^*) \right] \right] \leq \sqrt{\frac{\log \frac{2}{\delta}}{2N}}. \qquad (15)$$

(iv) This term is non-positive:

For any $u \in \mathcal{U}$, let $\mathbf{w}_u^* \in \operatorname{argmin}_{\mathbf{w} \in \Delta^{K-1}} \mathbb{E}_{(\mathbf{x}_l, \mathbf{x}_r, y) \sim \mathcal{P}_u} \left[ \ell \left( y \cdot \Phi(\mathbf{x}; \mathbf{M}^*, \mathbf{Q}^*, \mathbf{w}) \right) \right]$. Then,

$$\mathbb{E}_u \left[ \mathbb{E}_{S \sim \mathcal{P}_u^m} \left[ \rho(S; \mathbf{M}^*, \mathbf{Q}^*) \right] \right]$$

$$= \mathbb{E}_u \left[ \mathbb{E}_{S \sim \mathcal{P}_u^m} \left[ \min_{\mathbf{w}} \frac{1}{m} \sum_{j=1}^{m} \ell \left( y_j \cdot \Phi(\mathbf{x}_j; \mathbf{M}^*, \mathbf{Q}^*, \mathbf{w}) \right) \right] \right]$$

$$\leq \mathbb{E}_u \left[ \mathbb{E}_{S \sim \mathcal{P}_u^m} \left[ \frac{1}{m} \sum_{j=1}^{m} \ell \left( y_j \cdot \Phi(\mathbf{x}_j; \mathbf{M}^*, \mathbf{Q}^*, \mathbf{w}_u^*) \right) \right] \right]$$

$$= \mathbb{E}_u \left[ \mathbb{E}_{(\mathbf{x}_l, \mathbf{x}_r, y) \sim \mathcal{P}_u} \left[ \ell \left( y \cdot \Phi(x; \mathbf{M}^*, \mathbf{Q}^*, \mathbf{w}_u^*), y) \right) \right] \right] =: L^{\text{unseen}}_{\text{user}}(\mathbf{M}^*, \mathbf{Q}^*).$$

The proof is completed by combining Eq. (13), Eq. (14), and Eq. (15) and applying the union bound. $\qquad \square$

**Lemma C.7.** *With probability at least $1 - \frac{\delta}{2}$,*

$$\sup_{\mathbf{M}, \mathbf{Q}} \mathbb{E}_u \left[ \mathbb{E}_{S \sim \mathcal{P}_u^m} \left[ \rho(S; \mathbf{M}, \mathbf{Q}) \right] \right] - \frac{1}{N} \sum_{i=1}^{N} \rho(S_i; \mathbf{M}, \mathbf{Q}) \leq 18L \sqrt{\frac{\zeta_M^2 + K^2 \zeta_v^2}{N}} + 3 \sqrt{\frac{\log \frac{4}{\delta}}{2N}}.$$

*Proof.* To avoid cluttering the notation, we denote $\mathbb{E}_{u \sim \eta} \left[ \mathbb{E}_{S \sim \mathcal{P}_u^m}[\cdot] \right]$ by $\mathbb{E}_{u, S}[\cdot]$. Let $\boldsymbol{\epsilon} = (\epsilon_1, \ldots, \epsilon_N)$ be an array of independent standard normal random variables. Then, by (Mohri et al.,

2018, Theorem 3.3) and (Wainwright, 2019, Excercise 5.5), with probability at least $1 - \frac{\delta}{2}$,

$$\sup_{\mathbf{M},\mathbf{Q}} \mathbb{E}_{u,S}\left[\rho(S;\mathbf{M},\mathbf{Q})\right] - \frac{1}{N}\sum_{i=1}^{N} \rho(S_i;\mathbf{M},\mathbf{Q})$$

$$\leq \sqrt{2\pi} \cdot \mathbb{E}_{\boldsymbol{\epsilon}}\left[\sup_{\mathbf{M},\mathbf{Q}} \frac{1}{N}\sum_{i=1}^{N} \epsilon_i \cdot \rho(S_i;\mathbf{M},\mathbf{Q})\right] + 3\sqrt{\frac{\log\frac{4}{\delta}}{2N}}, \tag{16}$$

where $\mathbb{E}_{\boldsymbol{\epsilon}}\left[\sup_{\mathbf{M},\mathbf{Q}} \frac{1}{N}\sum_{i=1}^{N} \epsilon_i \cdot \rho(S_i;\mathbf{M},\mathbf{Q})\right]$ is the empirical Gaussian complexity of $\{S \mapsto \rho(S;\mathbf{M},\mathbf{Q}) : \mathbf{M},\mathbf{Q}\}$ given $S_1,\ldots,S_N$, which we bound in the following.

Let $\boldsymbol{\gamma} \in \mathbb{R}^{N \times m}$ and $\boldsymbol{\sigma} \in \mathbb{R}^{N \times m \times K}$ be arrays of independent standard normal random variables, where $\boldsymbol{\epsilon}, \boldsymbol{\gamma}$, and $\boldsymbol{\sigma}$ are mutually independent. We have

$$\frac{\sqrt{m}N}{\sqrt{2}L} \cdot \mathbb{E}_{\boldsymbol{\epsilon}}\left[\sup_{\mathbf{M},\mathbf{Q}} \frac{1}{N}\sum_{i=1}^{N} \epsilon_i \cdot \rho(S_i;\mathbf{M},\mathbf{Q})\right]$$

$$\overset{(a)}{\leq} \mathbb{E}_{\boldsymbol{\gamma},\boldsymbol{\sigma}}\left[\sup_{\mathbf{M},\mathbf{Q}}\left(\sum_{i=1}^{N}\sum_{j=1}^{m}\gamma_{ij}\left\langle \mathbf{x}_{j,l}^{(i)}\mathbf{x}_{j,l}^{(i)\top} - \mathbf{x}_{j,r}^{(i)}\mathbf{x}_{j,r}^{(i)\top}, \mathbf{M}\right\rangle + \sum_{i=1}^{N}\sum_{j=1}^{m}\sum_{k=1}^{K}\sigma_{ijk}\left(\mathbf{Q}^\top\left(\mathbf{x}_{j,l}^{(i)} - \mathbf{x}_{j,r}^{(i)}\right)\right)_k\right)\right]$$

$$= \mathbb{E}_{\boldsymbol{\gamma}}\left[\sup_{\mathbf{M}} \sum_{i=1}^{N}\sum_{j=1}^{m}\gamma_{ij}\left\langle \mathbf{x}_{j,l}^{(i)}\mathbf{x}_{j,l}^{(i)\top} - \mathbf{x}_{j,r}^{(i)}\mathbf{x}_{j,r}^{(i)\top}, \mathbf{M}\right\rangle\right] + \mathbb{E}_{\boldsymbol{\sigma}}\left[\sup_{\mathbf{Q}} \sum_{i=1}^{N}\sum_{j=1}^{m}\sum_{k=1}^{K}\sigma_{ijk}\left(\mathbf{Q}^\top\left(\mathbf{x}_{j,l}^{(i)} - \mathbf{x}_{j,r}^{(i)}\right)\right)_k\right]$$

$$\overset{(b)}{\leq} 2\sqrt{\zeta_M^2 Nm} + \mathbb{E}_{\boldsymbol{\sigma}}\left[\sup_{\mathbf{Q}} \sum_{i=1}^{N}\sum_{j=1}^{m}\sum_{k=1}^{K}\sigma_{ijk}\left(\mathbf{Q}^\top\left(\mathbf{x}_{j,l}^{(i)} - \mathbf{x}_{j,r}^{(i)}\right)\right)_k\right]$$

$$\overset{(c)}{\leq} 2\sqrt{\zeta_M^2 \cdot Nm} + 2\sqrt{K\zeta_v^2 \cdot NmK}$$

$$\overset{(d)}{\leq} 2\sqrt{2\left(\zeta_M^2 + K^2\zeta_v^2\right)mN}, \tag{17}$$

where

(a) follows from Lemma C.8;

(b) uses the Cauchy-Schwarz inequality, Jensen's inequality, and Lemma C.11:

$$\mathbb{E}_{\boldsymbol{\gamma}}\left[\sup_{\mathbf{M}} \sum_{i=1}^{N}\sum_{j=1}^{m}\gamma_{ij}\left\langle \mathbf{x}_{j,l}^{(i)}\mathbf{x}_{j,l}^{(i)\top} - \mathbf{x}_{j,r}^{(i)}\mathbf{x}_{j,r}^{(i)\top}, \mathbf{M}\right\rangle\right]$$

$$\leq \sup_{\mathbf{M}} \|\mathbf{M}\|_{\mathrm{F}} \cdot \sqrt{\mathbb{E}\left[\left\|\sum_{i=1}^{N}\sum_{j=1}^{m}\gamma_{ij}\left(\mathbf{x}_{j,l}^{(i)}\mathbf{x}_{j,l}^{(i)\top} - \mathbf{x}_{j,r}^{(i)}\mathbf{x}_{j,r}^{(i)\top}\right)\right\|_{\mathrm{F}}^2\right]}$$

$$\leq 2\zeta_M\sqrt{Nm};$$

(c) again uses the Cauchy-Schwarz inequality and Jensen's inequality:

$$\mathbb{E}_{\boldsymbol{\sigma}}\left[\sup_{\mathbf{Q}}\sum_{i=1}^{N}\sum_{j=1}^{m}\sum_{k=1}^{K}\sigma_{ijk}\left(\mathbf{Q}^{\top}\left(\mathbf{x}_{j,l}^{(i)}-\mathbf{x}_{j,r}^{(i)}\right)\right)_{k}\right]$$

$$=\mathbb{E}_{\boldsymbol{\sigma}}\left[\sup_{\mathbf{Q}}\sum_{k=1}^{K}\left\langle\mathbf{Q}_{k},\sum_{i=1}^{N}\sum_{j=1}^{m}\sigma_{ijk}\left(\mathbf{x}_{j,l}^{(i)}-\mathbf{x}_{j,r}^{(i)}\right)\right\rangle\right]$$

$$\le\mathbb{E}_{\boldsymbol{\sigma}}\left[\sup_{\mathbf{Q}}\left(\sum_{k=1}^{K}\|\mathbf{Q}_k\|^2\right)^{\frac{1}{2}}\left(\sum_{k=1}^{K}\left\|\sum_{i=1}^{N}\sum_{j=1}^{m}\sigma_{ijk}\left(\mathbf{x}_{j,l}^{(i)}-\mathbf{x}_{j,r}^{(i)}\right)\right\|_2^2\right)^{\frac{1}{2}}\right]$$

$$\le\sqrt{K\zeta_v^2}\cdot\sqrt{\sum_{k=1}^{K}\mathbb{E}_{\boldsymbol{\sigma}}\left[\left\|\sum_{i=1}^{N}\sum_{j=1}^{m}\sigma_{ijk}\left(\mathbf{x}_{j,l}^{(i)}-\mathbf{x}_{j,r}^{(i)}\right)\right\|_2^2\right]},$$

and for any $k \in [K]$, by the independence of the elements of $\boldsymbol{\sigma}$,

$$\mathbb{E}_{\boldsymbol{\sigma}}\left[\left\|\sum_{i=1}^{N}\sum_{j=1}^{m}\sigma_{ijk}\left(\mathbf{x}_{j,l}^{(i)}-\mathbf{x}_{j,r}^{(i)}\right)\right\|_2^2\right]=\sum_{i=1}^{N}\sum_{j=1}^{m}\mathbb{E}\left[\sigma_{ijk}^2\right]\cdot\left(\mathbf{x}_{j,l}^{(i)}-\mathbf{x}_{j,r}^{(i)}\right)^{\top}\left(\mathbf{x}_{j,l}^{(i)}-\mathbf{x}_{j,r}^{(i)}\right)\le 4Nm;$$

(d) follows from the simple fact that $\sqrt{a}+\sqrt{b}\le\sqrt{2(a+b)}$.

Combining Eq. (16) with Eq. (17) completes the proof. □

We now present Lemma C.8 used in the proof of Lemma C.7.

**Lemma C.8.** *Given* $S_1,\dots,S_N$,

$$\mathbb{E}_{\boldsymbol{\epsilon}}\left[\sup_{\mathbf{M},\mathbf{Q}}\frac{1}{N}\sum_{i=1}^{N}\epsilon_i\cdot\rho(S_i;\mathbf{M},\mathbf{Q})\right]$$

$$\le\frac{\sqrt{2}L}{\sqrt{m}N}\cdot\mathbb{E}_{\boldsymbol{\gamma},\boldsymbol{\sigma}}\left[\sup_{\mathbf{M},\mathbf{Q}}\left(\sum_{i=1}^{N}\sum_{j=1}^{m}\gamma_{ij}\left\langle\mathbf{x}_{j,l}^{(i)}\mathbf{x}_{j,l}^{(i)\top}-\mathbf{x}_{j,r}^{(i)}\mathbf{x}_{j,r}^{(i)\top},\mathbf{M}\right\rangle+\sum_{i=1}^{N}\sum_{j=1}^{m}\sum_{k=1}^{K}\sigma_{ijk}\left(\mathbf{Q}^{\top}\left(\mathbf{x}_{j,l}^{(i)}-\mathbf{x}_{j,r}^{(i)}\right)\right)_k\right)\right].$$

*Proof.* We apply the technique used in (Maurer et al., 2016), which utilizes the Sudakov-Fernique inequality (e.g., Adler, 1990, reproduced as Lemma C.12). To this end, we define two Gaussian processes, indexed by $(\mathbf{M},\mathbf{Q})$:

$$Y_{\mathbf{M},\mathbf{Q}}=\sum_{i=1}^{N}\epsilon_i\cdot\rho(S_i;\mathbf{M},\mathbf{Q}),$$

where $\boldsymbol{\epsilon}\in\mathbb{R}^N$ is an array of independent standard normal random variables, and

$$W_{\mathbf{M},\mathbf{Q}}=\frac{\sqrt{2}L}{\sqrt{m}}\left(\sum_{i=1}^{N}\sum_{j=1}^{m}\gamma_{ij}\left\langle\mathbf{x}_{j,l}^{(i)}\mathbf{x}_{j,l}^{(i)\top}-\mathbf{x}_{j,r}^{(i)}\mathbf{x}_{j,r}^{(i)\top},\mathbf{M}\right\rangle+\sum_{i=1}^{N}\sum_{j=1}^{m}\sum_{k=1}^{K}\sigma_{ijk}\left(\mathbf{Q}^{\top}\left(\mathbf{x}_{j,l}^{(i)}-\mathbf{x}_{j,r}^{(i)}\right)\right)_k\right),$$

where $\boldsymbol{\gamma}\in\mathbb{R}^{N\times m}$ and $\boldsymbol{\sigma}\in\mathbb{R}^{N\times m\times K}$ are mutually independent arrays of independent standard normal random variables. By the Sudakov-Fernique inequality (reproduced as Lemma C.12), it suffices to show that, for any $(\mathbf{M},\mathbf{Q})$ and $(\mathbf{M}',\mathbf{Q}')$,

$$\mathbb{E}\left[(Y_{\mathbf{M},\mathbf{Q}}-Y_{\mathbf{M}',\mathbf{Q}'})^2\right]\le\mathbb{E}\left[(W_{\mathbf{M},\mathbf{Q}}-W_{\mathbf{M}',\mathbf{Q}'})^2\right].$$

Then, $\mathbb{E}\left[\sup_{\mathbf{M},\mathbf{Q}}Y_{\mathbf{M},\mathbf{Q}}\right]\le\mathbb{E}\left[\sup_{\mathbf{M},\mathbf{Q}}W_{\mathbf{M},\mathbf{Q}}\right]$, which completes the proof.

Recall that $\rho(S_i; \mathbf{M}, \mathbf{Q}) := \min_{\mathbf{w} \in \Delta^{K-1}} \frac{1}{m} \sum_{j=1}^{m} \ell\left(y_j^{(i)} \cdot \Phi(\mathbf{x}_j^{(i)}; \mathbf{M}, \mathbf{Q}, \mathbf{w})\right)$. Since $\epsilon_i$'s are independent, we have

$$\mathbb{E}\left[(Y_{\mathbf{M},\mathbf{Q}} - Y_{\mathbf{M}',\mathbf{Q}'})^2\right]$$

$$= \sum_{i=1}^{N} \left(\min_{\mathbf{w}} \frac{1}{m} \sum_{j=1}^{m} \ell\left(y_j^{(i)} \cdot \Phi(\mathbf{x}_j^{(i)}; \mathbf{M}, \mathbf{Q}, \mathbf{w})\right) - \min_{\mathbf{w}} \frac{1}{m} \sum_{j=1}^{m} \ell\left(y_j^{(i)} \cdot \Phi(\mathbf{x}_j^{(i)}; \mathbf{M}', \mathbf{Q}', \mathbf{w})\right)\right)^2.$$

Now, for any $i \in [N]$,

$$\left(\min_{\mathbf{w}} \frac{1}{m} \sum_{j=1}^{m} \ell\left(y_j^{(i)} \cdot \Phi(\mathbf{x}_j^{(i)}; \mathbf{M}, \mathbf{Q}, \mathbf{w})\right) - \min_{\mathbf{w}} \frac{1}{m} \sum_{j=1}^{m} \ell\left(y_j^{(i)} \cdot \Phi(\mathbf{x}_j^{(i)}; \mathbf{M}', \mathbf{Q}', \mathbf{w})\right)\right)^2$$

$$\overset{(a)}{\leq} \sup_{\mathbf{w}} \left(\frac{1}{m} \sum_{j=1}^{m} \left(\ell\left(y_j^{(i)} \cdot \Phi(\mathbf{x}_j^{(i)}; \mathbf{M}, \mathbf{Q}, \mathbf{w})\right) - \ell\left(y_j^{(i)} \cdot \Phi(\mathbf{x}_j^{(i)}; \mathbf{M}', \mathbf{Q}', \mathbf{w})\right)\right)\right)^2$$

$$\overset{(b)}{\leq} \frac{1}{m} \sup_{\mathbf{w}} \sum_{j=1}^{m} \left(\ell\left(y_j^{(i)} \cdot \Phi(\mathbf{x}_j^{(i)}; \mathbf{M}, \mathbf{Q}, \mathbf{w})\right) - \ell\left(y_j^{(i)} \cdot \Phi(\mathbf{x}_j^{(i)}; \mathbf{M}', \mathbf{Q}', \mathbf{w})\right)\right)^2$$

$$\overset{(c)}{\leq} \frac{L^2}{m} \sup_{\mathbf{w}} \sum_{j=1}^{m} \left(\Phi(\mathbf{x}_j^{(i)}; \mathbf{M}, \mathbf{Q}, \mathbf{w}) - \Phi(\mathbf{x}_j^{(i)}; \mathbf{M}', \mathbf{Q}', \mathbf{w})\right)^2$$

$$= \frac{L^2}{m} \sup_{\mathbf{w}} \sum_{j=1}^{m} \left(\left\langle \mathbf{x}_{j,l}^{(i)} \mathbf{x}_{j,l}^{(i)\top} - \mathbf{x}_{j,r}^{(i)} \mathbf{x}_{j,r}^{(i)\top}, \mathbf{M} - \mathbf{M}' \right\rangle + \left\langle (\mathbf{Q} - \mathbf{Q}')^\top \left(\mathbf{x}_{j,l}^{(i)} - \mathbf{x}_{j,r}^{(i)}\right), \mathbf{w} \right\rangle\right)^2$$

$$\overset{(d)}{\leq} \frac{2L^2}{m} \sum_{j=1}^{m} \left(\left\langle \mathbf{x}_{j,l}^{(i)} \mathbf{x}_{j,l}^{(i)\top} - \mathbf{x}_{j,r}^{(i)} \mathbf{x}_{j,r}^{(i)\top}, \mathbf{M} - \mathbf{M}' \right\rangle\right)^2 + \frac{2L^2}{m} \sup_{\mathbf{w}} \sum_{j=1}^{m} \left(\left\langle (\mathbf{Q} - \mathbf{Q}')^\top \left(\mathbf{x}_{j,l}^{(i)} - \mathbf{x}_{j,r}^{(i)}\right), \mathbf{w} \right\rangle\right)^2$$

$$\overset{(e)}{\leq} \frac{2L^2}{m} \sum_{j=1}^{m} \left(\left\langle \mathbf{x}_{j,l}^{(i)} \mathbf{x}_{j,l}^{(i)\top} - \mathbf{x}_{j,r}^{(i)} \mathbf{x}_{j,r}^{(i)\top}, \mathbf{M} - \mathbf{M}' \right\rangle\right)^2 + \frac{2L^2}{m} \sum_{j=1}^{m} \sum_{k=1}^{K} \left(\left((\mathbf{Q} - \mathbf{Q}')^\top \left(\mathbf{x}_{j,l}^{(i)} - \mathbf{x}_{j,r}^{(i)}\right)\right)_k\right)^2.$$

$$(18)$$

where

(a) follows directly from Lemma C.13;

(b) uses the AM-QM inequality;

(c) follows because $\ell$ is $L$-Lipschitz, and $(yz - yz')^2 = (z - z')^2$ for $y \in \{\pm 1\}$ and $z, z' \in \mathbb{R}$;

(d) uses the simple fact that $(a + b)^2 \leq 2(a^2 + b^2)$; and

(e) uses the Cauchy-Schwarz inequality and the fact that for any $\mathbf{w} \in \Delta^{K-1}$, $\|\mathbf{w}\|_2 \leq 1$.

We now turn our attention to $(W_{\mathbf{M},\mathbf{Q}})$. Since the elements of $\boldsymbol{\gamma}$ and $\boldsymbol{\sigma}$ are independent, it follows that

$$\mathbb{E}\left[(W_{\mathbf{M},\mathbf{Q}} - W_{\mathbf{M}',\mathbf{Q}'})^2\right]$$

$$= \frac{2L^2}{m} \mathbb{E}\left[\left(\sum_{i=1}^{N} \sum_{j=1}^{m} \gamma_{ij} \left\langle \mathbf{x}_{j,l}^{(i)} \mathbf{x}_{j,l}^{(i)\top} - \mathbf{x}_{j,r}^{(i)} \mathbf{x}_{j,r}^{(i)\top}, \mathbf{M} - \mathbf{M}' \right\rangle + \sum_{i=1}^{N} \sum_{j=1}^{m} \sum_{k=1}^{K} \sigma_{ijk} \left((\mathbf{Q} - \mathbf{Q}')^\top \left(\mathbf{x}_{j,l}^{(i)} - \mathbf{x}_{j,r}^{(i)}\right)\right)_k\right)^2\right]$$

$$= \frac{2L^2}{m} \sum_{j=1}^{m} \left(\left\langle \mathbf{x}_{j,l}^{(i)} \mathbf{x}_{j,l}^{(i)\top} - \mathbf{x}_{j,r}^{(i)} \mathbf{x}_{j,r}^{(i)\top}, \mathbf{M} - \mathbf{M}' \right\rangle\right)^2 + \frac{2L^2}{m} \sum_{j=1}^{m} \sum_{k=1}^{K} \left(\left((\mathbf{Q} - \mathbf{Q}')^\top \left(\mathbf{x}_{j,l}^{(i)} - \mathbf{x}_{j,r}^{(i)}\right)\right)_k\right)^2$$

$$\geq \mathbb{E}\left[(Y_{\mathbf{M},\mathbf{Q}} - Y_{\mathbf{M}',\mathbf{Q}'})^2\right],$$

where the inequality follows from Eq. (18). This completes the proof. $\qquad\square$

**Lemma C.9.** *Suppose for any user, when drawing a sample, $m$ pairs of items are independently drawn from* $\mathrm{Unif}(\mathbb{S}^{D-1})$. *Then,*

$$\sup_{\mathbf{M},\mathbf{Q}} \mathbb{E}_u \left[ \mathbb{E}_{S \sim \mathcal{P}_u^m} \left[ \mathbb{E}_{(\mathbf{x}_l,\mathbf{x}_r,y) \sim \mathcal{P}_u} \left[ \ell \left( y \cdot \Phi(\mathbf{x}; \mathbf{M}, \mathbf{Q}, \tilde{\mathbf{w}}_{S;\mathbf{M},\mathbf{Q}}) \right) \right] - \rho(S; \mathbf{M}, \mathbf{Q}) \right] \right] \le 3L \sqrt{\frac{K\zeta_v^2}{Dm}}.$$

*Proof.* Let $\boldsymbol{\sigma} = (\sigma_i, \ldots, \sigma_m)$ be an array of independent Rademacher random variables. We have

$$\sup_{\mathbf{M},\mathbf{Q}} \mathbb{E}_u \left[ \mathbb{E}_S \left[ \mathbb{E}_{(x,y)} \left[ \ell \left( y \cdot \Phi(x; \mathbf{M}, \mathbf{Q}, \tilde{\mathbf{w}}_{S;\mathbf{M},\mathbf{Q}}) \right) \right] - \frac{1}{m} \sum_{j=1}^m \ell \left( \Phi(y_j \cdot \mathbf{x}_j; \mathbf{M}, \mathbf{Q}, \tilde{\mathbf{w}}_{S;\mathbf{M},\mathbf{Q}}) \right) \right] \right]$$

$$\le \sup_{\mathbf{M},\mathbf{Q}} \mathbb{E}_u \left[ \mathbb{E}_S \left[ \sup_{\mathbf{w}} \left( \mathbb{E}_{(x,y)} \left[ \ell \left( y \cdot \Phi(x; \mathbf{M}, \mathbf{Q}, \mathbf{w}) \right) \right] - \frac{1}{m} \sum_{i=1}^m \ell \left( y_j \cdot \Phi(\mathbf{x}_j; \mathbf{M}, \mathbf{Q}, \mathbf{w}) \right) \right) \right] \right]$$

$$\stackrel{(a)}{\le} 2 \cdot \sup_{\mathbf{M},\mathbf{Q}} \mathbb{E}_u \left[ \mathbb{E}_{S,\boldsymbol{\sigma}} \left[ \sup_{\mathbf{w}} \frac{1}{m} \sum_{j=1}^m \sigma_j \cdot \ell \left( y_j \cdot \Phi(\mathbf{x}_j; \mathbf{M}, \mathbf{Q}, \mathbf{w}) \right) \right] \right]$$

$$\stackrel{(b)}{\le} 2L \cdot \sup_{\mathbf{M},\mathbf{Q}} \mathbb{E}_{u,S,\boldsymbol{\sigma}} \left[ \sup_{\mathbf{w}} \frac{1}{m} \sum_{j=1}^m \sigma_j \cdot \left( \langle \mathbf{x}_{j,l}\mathbf{x}_{j,l}^\top - \mathbf{x}_{j,r}\mathbf{x}_{j,r}^\top, \mathbf{M} \rangle + \langle \mathbf{x}_{j,l} - \mathbf{x}_{j,r}, \mathbf{Q}\mathbf{w} \rangle \right) \right]$$

$$= 2L \cdot \sup_{\mathbf{M},\mathbf{Q}} \mathbb{E}_{u,S,\boldsymbol{\sigma}} \left[ \sup_{\mathbf{w}} \frac{1}{m} \sum_{j=1}^m \sigma_j \cdot \langle \mathbf{x}_{j,l} - \mathbf{x}_{j,r}, \mathbf{Q}\mathbf{w} \rangle \right] + \underbrace{\mathbb{E}_{u,S,\boldsymbol{\sigma}} \left[ \frac{1}{m} \sum_{j=1}^m \sigma_j \cdot \langle \mathbf{x}_{j,l}\mathbf{x}_{j,l}^\top - \mathbf{x}_{j,r}\mathbf{x}_{j,r}^\top, \mathbf{M} \rangle \right]}_{=0}$$

$$= \frac{2L}{m} \cdot \sup_{\mathbf{Q}} \mathbb{E}_{u,S,\boldsymbol{\sigma}} \left[ \sup_{\mathbf{w}} \left\langle \sum_{j=1}^m \sigma_i \mathbf{Q}^\top (\mathbf{x}_{j,l} - \mathbf{x}_{j,r}), \mathbf{w} \right\rangle \right]$$

$$\stackrel{(c)}{\le} \frac{2L}{m} \cdot \sup_{\mathbf{Q}} \sqrt{ \mathbb{E}_{u,S,\boldsymbol{\sigma}} \left[ \left\| \sum_{j=1}^n \sigma_i \mathbf{Q}^\top (\mathbf{x}_{j,l} - \mathbf{x}_{j,r}) \right\|_2^2 \right] }$$

$$\stackrel{(d)}{\le} \frac{2L}{m} \cdot \sup_{\mathbf{Q}} \sqrt{ \sum_{j=1}^m \mathbb{E}_{\boldsymbol{\sigma}}[\sigma_j^2] \cdot \mathrm{tr} \left( \mathbf{Q}\mathbf{Q}^\top \mathbb{E}_{u,S} \left[ (\mathbf{x}_{j,l} - \mathbf{x}_{j,r})(\mathbf{x}_{j,l} - \mathbf{x}_{j,r})^\top \right] \right) }$$

$$\stackrel{(e)}{=} \frac{2L}{m} \cdot \sup_{\mathbf{Q}} \sqrt{ \frac{2}{D} \sum_{j=1}^m \mathrm{tr} \left( \mathbf{Q}\mathbf{Q}^\top \right) }$$

$$\stackrel{(f)}{\le} 3L \sqrt{\frac{K\zeta_v^2}{Dm}},$$

where

(a) introduces the Rademacher random variables and uses the standard symmetrization technique (e.g., Shalev-Shwartz & Ben-David, 2014, Lemma 26.2);

(b) uses Talagrand's contraction lemma (e.g., Mohri et al., 2018, Lemma 5.7) along with the fact that the loss function $\ell$ is $L$-Lipschitz;

(c) uses the Cauchy-Schwarz inequality, Jensen's inequality and the fact that any vector $\mathbf{w} \in \Delta^{K-1}$ satisfies $\|\mathbf{w}\|_2 \le 1$.

(d) follows because $\sigma_i$'s are independent Rademacher random variables and uses the linearity of expectation and the trace operator;

(e) uses the observation that for two independent vectors $\mathbf{z}, \mathbf{z}'$ from $\mathrm{Unif}(\mathbb{S}^{D-1})$, $\mathbb{E}\left[(\mathbf{z} - \mathbf{z}')(\mathbf{z} - \mathbf{z}')^\top\right] = \frac{2}{D}\mathbf{I}_D$; and

(f) uses the observation that $\mathrm{tr}(\mathbf{Q}\mathbf{Q}^\top) = \|\mathbf{Q}\|_\mathrm{F}^2 \le K\zeta_v^2$. $\hfill\square$

## C.3 Auxiliary Lemmas

**Lemma C.10** (Matrix Bernstein, Tropp et al., 2015)**.** *Let* $\mathbf{X}_1, \ldots, \mathbf{X}_m \in \mathbb{R}^{d_1 \times d_2}$ *be independent random matrices such that* $\mathbb{E}\left[\mathbf{X}_i\right] = 0$ *and* $\|\mathbf{X}_i\| \le R$ *for each* $i \in [m]$*. Let*

$$\iota = \max\left\{\left\|\sum_{i=1}^m \mathbb{E}\left[\mathbf{X}_i\mathbf{X}_i^\top\right]\right\|, \left\|\sum_{i=1}^m \mathbb{E}\left[\mathbf{X}_i^\top\mathbf{X}_i\right]\right\|\right\}.$$

*Then,*

$$\mathbb{E}\left[\left\|\sum_{i=1}^m \mathbf{X}_i\right\|\right] \le \sqrt{2\iota \log(d_1 + d_2)} + \frac{1}{3}R\log(d_1 + d_2).$$

**Lemma C.11.** *Let* $\mathbf{Z} = (\mathbf{Z}_1, \ldots, \mathbf{Z}_n)$ *be an array of matrices such that* $\|\mathbf{Z}_i\|_\mathrm{F} \le B_i$*, and* $\boldsymbol{\sigma} = (\sigma_1, \ldots, \sigma_n)$ *be an array of independent random variables such that* $\mathbb{E}\left[\sigma_i\right] = 0$ *and* $\mathbb{E}\left[\sigma_i^2\right] = 1$ *for all* $i \in [n]$*. Then,*

$$\mathbb{E}_\sigma\left[\left\|\sum_{i=1}^n \sigma_i\mathbf{Z}_i\right\|_\mathrm{F}^2\right] \le \sum_{i=1}^n B_i^2$$

*Proof.* We have

$$\begin{aligned}
\mathbb{E}_{\boldsymbol{\sigma}}\left[\left\|\sum_{i=1}^n \sigma_i\mathbf{Z}_i\right\|_\mathrm{F}^2\right] &= \mathbb{E}_{\boldsymbol{\sigma}}\left[\mathrm{tr}\left(\left(\sum_{i=1}^n \sigma_i\mathbf{Z}_i\right)^\top\left(\sum_{i=1}^n \sigma_i\mathbf{Z}_i\right)\right)\right] \\
&= \mathbb{E}_{\boldsymbol{\sigma}}\left[\mathrm{tr}\left(\sum_{i\ne j}\sigma_i\sigma_j\mathbf{Z}_i^\top\mathbf{Z}_j + \sum_i \sigma_i^2\mathbf{Z}_i^\top\mathbf{Z}_i\right)\right] \\
&= \mathrm{tr}\left(\sum_{i\ne j}\underbrace{\mathbb{E}_{\boldsymbol{\sigma}}\left[\sigma_i\sigma_j\right]}_{=0}\mathbf{Z}_i^\top\mathbf{Z}_j + \sum_i\underbrace{\mathbb{E}_{\boldsymbol{\sigma}}\left[\sigma_i^2\right]}_{=1}\mathbf{Z}_i^\top\mathbf{Z}_i\right) \\
&= \sum_{i=1}^n \mathrm{tr}\left(\mathbf{Z}_i^\top\mathbf{Z}_i\right) \\
&\le \sum_{i=1}^n B_i^2. \hfill\square
\end{aligned}$$

**Lemma C.12** (Sudakov-Fernique inequality, e.g., Adler, 1990, Theorem 2.9)**.** *Let* $X$ *and* $Y$ *be two centered, almost surely bounded Gaussian processes indexed by* $t \in T$ *such that, for all* $t, s \in T$,

$$\mathbb{E}\left[(X_t - X_s)^2\right] \le \mathbb{E}\left[(Y_t - Y_s)^2\right].$$

*Then,*

$$\mathbb{E}\left[\sup_{t\in T} X_t\right] \le \mathbb{E}\left[\sup_{t\in T} Y_t\right].$$

**Lemma C.13.** *Let* $f$ *and* $g$ *be functions from* $\mathcal{X}$ *to* $[0, 1]$ *that are parameterized by* $w \in \mathcal{W}$*. Given* $x_1, \ldots, x_n \in \mathcal{X}$*, we have*

$$\left(\min_w \frac{1}{n}\sum_{i=1}^n f(x_i; w) - \min_w \frac{1}{n}\sum_{i=1}^n g(x_i; w)\right) \le \sup_w\left(\frac{1}{n}\sum_{i=1}^n f(x_i; w) - g(x_i; w)\right)^2.$$

*Proof.* Let us consider two cases:

1. $\min_w \frac{1}{n} \sum_{i=1}^n f(x_i; w) - \min_w \frac{1}{n} \sum_{i=1}^n g(x_i; w) \geq 0$:

   Let $w_g^* \in \operatorname{argmin} \frac{1}{n} \sum_{i=1}^n g(x_i; w)$. It follows that

   $$0 \leq \min_w \frac{1}{n} \sum_{i=1}^n f(x_i; w) - \frac{1}{n} \min_w \sum_{i=1}^n g(x_i; w) \leq \frac{1}{n} \sum_{i=1}^n \left( f(x_i; w_g^*) - g(x_i; w_g^*) \right).$$

   Therefore,

   $$\left( \min_w \frac{1}{n} \sum_{i=1}^n f(x_i; w) - \frac{1}{n} \min_w \sum_{i=1}^n g(x_i; w) \right)^2 \leq \left( \frac{1}{n} \sum_{i=1}^n \left( f(x_i; w_g^*) - g(x_i; w_g^*) \right) \right)^2$$

   $$\leq \sup_w \left( \frac{1}{n} \sum_{i=1}^n \left( f(x_i; w) - g(x_i; w) \right) \right)^2.$$

2. $\min_w \frac{1}{n} \sum_{i=1}^n f(x_i; w) - \min_w \frac{1}{n} \sum_{i=1}^n g(x_i; w) < 0$:

   Similarly, let $w_f^* \in \operatorname{argmin} \frac{1}{n} \sum_{i=1}^n f(x_i; w)$. Then,

   $$\frac{1}{n} \sum_{i=1}^n \left( f(x_i; w_f^*) - g(x_i; w_f^*) \right) < \min_w \frac{1}{n} \sum_{i=1}^n f(x_i; w) - \frac{1}{n} \min_w \sum_{i=1}^n g(x_i; w) < 0.$$

   It follows that

   $$\left( \min_w \frac{1}{n} \sum_{i=1}^n f(x_i; w) - \frac{1}{n} \min_w \sum_{i=1}^n g(x_i; w) \right)^2 \leq \left( \frac{1}{n} \sum_{i=1}^n \left( f(x_i; w_f^*) - g(x_i; w_f^*) \right) \right)^2$$

   $$\leq \sup_w \left( \frac{1}{n} \sum_{i=1}^n \left( f(x_i; w) - g(x_i; w) \right) \right)^2. \quad \square$$

# D    EXPERIMENT DETAILS

## D.1    GENERAL PROCEDURE

We initialize our models with random weights for all of our experiments, including the prototypical weight for each user. We apply the softmax function to each prototypical weight to ensure that it is a probability vector. For the unseen user, we initialize its prototypical weight randomly. However, when we update the model using gradient descent, we fix the learned projectors and the prototypical points and only update the prototypical weight of the unseen users. During the training of the `Large` variant of `PAL`, we sample # batchsize samples from the preference datasets and concurrently update the shared foundation model, the mapping functions, and the corresponding user-specific weights for each sample. For the `Tiny` variant of `PAL`, we fix the foundation model and keep updating other components.

## D.2    NUMERICAL SIMULATION

**Experiment Setup.** We introduce the dataset simulation procedure in Section 4.2. We use the following hyperparameters to generate the synthetic dataset: $d = 16, K = 3, N = 100, n = 100, \delta = 1$. We generate another 50 comparison pairs per user as the held-out dataset. Note that here we don't follow the prompt-guided item generation, i.e. conditioning $x_c$ generates $x_l$ and $x_r$. Instead, we directly draw the item $\{x_l, x_r\}$ from a normal distribution for simplicity. In the experimental setup, we apply a toy version of `PAL`-A, where the distance between the synthetic item and the user's ideal point is measured by $\|f(x) - f(u)\|_2$. We use a projection matrix (i.e. one-layer MLP network without bias term and activation function) as the model architecture. We randomly initialize the learnable parameters of prototypical user groups and user weights and use the Adam optimizer. The projector $f$ has learning rate $5e - 4$ and weight decay $1e - 3$. The learning rate of the learnable parameters of prototypical user groups and user weights is $5e - 3$. With the aim of good convergence, we train for 1000 epochs per run. We run multiple trials to explore the influence of each hyperparameter: 1) varying the number of samples of seen users $n = \{20, 40, 60, 80, 100, 400, 800, 1000\}, d = \{2, 16\}, K = 5, N = 250$, 2) varying the number of samples of new users $n_{new} = \{5, 10, 20, 30, 40, 50, 100\}, d = \{2, 16\}, K = 5, n = 50$, 3) varying the number of groups $K = \{2, 3, 4, 5, 6\}, d = \{2, 16\}, n = 50, N = 50 * K$. We plot the results of this experiment in Figure 5 and discuss implications in Section 4.2.

We consider two variants of modeling each user's ideal point through the lens of a shared group structure of preferences via prototypical ideal points (henceforth referred to as "prototypes"):

**S1.** Mixture Model: a user ideal point is a convex combination of $K$ prototypes, i.e. lies in the convex hull of all prototypes.

**S2.** Partition Model: a user ideal point is one of $K$ prototypes.

To visualize how well `PAL` can adjust to the *true* number of user groups present in data, via learnable prototypical points to represent each group, we consider a simple setting with $d = 2, K^* = 3$, $K = \{1, 2, 3\}$ and $N = 100$ and plot the results in Figure D.1 for both partition and mixture settings. We also plot items in the partition setting in Figure D.2.

**Partition Model**    : With only a single allowed assignment for a learnable prototype (Figure D.1a., $K = 1$), the predicted prototype is approximately the centroid of the true prototypes, i.e. the model tries to predict a good group assignment on average. Also note that since we have a single prototype, all predicted user ideal points lie on the prototype itself and performance is close to random. As we increase the degrees of freedom for learnable prototypes to two (Figure D.1b., $K = 2$), the model can predict one prototype close to a true prototype (in red), while the other predicted prototype is approximately an average of the blue and green true prototypes. User ideal points now lie in the convex hull of these two predicted prototypes, i.e. the line joining these points. It is only when we increase $K = K^*$, i.e. we match the "true" number of groups in the data (Figure D.1c., $K = 3$), the model can correctly predict close to all three true prototypes, and user ideal points are concentrated around the predicted prototypes. These observations extend to Figure D.2, where we additionally plot normally distributed items. Recall that in our modeling design, the distance between the user

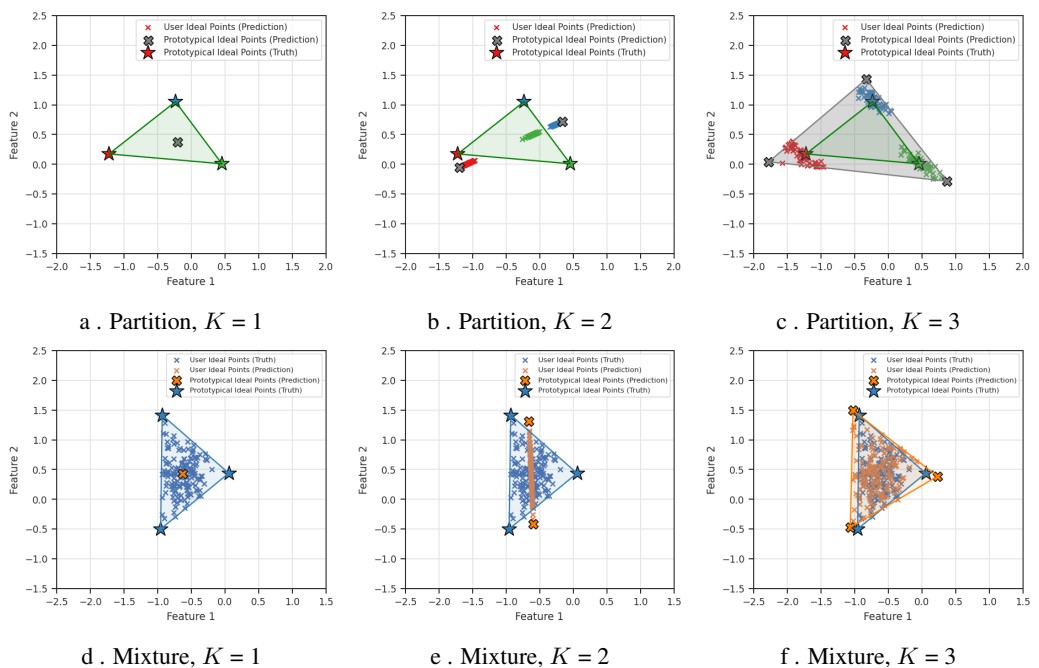

Figure D.1: In synthetic dataset experiments (Section 4.2), we model user ideal points in two distinct ways: the partition model and the mixture model. To visualize how PAL performs in these settings, we set $d = 2$, $K^* = 3$, $K = \{1, 2, 3\}$ and $N = 100$ where each user ideal point and prototype is represented as a point in a two-dimensional space. In both scenarios, as the number of prototypes or user groups in our model ($K$) approaches the true number in the synthetic dataset ($K^*$), the PAL framework effectively learns both the prototypes and the heterogeneous user ideal points.

ideal point and the item reflects the user's preference; hence the closer the predicted user ideal point is to the true ideal point, the higher the performance.

**Mixture Model** : The results for the mixture model are similar to those of the partition model. With a single allowed assignment for a learnable prototype (Figure D.1(d.), $K = 1$), the predicted prototype is approximately the centroid of the true prototypes. As we increase the degrees of freedom to two (Figure D.1(e.), $K = 2$), predicted prototypes are close to two true prototypes, but one is neglected. When we increase $K = K^*$ (Figure D.1(f.), $K = 3$), matching the true number of groups in the data, the mixture model successfully predicts prototypes that lie close to all three true prototypes. This demonstrates that similar to the partition model, the mixture model can also adjust well to the true number of user groups present in the data.

### D.3   REDDIT TL;DR SUMMARY (TEXT-TO-TEXT)

**Detials of seen dataset.** We train PAL reward models on a variant of the Reddit TL;DR Summary dataset from Li et al. (2024). In this variant, only the ten workers who gave the most feedback were chosen (Each user contains at least 1,000 samples). These ten workers are then divided into a majority and minority group, where the majority prefers the longer response, and the minority prefers the shorter response. More details about how the dataset is generated can be found in Section 6.1 of Li et al. (2024). This processed dataset contains 20,969 training samples, 2,330 validation samples, and 4,921 test samples. Each example consists of one user ID, one prompt, two responses, and the user's preference.

**Details of unseen dataset.** We selected all workers, excluding the ten used in the seen dataset, as candidates for the unseen dataset. From this pool, we filtered users with at least 100 valid comparison pairs (i.e., no missing values), resulting in a total of 31 users. We randomly assigned 70% of these users to prefer longer summaries, while the remaining users were designated as preferring shorter

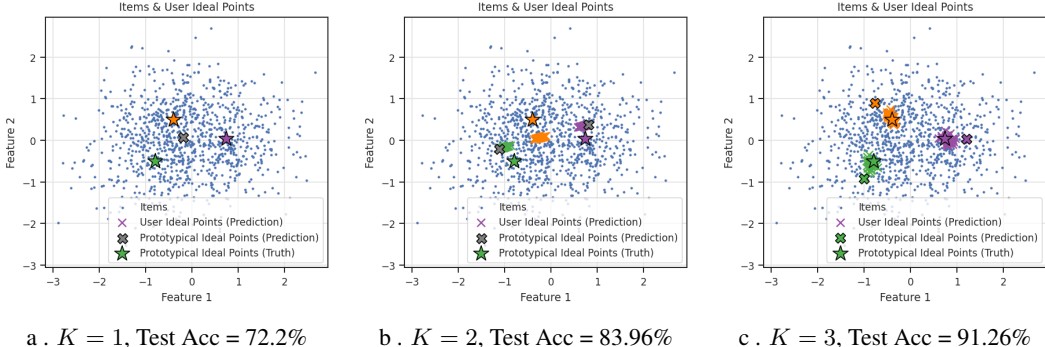

a . $K = 1$, Test Acc = 72.2%          b . $K = 2$, Test Acc = 83.96%          c . $K = 3$, Test Acc = 91.26%

Figure D.2: Here we plot all items, the predicted user ideal points, and predicted and true prototypes in two-dimensional feature space. The items are normally distributed with $d = 2, K^* = 3, N = 100, n = 100$. As seen in the figure, when we set the number of prototypes in the model $K$ equal to the true number of user groups $K^* = 3$, `PAL` can accurately capture the group structure and predict each user's ideal preference point, as well as the prototypes that represent each group ($K = 3$).

Table D.1: Seen user test accuracy of `PAL` vs. P-DPO (Li et al. v2) and vanilla DPO on the Reddit TL;DR Summary dataset. We run 5 trials with $K = 2$ and $K = 1$. Our method consistently performs better than the baseline methods. For the `Large` version of `PAL`, we use `OPT-350M` as the foundation model.

| Model | Params | Seen User Test Accuracy |
|---|---|---|
| `PAL-A-Tiny` ($K = 2$) | 1.6M | $52.91 \pm 0.55$ |
| `PAL-B-Tiny` ($K = 2$) | 2.4M | $79.54 \pm 0.54$ |
| `PAL-A-Large` ($K = 2$) | 352M | $90.00 \pm 1.89$ |
| **`PAL-B-Large` (K = 2)** | 352M | $\mathbf{92.82 \pm 0.95}$ |
| P-DPO Individual | 6.7B | 91.04 |
| P-DPO Cluster ($K = 5$) | 6.7B | 91.12 |
| `PAL-A-Tiny` ($K = 1$) | 1.6M | $49.99 \pm 0.17$ |
| `PAL-B-Tiny` ($K = 1$) | 1.6M | $51.51 \pm 0.08$ |
| **`PAL-A-Large` ($K = 1$)** | 352M | $\mathbf{61.28 \pm 2.25}$ |
| `PAL-B-Large` ($K = 1$) | 352M | $59.96 \pm 3.45$ |
| Vanilla DPO | 6.7B | 58.91 |

summaries. Based on these assignments, the users' preferences were re-labeled. Given the varying numbers of few-shot samples in the training set, we partitioned each user's comparison pairs into training, validation, and test sets, resulting in multiple datasets.

We leverage multiple pretrained LLMs, including `OPT-350M`, `DistillBERT`, `Bge-m3`, and `gte-Qwen2`,[7] as the base model, combined with two-layer MLPs utilizing GELU activation. In the `Tiny` variant of `PAL`, we fix the pretrained foundation model and only train the two-layer MLP, whereas in the `Large` variant, we also train the foundation model. In our `PAL` reward model, we set $K = 2$ and apply different learning rates for various model components: 9.65e-6 for pretrained LLMs (`Large`), 1e-4 for the two-layer MLPs, and 5e-3 for user weights. The higher learning rate of user weights can enhance the exploration of each user's weight across user groups. As with typical reward models, we train for only 1 epoch to avoid overfitting. The hyperparameter configurations are detailed in Table D.2. The training process takes roughly 1 hour on 1×RTX4090 GPU.

The loss design follows the typical loss of the Reward Model, we use the cumulative loss which weights the per-token reward loss,

---

[7]Li et al. (2024) use `GPT-J 6.7B`. However, the model card for that model on Hugging Face is broken.

Table D.2: The training hyperparameter setting of `PAL` reward modeling on Reddit TL;DR. The corresponding experiment setup is described in Section 3.1.

| Hyperparameters | Values |
|---|---|
| K | 2 |
| Batch size | 4 |
| Projectors | mlp-2layer-gelu-dropout0 |
| Epoch | 1 |
| Learning rate of LLM | 9.65e-6 |
| Learning rate of projectors | 1e-4 |
| Learning rate of user weights | 5e-3 |
| Weight decay of LLM | 0.0 |
| Weight decay of projectors | 0.01 |
| Weight decay of user weights | 0.0 |
| Loss weighting | cumulative |
| Dimension of preference embedding | 512 |
| End of conversation token | `<|endoftext|>` |
| Maximum sequence length | 600 |

$$L_{RM}(x, y_w, y_l; \theta) = \frac{\sum_{i=1}^{L} i \cdot \log \sigma \left( r(x, y_l^{(i)}) - r(x, y_w^{(i)}) \right)}{(L+1)L/2}$$

where $x$ is the prompt, $y^{(i)}$ represents the LLM backbone prediction at generation timestep $i$, $y_w$ and $y_l$ separately represent the winning response and the losing response. Note that in our implementation of `PAL-A`, we concatenate the prompt and the item on the token level. Therefore, the embedding for an item produced by a foundation model already contains the information of the prompt. This implementation allows us to use the embedding directly without needing to concatenate it with the embedding of the prompt. Table D.1 reports the performance of our models and the numbers reported in Li et al. (2024). We run our model 5 times and report the mean and standard deviation. We want to note that even though we did not conduct any hyperparameter tuning, With heterogeneous modeling ($K > 1$), `PAL-B-Large` achieves approximately **+1.8%** higher prediction accuracy compared to the state-of-the-art heterogeneous P-DPO (Li et al. (2024)) with 5 clusters. With homogeneous modeling ($K = 1$), `PAL-A-Large` is able to outperform vanilla DPO by **+2.4%**.

### D.3.1 FEW-SHOT GENERALIZATION TO UNSEEN USERS

The procedure for few-shot generalization to unseen users is as follows: We randomly initialize the user weights, as done during seen user training, and then learn the user weights while keeping the LLM components and MLP projectors fixed. Since only the user weights need to be learned, the sample efficiency is significantly higher compared to seen user training. Results indicate that with just 20 samples per new user, we can achieve performance comparable to that of seen user generalization (Figure 3). In Table D.3, we compare the performance of `PAL-A-Tiny`, `PAL-A-Large`, `PAL-B-Tiny`, and `PAL-B-Large` trained on `OPT-350M` embeddings when $K = 2$ and $N = 10, 20, 50, 100$. Our results show that `PAL-A-Large`, `PAL-B-Tiny`, and `PAL-B-Large` outperform the baselines substantially. We note that while P-DPO was state-of-the-art on seen users, it's performance drops off dramatically for unseen users (-36.6%), while **`PAL-B-Tiny` (91.63%) exceeds P-DPO's state-of-the-art seen accuracy (91.12%) with only 10 samples on unseen users**! This indicates the promising potential of `PAL` for cheap few-shot adaptation to new, unseen users in a sample-efficient manner.

We observe that increasing sample complexity $N$ from 10 to 100 is impactful only for `PAL-B-Tiny` (+3%), while the other configurations gain only +0.4 to +1%. `PAL-B` vastly outperforms `PAL-A` for the same size and sample complexity (up to +27%). Lastly, `Large` models outperform their `Tiny` counterparts across sample complexities from +14.2 to +23.5%.

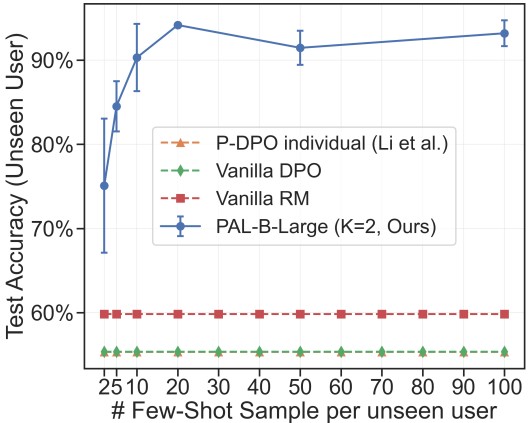

Figure D.3: We evaluate few-shot adaptation capabilities via unseen accuracy on the Reddit TL;DR dataset comparing to state-of-the-art Li et al. (2024) and simple vanilla DPO (Rafailov et al., 2024).

Table D.3: Unseen user generalization of PAL compared to baselines on the Reddit TL;DR Summary dataset. Here $N$ refers to the number of samples used to learn the weights for the unseen users, and seen accuracies are those reported in Table D.1. We note that with just 10 samples for each new (unseen) user, PAL-B-Large exceeds state-of-the-art performance of P-DPO with K=5 (91.12%), demonstrating the suitability of PAL for few-shot adaptation and generalization to new users.

| Model | Seen Accuracy (%) | Unseen Accuracy (%) |
|---|---|---|
| P-DPO individual | 91.04 | 55.34 |
| P-DPO K=5 | 91.12 | 54.55 |
| Vanilla DPO | 58.91 | 55.37 |
| PAL-A-Tiny $K = 2, N = 10$ | | $50.12 \pm 1.20$ |
| PAL-A-Tiny $K = 2, N = 20$ | $52.91 \pm 0.55$ | $50.55 \pm 0.43$ |
| PAL-A-Tiny $K = 2, N = 50$ | | $50.49 \pm 0.39$ |
| PAL-A-Tiny $K = 2, N = 100$ | | $50.40 \pm 0.55$ |
| PAL-B-Tiny $K = 2, N = 10$ | | $74.88 \pm 0.79$ |
| PAL-B-Tiny $K = 2, N = 20$ | $79.54 \pm 0.54$ | $77.37 \pm 0.37$ |
| PAL-B-Tiny $K = 2, N = 50$ | | $76.19 \pm 0.49$ |
| PAL-B-Tiny $K = 2, N = 100$ | | $77.80 \pm 0.07$ |
| PAL-A-Large $K = 2, N = 10$ | | $73.39 \pm 1.82$ |
| PAL-A-Large $K = 2, N = 20$ | $90.00 \pm 1.89$ | $74.04 \pm 2.50$ |
| PAL-A-Large $K = 2, N = 50$ | | $72.33 \pm 2.36$ |
| PAL-A-Large $K = 2, N = 100$ | | $74.39 \pm 1.70$ |
| **PAL-B-Large $K = 2, N = 10$** | | $\mathbf{91.63 \pm 1.43}$ |
| **PAL-B-Large $K = 2, N = 20$** | $\mathbf{92.82 \pm 0.95}$ | $\mathbf{91.72 \pm 1.40}$ |
| **PAL-B-Large $K = 2, N = 50$** | | $\mathbf{92.02 \pm 1.10}$ |
| **PAL-B-Large $K = 2, N = 100$** | | $\mathbf{91.97 \pm 1.91}$ |

### D.3.2 CHOICE OF FOUNDATION MODEL

The choice of foundation model is an important factor that impacts the performance of PAL, especially for PAL-A-Tiny and PAL-A-Large. This is because the foundation model directly decides the quality of the embeddings we obtain for the items. Table D.4 illustrates that there is a performance gap between using OPT-350M and DistilBERT as the foundation model, especially for Tiny variants where we do not train the foundation model. The existence of such a gap is possibly due to the fact that DistilBERT is an encoder-based model, which provides a better sentence embedding than the decoder-based OPT-350M.

Table D.4: Comparison of performance of `PAL-A-Tiny` and `PAL-A-Large` with different foundation models on the Summary dataset.

| Model | Foundation Model | Seen User Test Accuracy |
|---|---|---|
| `PAL-A-Tiny` ($K = 2$) | `OPT-350M` | $52.91 \pm 0.55$ |
|  | `DistilBERT` | $72.99 \pm 1.21$ |
| `PAL-A-Large` ($K = 2$) | `OPT-350M` | $90 \pm 1.89$ |
|  | `DistilBERT` | $91.75 \pm 0.41$ |

Table D.5: Personas used across various "true" number of user groups $K^\star$ in our heterogeneous persona dataset.

| $K^\star$ | Personas |
|---|---|
| 2 | interest in art, interest in literature |
| 3 | interest in art, interest in literature, interest in math |
| 4 | interest in art, interest in literature, interest in math, interest in music |
| 5 | interest in art, interest in literature, interest in math, interest in music, interest in science |
| 6 | interest in art, interest in literature, interest in math, interest in music, interest in science, interest in sports |

### D.4 PERSONA (TEXT-TO-TEXT)

Anthropic's Persona dataset Perez et al. (2022) consists of a series of personalities (personas), each corresponding with 500 statements that agree with the persona and 500 statements that do not. We denote the set of statements that agrees with a persona $\rho$ as $S(\rho)$. We construct a semi-synthetic dataset using Anthropic's Persona to evaluate `PAL`.

**Dataset.** Let $\rho = \{\rho_1, \ldots, \rho_{K^\star}\}$ denote the set of personas that exists in our semi-synthetic heterogeneous dataset with $K^\star$ "true" preference (prototypical) groups i.e. each person (user) has one of the $K^\star$ personalities. For each $\rho_j \in \rho$, we generate $N$ synthetic *seen* and *unseen* users. For each seen synthetic user, we generate $n_p$ queries that ask if the user agrees with a given statement from the persona dataset. For each unseen synthetic user, we generate $n_{p,\text{unseen}}$ queries. If the statement aligns with the persona $\rho_j$ of the user, i.e. the statement belongs to $S(\rho_j)$, then the user answers yes, otherwise no. Table D.5 lists the personas used for each $K^\star$, and Figure D.4 shows a sample question.

**Experiment Setup.** We evaluate the performance of `PAL-A-Tiny` with hinge loss and model `PAL-B-Tiny` with logistic loss on the heterogeneous persona dataset in various settings. Both model utitlize a 2-layer MLP as the $f$ function. To examine the impact of various hyperparameters, we conduct experiments varying the number of true prototypes in the dataset $K^\star$, the number of prototypical groups used in the model $K$, queries per seen user $n_p$, and latent dimension $d$ with a fixed number of users per group $N = 10,000$. Details of the values for each hyperparameter used are listed below:

(a) varying $K^\star = \{2, \ldots, 6\}$ and $K = \{1, \ldots, 8\}$ while fixing $n_p = 1000$, $d = 16$,
(b) varying $n_p = \{75, 100, 200, 500, 1000\}$ and $K = \{1, \ldots, 5\}$ while fixing $K^\star = 4$, and $d = 16$,
(c) varying $d = \{4, 8, 16, 32, 64\}$ and $K = \{1, \ldots, 5\}$ while fixing $K^\star = 4$, and $n_p = 1000$,
(d) varying $n_{p,\text{unseen}} = \{1, 10, 20, 50, 100, 200, 500, 1000\}$ and $K = \{1, \ldots, 5\}$ while fixing $K^\star = 4$, $n_p = 1000$, and $d = 16$.

Both `PAL-A-Tiny` and `PAL-B-Tiny` used the same value, except for $d$. This is because `PAL-A` utilizes a residual connection. Therefore, the latent dimension is fixed to 768, the dimension of the input embedding.

**Results.** We repeat these experiments five times and report the results on `PAL-A-Tiny` in Figure D.5 and on `PAL-B-Tiny` in Figure D.6. Both Figure D.6 and Figure D.5 (a, b) illustrate the generalization performance of `PAL-A-Tiny` and `PAL-B-Tiny` on the heterogeneous persona dataset. We observe that as $K \to K^\star$, the seen accuracy increases to 100% given a sufficient number

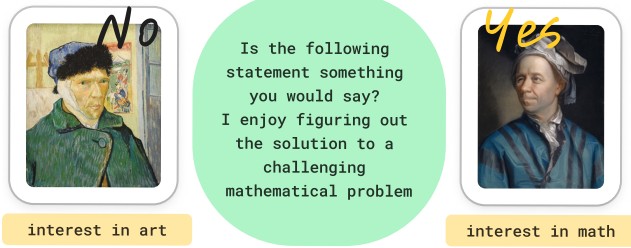

Figure D.4: An example of a pairwise comparison query with a prompt from our heterogeneous persona dataset generated using Anthropic's Personas. For example, a synthetic user assigned the persona *interest in art* will have ground truth $y = -1$ by answering no, whereas a synthetic user assigned the persona *interest in math* will have ground truth $y = +1$ by answering yes.

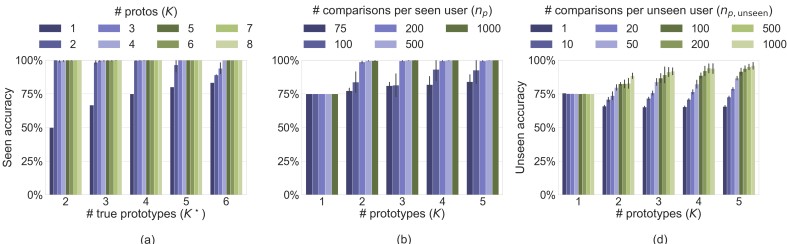

Figure D.5: Seen accuracy (a,b) and unseen accuracy (c) evaluated on the heterogeneous persona dataset across the number of prototypes $K$ used in PAL-A-Tiny. The number of true prototypes $K^\star$ in (b, c) is 4. We vary (a) the number of true prototypes $K^\star$, (b) the number of comparisons per seen user $n_p$, (c) the number of comparisons per unseen user $n_{p,\text{unseen}}$. Since we used a residual connection in the PAL-A-Tiny, we could not vary the size of the latent dimension.

of users and number of comparisons per user. Figure D.5 and Figure D.6 (b) shows that as we get more comparisons per user, we achieve better *seen user* accuracy, i.e. we can generalize to unseen pairs for users who are seen (provide training samples) in the dataset. Figure D.6 (c) shows that the size of latent dimension $d$ does not affect the seen accuracy dramatically. Figure D.5 (c) and Figure D.6 show the accuracy for *unseen users*, i.e., users who do not provide training samples. When $K = 1$, no further learning is needed to generalize to new users. However, when $K > 1$, we require weights over the $K$ prototypes for the new users to be learned. To learn these new user weights, as discussed in Section 2.2, we fix the $K$ prototypes and the mapping $f$ and use only a few test data samples to learn the user weights (**C4**). We use these learned weights to make predictions on the remaining test data. From Figure D.5 (c) and Figure D.6 (d), we see that for $K = 1$ the number of samples used to learn weights makes no difference since there are no weights to learn over a single prototype. For $K = 2$, we see that as we use more data for learning the new user weights, the performance shows diminishing returns until saturation. We also demonstrate that as the number of prototypes $K$ increases, more comparisons per user are needed to learn the new user weights, since the dimension of the weight vector increases with $K$.

## D.5 PICK-A-PIC (TEXT-TO-IMAGE)

**Dataset.** The Pick-a-Pic dataset is a large, open dataset designed to capture human preferences in text-to-image generation. It includes over 500,000 examples where users compare two AI-generated images based on a text prompt and choose their preferred one. This dataset is used to align models with human preferences.

**Experiment Setup.** We apply PAL-B-Tiny on the Pick-a-Pic dataset. Since the Pick-a-Pic dataset collection process requires strict rubrics, the labels collected from workers may not reflect the worker's diverse preferences. Thus we set $K = 1$ for the PAL model. We use two-layer MLP networks with ReLU activation and residual connections as the mapping functions. To avoid over-fitting we set the dropout rate to $0.5$ and weight decay to $1e - 2$. We apply different learning rates for various model components: $1e - 4$ for the two-layer MLPs and $5e - 3$ for user weights.

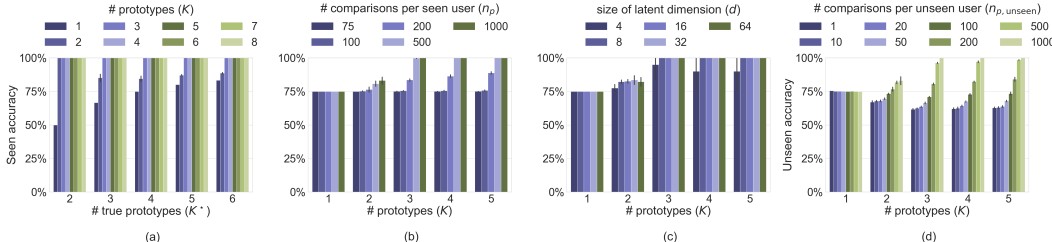

Figure D.6: Seen accuracy (a,b,c) and unseen accuracy (d) evaluated on the heterogeneous persona dataset across the number of prototypes $K$ used in the `PAL-B-Tiny`. The number of true prototypes $K^\star$ in (b, c, d) is 4. We vary (a) the number of true prototypes $K^\star$, (b) the number of comparisons per seen user $n_p$, (c) the size of latent dimension $d$, (d) the number of comparisons per unseen user $n_{p,\text{unseen}}$.

Table D.6: Number of samples in each split of the newly constructed Pick-a-Filter dataset.

| | Category | Train | Val | Test |
|---|---|---|---|---|
| | Seen | 58831 | 628 | 1597 |
| Group 1 | Unseen | 9527 | 79 | 1886 |
| | Total | 68358 | 707 | 3483 |
| | Seen | 57200 | 404 | 2096 |
| Group 2 | Unseen | 9402 | 52 | 1812 |
| | Total | 66602 | 456 | 3908 |

## D.6 PICK-A-FILTER (TEXT-TO-IMAGE)

**Dataset:** due to the high level of "agreement" among labelers over image preferences on Pick-a-Pic v1 (Kirstain et al., 2024), we construct a semi-synthetic dataset by applying filters to a subset of Pick-a-Pic v1, which we call the Pick-a-Filter dataset. To construct the dataset, we consider only samples that have no ties, i.e. the labeler decides that one image is decisively preferable to the other, given the text prompt. As Pick-a-Pic provides unique and anonymous user IDs for all preference pairs, we consider a subset of users who provide samples in **both** the train and test sets (468 / 4223 users). We further only consider users who provide more than 50 labels (234 / 468 users) and sort the users by number of samples provided. We split these users into equal groups of 117 each, and we assume without loss of generality that the first group of users (G1) prefers "cold" tones (blue filter) and the second group (G2) prefers "warm" tones (red filter). Lastly, we arbitrarily consider the first 50 users (who provide the most number of samples) as "seen" users, i.e. users that provide samples in both the train and test sets of Pick-a-Filter. We add this seen vs. unseen distinction to evaluate how well `PAL` can adapt to unseen (i.e. new) users after training. Currently, our experiments on Pick-a-Filter (Section 3.3) train on v1-train-seen (116031 samples) and evaluate on v1-test-seen (3693 samples). We show the number of samples in each of these splits in Table D.6. After constructing splits, we apply the following filtering logic:

1. Apply "winning" and "losing" filters to appropriate images depending on label. For G1 the winning filter is blue, and for G2 the winning filter is red.

2. Randomly shortlist $\beta\%$ of samples to add filters. The remaining $(1-\beta)\%$ of samples will remain unaltered (default images from Pick-a-Pic v1).

3. Randomly select $50\%$ of above-shortlisted samples to apply a filter to only the winning image, and the remaining $50\%$ to apply a filter to only losing image

We add these sources of randomness to make learning preferences on Pick-a-Filter less prone to hacking (e.g. the model could trivially learn to predict an image with a filter as the preferred image).

**Experiment Setup.** We choose 2-layer MLP networks with ReLU activation and residual connection as the prompt mapping function $g_k$ and the output mapping function $f$. To avoid overfitting, we set the dropout rate to $0.5$ and weight decay to $1e-2$. We use the Adam optimizer with a learning

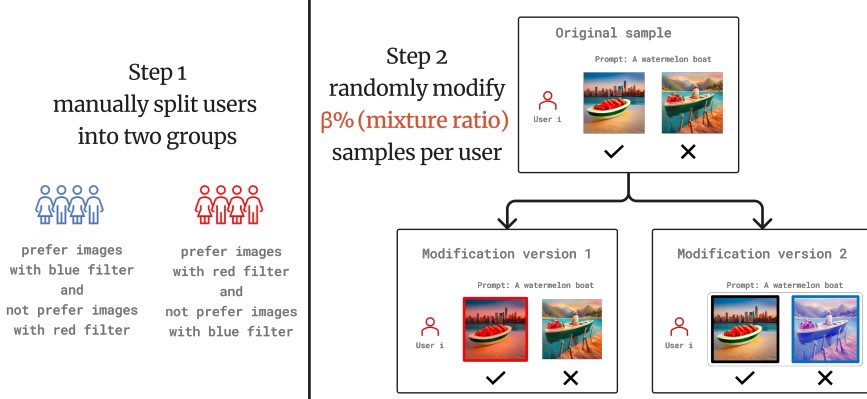

Figure D.7: The construction diagram for the semi-synthetic Pick-a-Filter dataset. It involves randomly selecting approximately 135,000 samples from the Pick-a-Pic v1 dataset and dividing the user IDs into two disjoint groups. We assume one group prefers images with "cold tone" (blue) filters and the other with "warm tone" (red) filters. To incorporate diverse color filter preferences, we randomly select $\beta\%$ of samples per user on which to apply filters.

rate $1e-4$. To evaluate the model's performance, we use the checkpoint with the highest accuracy on the validation set.

# E   COMPUTATIONAL RESOURCES

We conducted most of our experiments using $4\times$RTX 4090, each with 24 GB of VRAM. For the experiments involving a foundation model that has 1.3B parameters or more, we used $2 \times$ A100, each with 80GB of VRAM. A typical run of the experimentw finished within 2 hours.

