# OpenReview forum: "PAL: Sample-Efficient Personalized Reward Modeling for Pluralistic Alignment"
_ICLR.cc/2025/Conference — ICLR 2025 Poster_

### Official Review · Reviewer_g42M · 2024-10-30

**Soundness:** 3
**Presentation:** 2
**Contribution:** 3
**Rating:** 6
**Confidence:** 3

**Summary:**

The PAL framework aims to capture diverse human preferences from internet-scale data trained foundational models for pluralistic alignment. The modular design of the framework allows it to leverage commonalities among users while providing personalized services to each user, achieving efficient few-shot localization of preferences for new users. Through extensive empirical evaluation, PAL has demonstrated performance that matches or exceeds the state-of-the-art methods in both text-to-text and text-to-image tasks.

**Strengths:**

1. The PAL framework enables personalized reward modeling with fewer samples, which is highly beneficial for data collection and model training, especially when data annotation costs are high.
2. PAL has shown superior performance across multiple tasks, not only surpassing or matching the accuracy of existing state-of-the-art methods but also significantly reducing the number of parameters, demonstrating dual advantages in efficiency and effectiveness.
3. The paper not only provides empirical evaluations but also theoretical analyses, proving the sample complexity of PAL in generalizing to new users and unseen samples, providing a solid theoretical foundation for the model's reliability and effectiveness.

**Weaknesses:**

1. Figure 1, as the first graph of the paper, is somewhat confusing, with pasted images and formulas that are unclear. There is a lack of a clear caption explanation, and the introduction does not provide specific details, making it difficult to understand the meaning of the symbols in the graph or the order in which to view them.
2. The modeling of user preferences in the paper mainly focuses on the preference for summary length, which may not cover a broader range of personalized preferences. The preference distribution of seen and unseen users is limited to the preference for the length of summaries and may not be comprehensive enough.
3. The datasets used in the paper may lack sufficient diversity, which limits the model's generalization capabilities in a broader range of scenarios.

**Questions:**

See Weaknesses.

**Details Of Ethics Concerns:**

Please verify if the similar work at https://pal-alignment.github.io/ is by the same authors to check for plagiarism or ICLR's policy on prior publication at ICML workshops.

---

> ### Author Response · Authors · 2024-11-21
> **Author Response**
>
> Thank you for your insightful feedback. To follow up on your comments and questions:
>
> **"... focuses on the preference for summary length…"**: It seems that there is a misunderstanding here. While we applied our framework to model preferences on summary dataset, it is not developed specifically for this dataset.
>
> *Our reward modeling framework is general and versatile* – it can be applied across different domains as we demonstrate in both text-to-text (see Sections 3.1 and Appendix D.4) and text-to-image tasks (see Sections 3.2 and 3.3).
>
> We additionally evaluated our framework on the Pick-a-Pic dataset (text-to-image, Section 3.2), the Pick-a-Filter dataset (text-to-image, Section 3.3), and the Anthropic Persona dataset (text-to-text, Appendix D.4). In addition to the summary dataset, our experiment on the Anthropic Persona dataset also showcases PAL’s strong few-shot generalization to *unseen users* (see Figure D.5 and D.6); these results are deferred to the appendix in the interest of space.
>
> **"... diversity in datasets..."**:  We have performed extensive empirical evaluation on the datasets and benchmarks available. We also recognize and highlight the limitations of existing datasets (see our remark in Section 3.2). There is a need for new datasets and benchmarks for learning plurality of preferences and pluralistic alignment. This is, however, beyond the scope of this paper. That said, if the reviewer has specific datasets in mind, we would appreciate it if they point them to us.
>
> **"Figure 1 ... insufficient details"**:  Thank you for the helpful feedback. Could you please provide us additional feedback as to which aspects of the figure/caption were confusing or insufficient? This will help us make changes to the figure to improve clarity, and we will update Figure 1 in the next version accordingly.
>
> We hope that this response addresses your questions and concerns. We are happy to have further discussion to clarify any other concerns and improve the manuscript.

---

> ### Comment · Reviewer_g42M · 2024-11-23
>
> Thank you for your response, which has addressed some of my concerns. However, regarding Figure 1, I have the following suggestions: Figure 1 is a combination of multiple subfigures, but the authors did not explain the meaning of each subfigure individually. I recommend labeling each part as (a), (b), (c), (d), etc., and providing explanations for each part in the caption or referencing the corresponding subfigures (a), (b), (c), (d) in the relevant sections of the text. Currently, combining formulas and illustrations into the same figure makes it feel cluttered and confusing, leaving readers uncertain about how to interpret the figure.

---

> > ### Author Response · Authors · 2024-11-25
> >
> > Thank you for this valuable feedback. We have made a revision to simplify Figure 1 according to your suggestions and reduce visual clutter. Each part of the figure is now clearly labeled, and detailed explanations have been added to the caption. We are more than happy to address any further questions or suggestions to increase the score and facilitate acceptance of the paper.

---

> > > ### Comment · Reviewer_g42M · 2024-11-25
> > >
> > > Thanks to the author's reply, figure 1 is much clearer now and I've updated my scoring.

---

> > > > ### Author Response · Authors · 2024-11-25
> > > >
> > > > Dear Reviewer g42M,
> > > >
> > > > We appreciate your constructive feedback throughout the reviewing process and are glad to see your positive assessment. Thank you again for helping us improve our work!

---

### Official Review · Reviewer_Eo4u · 2024-11-01

**Soundness:** 3
**Presentation:** 3
**Contribution:** 3
**Rating:** 8
**Confidence:** 2

**Summary:**

This paper tackles the issue of personalized reward modeling. The authors recognize that different users may not generate a consistent preference across different context-sample pairs. They propose two novel modeling method that represent preference with a latent reward function. PAL-A assumes the reward of a sample-context pair is determined by its distance to the user's ideal point in a latent space. It further assumes that the ideal points of a population lies in the convex hull of several supporting points in the latent space, and we can recover an individual's personalized preference through weighted average of supporting points. PAL-B represents preference as unknown preference mapping, and commonalities are similarly modeled as a convex combination of K prototypical mapping.

The author conducted extensive experiments on both NLP and T2I generation domain, achieving SOTA results

**Strengths:**

The paper challenges a commonly overlooked assumption in the alignment literature, that individuals' may have a different preference.  The proposed two approaches are novel and clearly differentiates from existing works (e.g. KTO), which treat such inconsistencies as noises in preference data.  The presentation is clear and easy to follow, with motivations and formulations of the proposed method covered in detail. The experiments showed strong performance of the proposed method. Moreover, the proposed method can be trained on a single consumer grade GPU whereas the baselines are trained on multiple A100s.

**Weaknesses:**

1.Some experiment details are lossy and not well-documented.  Presentation is a bit unclear. For example, while the authors clearly documented the choice of base model and training data on Pick-v2 results (table 3), such information is not included in Pick-v1 (table 2). **What is the base model for results in table 2? Is it vanilla CLIP-H or PickScore?**

Without this knowledge, it is hard to evaluate the claim on parameter efficiency. For example, if v1 results are reported via a model that is fine-tuned on Pick-v1 embedding, then it is hard to argue that the model is more parameter efficient since it starts with a fully fine-tuned model. Overall, presentation wise, I think it would make more sense to add table 2 as an extra column in table 3, which would reduce many confusions on the setup.

2.Results on Pick-a-Filter are unconvincing.

2.1  Pickscore baseline is missing. While the authors claim that they cannot compare PickScore as its training set overlaps with Pick-a-Filter’s val set. However, this can be trivially fixed be eliminating the overlapped examples, which the authors already did for table 3. Why not do the same? Alternatively, given the large samples of pick-a-pick v2, it is not hard to construct a custom validation split that does not overlap with the training data of PickScore.

**Please either eliminate the overlapping examples from the Pick-a-Filter validation set and include PickScore as a baseline, or
construct a custom validation split from Pick-a-Pic v2 that doesn't overlap with PickScore's training data, and use this for comparison, or justify why these options are not possible**


2.2 The red and blue filter examples seems too rival, and I suspect the obvious color differences will overshadow the "commonalities" in preference.  I think the key benefits of the proposed method that it captures both the "common preferences" and "individual variations". However, for the color filters, but a naive color classifier may also achieve high accuracy in this example. It is unclear if the proposed method offer any benefits. Such comparison is required.

This is also highlighted in Fig 4, where differences in low beta region is unclear (side note: presentation wise this figure needs improvement. It is hard to tell which line is higher). I think the low beta region might be more representative of the actual discrepancies in human preferences. However, as Pickscore baseline is missing from Figure 4 (See comments in 2.1), it is hard to tell if PAL offer any benefits in this region. I image a proper pickscore comparison would be a flat line that resembles CLIP and HPSv2. The question is whether the line of PickScore would be higher than PAL in low beta region.

**I would highly appreciate it if the authors can provide more discussions on significance of results on Pick-a-Filter, particularly on the non-linear improvement in Figure 4.** It may seem that the model just collapse to a color classifier at high beta region.  **Authors should discuss if PAL is simply collapsing to a color classifier. I suggest authors compare PAL against a naive color classifier. I'm open to other means/discussion on this topic as well.**

**Questions:**

See weakness. Additional questions are

1. Why Reddit experiments did not include results of PAL-A? Am I missing anything?

---

> ### Author Response · Authors · 2024-11-21
> **Author Response**
>
> We thank the reviewer for their detailed feedback and positive review. We are glad the reviewer found our work **novel** and our motivations and manuscript **clear to follow**, and that the reviewer highlighted the **empirical performance** and **efficiency** of PAL compared to status quo methods. Following are some clarifications requested in the review:
>
>
> **Experiment Documentation**: To confirm, PAL models in Table 2 (old version) are trained on Pick-a-pic V1 with CLIP-H embeddings. We have merged Table 3 into Table 2 to eliminate this confusion - we thank the reviewer for this helpful suggestion. We are happy to make additional edits to Table 2 (new version) if we can further improve the clarity.
>
>
> **Missing Pickscore Baseline**: we have updated Figure 4 with the Pickscore baseline as suggested in weakness 2.1 - for reference, PickScore accuracy across beta values is $72.59 \pm 0.15$ %, which is nearly 2\% lower than PAL-B-Tiny even at low beta values ($74.23 \pm 0.34$ \% at $ \beta \leq 0.6$). We also add a zoomed in portion to Figure 4 to aid visualization at low beta values.
>
>
> **Pick-a-Filter experiment**: The Pick-a-pic dataset was created using a strict rubric for evaluation and therefore by design this dataset is homogeneous. The injected heterogeneity in Pick-a-Filter setting is designed to test the hypothesis that PAL is *able* to learn heterogeneity when it exists *without knowing what dictates heterogeneity a priori*. So, the goal here is not to create a specific filtered approach for this specific dataset. The key point here is that PAL doesn’t know a priori that color is a signal, but it learns to find the heterogeneity from the data.
>
> In addition to Pick-a-Filter, we also applied PAL in Gaussian data setting (Fig 5b) as well as text statements generated by different Personas (Appendix D.4, Figure D.5a), which provides evidence for PAL’s ability to adapt to many forms of heterogeneity without prior causal knowledge or needing specific hand-crafted signals.
>
>
> **PAL-A on Reddit TL;DR**: These results were deferred to Appendix D (Table 3, 5) due to space constraints (L292). We will clarify this in the final version of the paper.
>
> We hope this rebuttal addresses your questions and concerns. We look forward to further discussion to improve the manuscript.

---

> > ### Comment · Reviewer_Eo4u · 2024-11-25
> >
> > Thank authors for the responses. Most of my concerns have been addressed. I maintain my recommendation for acceptance. I especially appreciate the Author's effort to fix the table 2, as well as discussions of low beta region and the incorporation of a pick-a-pick baseline in Figure 4.

---

> > > ### Author Response · Authors · 2024-11-25
> > >
> > > Dear Reviewer Eo4u,
> > >
> > > We are glad that most of your concerns have been addressed, and we appreciate your constructive feedback throughout the review and rebuttal process and helping us improve our work. Thank you again!

---

### Official Review · Reviewer_rMSN · 2024-11-14

**Soundness:** 3
**Presentation:** 4
**Contribution:** 2
**Rating:** 6
**Confidence:** 3

**Summary:**

While it is well-known that humans have diverse preferences, most foundation model alignment methods assume homogeneous preference across all users. The authors propose a novel reward modeling framework to capture shared as well as personalized preferences to enable pluralistic alignment. The method's ability to combine global and individual preferences allows the method to perform well without requiring an overwhelming number of samples for each persona. The method is also demonstrated to outperform homogeneous reward models when diverse preferences are introduced to the evaluation dataset.

**Strengths:**

1. The method addresses the problem of pluralistic alignment, which is currently unaddressed by most alignment methods.
2. The method has a native flexibility to adjust the shared and personalized portions of the modeled preference.
3. The method complements existing alignment methods.
4. The paper rigorously explores the behavior of the method through simulated experiments and the results are visualized clearly, as are the algorithms. The experimental setup is clearly documented.
5. The method converges quickly to capture the preferences of unseen users

**Weaknesses:**

It does not seem like there is a benchmark that reflects real use cases that can highlight the benefits of having a heterogeneous model; both Reddit TL;DR and Pick-a-Filter are both semi-synthesized datasets that artificially accentuates the diversity of human preference. This calls into question whether pluralistic alignment is a valuable problem to solve.

**Questions:**

1. Are there benchmarks curated with real world data that can highlight the benefits of pluralistic alignment?
2. In section D.3.2, it is mentioned that the choice of foundation model greatly impacts the performance of the proposed method. It would be great to do an ablation study to investigate whether this effect is observed with other alignment methods as well.
3. When should one opt for either PAL A or PAL B?

---

> ### Author Response · Authors · 2024-11-21
> **Rebuttal by Authors**
>
> Thank you for your insightful feedback. To follow up on your comments and questions:
>
>
> **On the importance of pluralistic alignment**: We respectfully disagree with the reviewer. When “aligning” AI to human preferences, it is extremely important to incorporate heterogeneity/plurality of human preferences. People inherently have diverse values and preferences, and so developing methods that can accommodate this heterogeneity is crucial. This is highlighted by ongoing efforts from different communities. See, for example, this position paper at ICML 2024 (https://arxiv.org/pdf/2402.05070). A dedicated workshop on Pluralistic Alignment is also being held this year at NeurIPS (https://pluralistic-alignment.github.io). Pluralistic alignment is an emerging area of research and its importance is going to grow rapidly as we move towards deploying AI/ML models widely in society.
>
> A key part of incorporating plurality is modeling and learning plural preferences. Our work addresses this aspect. We propose a novel **general** reward modeling framework to learn heterogeneous preferences in a *sample-efficient* way. Our reward modeling approach is *modular* and *applicable to many domains*. Whenever heterogeneity is present, our framework can adapt to it well — it achieves state-of-the-art performance, and we also present analysis of sample complexity per user needed for generalization.
>
> The current datasets for preference alignment are limited in their quality for benchmarking pluralistic alignment. We highlight this issue and the need for improved data collection methods in the paper; see our remark in Section 3.2. It is a very valuable future direction to establish new datasets and benchmarks. In fact, our theoretical analysis on the number of samples needed per user for generalization sheds light on the amount of data per user needed to capture the heterogeneity in preferences.
>
> &nbsp;
>
> **Impact of the choice of foundation models**: A key strength of our PAL framework lies in its modular and versatile design. Our reward modeling mechanism can flexibly integrate with **any** foundation model. This enables us to perform systematic experiments to understand the effect of embeddings coming from various foundation models as shown in Figure 2.
>
> We believe the foundation model used for initial representations does have an impact on downstream reward learning in general. However, such modularity is not necessarily a feature of other reward modeling approaches; in addition, many of them are not openly available for ablation studies.
>
> &nbsp;
>
> **PAL-A vs PAL-B in practice**: This is a modeling choice. We note that PAL-B is more natural in generative modeling settings, as it learns a personalized mapping, $z^{(i)}(x_c)$, for each user $i$ and any given prompt $x_c$, and learns a separate mapping for outputs $x$. In contrast, PAL-A learns an ideal point for each user $i$ fixed across all prompts and it learns to jointly map the prompt $x_c$ and output $x$ in the same space.
>
> From experiments, we found that PAL-B serves as a reliable default choice (see Figure 2, 3 and Tab. 1 on TL;DR; Tab 2 on Pick-a-Pic and Figure 4 on Pick-a-Filter). We are able to get PAL-A to work competitively to PAL-B in most settings (see Table 3, 5 and Sec. D.4.); however, in practice, this may require additional engineering effort to optimize effectively.
>
> &nbsp;
>
> We hope that this response addresses your questions and concerns. We are happy to have further discussion to clarify any other concerns and improve the manuscript.

---

> > ### Author Response · Authors · 2024-11-25
> >
> > Dear Reviewer rMSN,
> >
> > We thank you again for your time and efforts in reviewing our work.
> >
> > As the discussion period draws to a close very soon, we would greatly appreciate the opportunity to address any additional concerns or suggestions you may have.
> >
> > Thank you,
> > The authors

---

> ### Comment · Reviewer_rMSN · 2024-11-26
>
> I thank the authors for the thorough response, which addresses all of my concerns. In particular, I appreciate that the authors have brought my attention to the recent developments in the area of pluralistic alignment as well as the remark in section 3.2 about the difficulty of collecting data to benchmark alignment.
>
> I would like to offer a couple of suggestions for clarity:
> 1. Have a brief section that discusses the choice of PAL-A and PAL-B.
> 2. Highlight the difficulty of procuring data for evaluation earlier in section 3 and justify the data synthesis approaches.

---

> > ### Author Response · Authors · 2024-11-28
> >
> > Dear reviewer rMSN,
> >
> > Thank you again for your valuable suggestions. We have revised our paper as follows:
> >
> > 1. We included a discussion on the modeling choices between PAL-A and PAL-B in Appendix B.2, referenced in Section 2 where the two models are introduced (footnote 3 below L161).
> >
> > 2. We provided a description on the datasets used in our experiments at the beginning of Section 3 (L240-243), where we briefly introduce the challenges in data collection and the need for creating semi-synthetic datasets; these are discussed in detail later throughout Section 3 (L322-340).
> >
> > We appreciate your insightful and constructive feedback throughout the reviewing process. Thank you for helping us improve our work!

---

### Author Response · Authors · 2024-11-21
**General Response to Reviews**

We thank all reviewers for their insightful feedback. We will use these suggestions to improve our work. Before providing individual responses, we first summarize the strengths highlighted by the reviewers:

(1) **Novel personalized reward modeling** (g42m, Eo4u, rMSN): Reviewers appreciate our novel PAL framework that addresses the pluralistic alignment problem.

(2) **Strong performance with fewer parameters** (Eo4u, g42M): Our methods match or outperform the status quo methods but require much fewer parameters, and they can be deployed on consumer grade GPU.

(3) **Superior Sample Efficiency** (rMSN, g42M): Our methods achieve superior sample efficiency for both seen and unseen user generalization.

(4) **Thorough simulation and theoretical study** (g42M, rMSN): Reviewers appreciate that we not only provide empirical results, but also include simulated experiments and theoretical analysis of per-user sample complexity.

(5) **Clarity** (Eo4u, rMSN): Reviewers found our paper well-written and easy to follow.

**Our contributions**: In this paper, we propose a **novel** and **general** reward modeling framework for learning heterogeneous preferences for pluralistic alignment with strong mathematical foundations. Our PAL framework is **modular** and **versatile**. It can capture diverse, personalized preferences, and it enables few-shot localization of preferences to new, unseen users. **Empirically**, we demonstrated that our framework *outperforms state-of-the-art methods* in both text-to-text and text-to-image tasks, while using *significantly fewer learnable parameters*. **Theoretically**, we established sample complexity guarantees for generalization on the amount of per-user data needed, both for seen and unseen users.

We look forward to addressing any further questions during the discussion period.

---

### Meta-Review · Area_Chair_aA1N · 2024-12-18

**Metareview:**

This paper proposed PAL: a sample efficient personalized reward modeling for learning heterogeneous preferences for pluralistic alignment.

Reviewers agrees on the novelty of the proposed method, strong performance in experiments, theoretical analysis besides empirical evaluations. Major concerns raised by reviewers are: data set (simulated data only w/o no real benchmark data, not diverse enough), or some experiments can be more convincing, clarity of some technical or experiment parts, etc.

Given its good quality (sufficient novelty, solid experiments results, on a relatively new research direction), I decided to accept this paper.

**Additional Comments On Reviewer Discussion:**

Reviewer rMSN raised the concern about the importance of pluralistic alignment and the author addressed it by providing recent papers and workshops on this topic. Reviewer rMSN also questioned about some technical parts (e.g., impact of choice of foundation models). Reviewer Eo4u mainly raised the concerns about some experiment results, which are addressed by authors with clarification or additional results. Reviewer g42M asked questions about some parts of the proposed techniques (e.g., whether the preference studied is about summary length, insufficient details of figure 1 etc), and authors provided clarifications for these issues.

---

### Decision · Program_Chairs · 2025-01-22

Accept (Poster)